# If your data distribution shifts, use self-learning

**Evgenia Rusak**[*]                                                      *evgenia.rusak@bethgelab.org*
*University of Tübingen*

**Steffen Schneider**[*†]                                                *steffen.schneider@bethgelab.org*
*University of Tübingen*

**George Pachitariu**                                                    *george.pachitariu@tue.ai*
*University of Tübingen*

**Luisa Eck**                                                            *luisa.eck@physics.ox.ac.uk*
*University of Oxford*

**Peter Gehler**                                                         *pgehler@amazon.com*
*Amazon Tübingen*

**Oliver Bringmann**                                                     *oliver.bringmann@uni-tuebingen.de*
*University of Tübingen*

**Wieland Brendel**[‡]                                                   *wieland.brendel@tuebingen.mpg.de*
*Max-Planck Institute for Intelligent Systems Tübingen*

**Matthias Bethge**[‡]                                                   *matthias.bethge@bethgelab.org*
*University of Tübingen*

**Reviewed on OpenReview:** *https://openreview.net/forum?id=XXXX*

## Abstract

We demonstrate that self-learning techniques like entropy minimization and pseudo-labeling are simple and effective at improving performance of a deployed computer vision model under systematic domain shifts. We conduct a wide range of large-scale experiments and show consistent improvements irrespective of the model architecture, the pre-training technique or the type of distribution shift. At the same time, self-learning is simple to use in practice because it does not require knowledge or access to the original training data or scheme, is robust to hyperparameter choices, is straight-forward to implement and requires only a few adaptation epochs. This makes self-learning techniques highly attractive for any practitioner who applies machine learning algorithms in the real world. We present state-of-the-art adaptation results on CIFAR10-C (8.5% error), ImageNet-C (22.0% mCE), ImageNet-R (17.4% error) and ImageNet-A (14.8% error), theoretically study the dynamics of self-supervised adaptation methods and propose a new classification dataset (ImageNet-D) which is challenging even with adaptation.

---

[*]Equal contribution.
[†]Work initiated during an internship at Amazon Tübingen.
[‡]Joint senior authors.

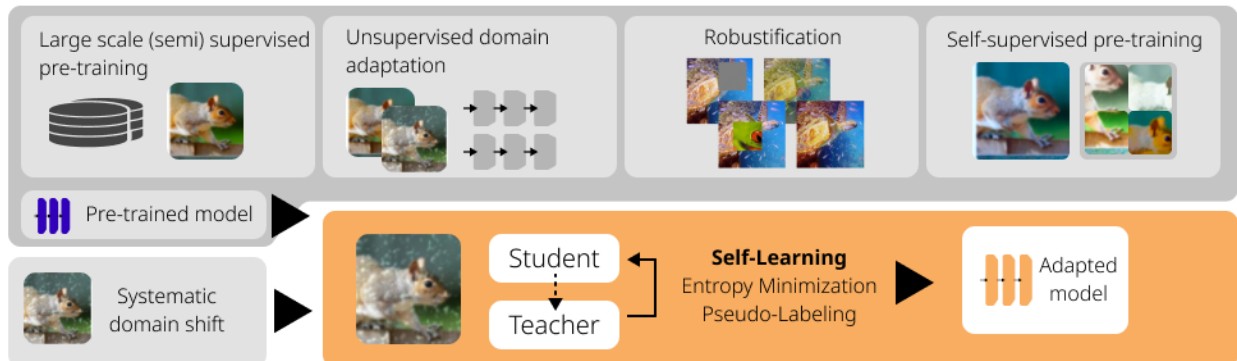

Figure 1: Robustness and adaptation to new datasets has traditionally been achieved by robust pre-training (with hand-selected/data-driven augmentation strategies, or additional data), unsupervised domain adaptation (with access to unlabeled samples from the test set), or, more recently, self-supervised learning methods. We show that on top of these different pre-training tasks, it is always possible (irrespective of architecture, model size or pre-training algorithm) to further adapt models to the target domain with simple self-learning techniques.

## 1 Introduction

Deep Neural Networks (DNNs) can reach human-level performance in complex cognitive tasks (Brown et al., 2020; He et al., 2016a; Berner et al., 2019) if the distribution of the test data is sufficiently similar to the training data. However, DNNs are known to struggle if the distribution of the test data is shifted relatively to the training data (Geirhos et al., 2018; Dodge & Karam, 2017).

Two largely distinct communities aim to increase the performance of models under test-time distribution shifts: The *robustness community* generally considers ImageNet-scale datasets and evaluates models in an *ad-hoc* scenario. Models are trained on a clean source dataset like ImageNet (Deng et al., 2009), using heavy data augmentation (Hendrycks et al., 2020a; Rusak et al., 2020; Geirhos et al., 2019) and/or large-scale pre-training (Xie et al., 2020a; Mahajan et al., 2018). The trained models are not adapted in any way to test-time distribution shifts. This evaluation scenario is relevant for applications in which very different distribution shifts are encountered in an unpredictable order, and hence misses out on the gains of adaptation to unlabeled samples of the target distribution.

The *unsupervised domain adaptation (UDA) community* often considers smaller-scale datasets and assumes that both the source and the (unlabeled) target dataset are known. Models are trained on both datasets, e.g., with an adversarial objective (Ganin et al., 2016; Tzeng et al., 2017; Hoffman et al., 2018), before evaluation on the target domain data. This evaluation scenario provides optimal conditions for adaptation, but the reliance on the source dataset makes UDA more computationally expensive, more impractical and prevents the use of pre-trained models for which the source dataset is unknown or simply too large. We refer the reader to Farahani et al. (2021) for a review of UDA.

In this work, we consider the *source-free domain adaptation setting*, a middle ground between the classical ad-hoc robustness setting and UDA in which models can adapt to the target distribution but without using the source dataset (Kundu et al., 2020; Kim et al., 2021; Li et al., 2020; Liang et al., 2020). This evaluation scenario is interesting for many practitioners and applications as an extension of the ad-hoc robustness scenario. It evaluates the possible performance of a *deployed* model on a systematic, unseen distribution shift at inference time: an embedded computer vision system in an autonomous car should adapt to changes without being trained on all available training data; an image-based quality control software may not necessarily open-source the images it has been trained on, but still has to be adapted to the lighting conditions at the operation location; a computer vision system in a hospital should perform robustly when tested on a scanner different from the one used for producing the training images—importantly, it might not be known at development time which scanner the vision system will be tested on, and it might be prohibited to share images from many hospitals to run UDA.

Can self-learning methods like *pseudo-labeling* and *entropy-minimization* also be used in this *source-free* domain adaptation setting?[1] To answer this question, we perform an extensive study of several self-learning variants, and find consistent and substantial gains in test-time performance across several robustness and out-of-domain benchmarks and a wide range of models and pre-training methods, including models trained with UDA methods that do not use self-learning, see Figure 1. We also find that self-learning outperforms state-of-the-art source-free domain adaptation methods, namely Test-Time Training which is based on a self-supervised auxiliary objective and continual training (Sun et al., 2020), test-time entropy minimization (Wang et al., 2021), Meta Test-Time Training (Bartler et al., 2022), and (gradient-free) BatchNorm adaptation (Schneider et al., 2020; Nado et al., 2020), and can be improved with additional techniques such as diversity regularization (Mummadi et al., 2021). We perform a large number of ablations to study important design choices for self-learning methods in source-free domain adaptation. Furthermore, we show that a variant of pseudo-labeling with a robust loss function consistently outperforms entropy minimization on ImageNet-scale datasets.

We begin by positioning our work in the existing literature (§ 2) and proceed with an overview of various self-learning variants that have been applied over the past years, and propose a new technique for robust pseudo-labeling (§ 3). We then outline a rigorous experimental protocol that aims to highlight the strengths (and shortcomings) of various self-learning methods. We test various model architectures, with different pre-training schemes covering the most important models in unsupervised domain adaptation, robustness, and large-scale pre-training (§ 4). Using this protocol, we show the effectiveness of self-learning across architectures, models and pre-training schemes (§ 5). We proceed with an in-depth analysis of self-learning, both empirical (§ 6) and theoretical (§ 7). Since the outlined results on ImageNet-C (22.0% mCE), ImageNet-R (17.4% error) and ImageNet-A (14.8%) approach clean performance (11.6% error for our baseline), we propose ImageNet-D as a new benchmark, which we analyse in § 8. We conclude by proposing a set of best practices for evaluating test-time adaptation techniques in the future to ensure scientific rigor and to enable fair model and method comparisons (§ 9).

## 2 Related Work

**Test-Time Adaptation** The main question we ask in this work is whether self-learning methods such as entropy minimization and different variants of pseudo-labeling can improve the performance of models when adapted at test-time to data with a distribution shift relative to the training data. Our work is most similar to test-time entropy minimization (TENT; Wang et al., 2021) since entropy minimization is one of our studied test-time adaptation techniques. The conceptual difference between our experiments and Wang et al. (2021) is that Wang et al. (2021) compare TENT to UDA while we argue that UDA can be regarded as a pretraining step, and self-learning can be used on top of any checkpoint pretrained with UDA. Wang et al. (2021) study the effectiveness of entropy minimization across different models, datasets and tasks. We expand upon their experimental setup and show that entropy minimization is effective across a large range of model architectures (convolutional neural networks, Vision Transformers (Dosovitskiy et al., 2021; Caron et al., 2021)), sizes (ResNet50 to EfficientNet-L2 (Xie et al., 2020a; Tan & Le, 2019)), and are orthogonal to robustification (e.g., DeepAugment (Hendrycks et al., 2020a)) and other pre-training schemes. Finally, we perform a large hyperparameter study for test-time entropy minimization, and thereby further improve upon the results reported by Wang et al. (2021).

We also compare our self-learning results to gradient-free adaptation of batch normalization (BN; Ioffe & Szegedy, 2015) statistics (BN adapt; Schneider et al., 2020) who proposed re-estimating the BN statistics on the shifted data distribution.

In Test-Time Training (TTT; Sun et al., 2020), the softmax cross-entropy loss is combined with a rotation prediction task (Gidaris et al., 2018) during pretraining on the source dataset. At test-time, the model is fine-tuned on the unlabeled test data with the self-supervised task. Sun et al. (2020) report results when

---

[1] Self-learning was defined by Tsypkin (1968) to denote "learning when there is no external indication concerning the correctness of the response of the automatic system to the presented patterns". We opted to use this term to name the superset of pseudo-labeling, entropy minimization and self-supervised learning variants to highlight the fact these methods do not require ground truth labels for adaptation.

adapting to a single test example, and also for online adaptation where the model successively adapts to a stream of data, where the data can either come from the same or a gradually changing distribution. We added a detailed comparison to Sun et al. (2020) in Appendix C.6. Liu et al. (2021) (TTT+++) replace the rotation prediction task with SimCLR (Chen et al., 2020a), and find this modification improves performance.

Eastwood et al. (2022) propose Feature Restoration where the approximate feature distribution under the target data is realigned with the feature distribution under source data. Eastwood et al. (2022) report results on CIFAR10 (among other datasets) for a ResNet18 architecture which allows us to include their method as a baseline. In a concurrent publication, Niu et al. (2022) propose an anti-forgetting test-time adaptation method called EATA, which combine sample-efficient entropy minimization with anti-forgetting weight regularization. They report accuracy numbers on ImageNet-C for a ResNet50 which we include as a baseline. Mummadi et al. (2021) introduce a novel loss to stabilize entropy minimization by replacing the entropy by a non-saturating surrogate and a diversity regularizer based on batch-wise entropy maximization that prevents convergence to trivial collapsed solutions. To partially undo the distribution shift at test time, they additionally propose to add an input transformation module to the model. Mummadi et al. (2021) report results on the highest severity of ImageNet-C for a ResNet50 which we include as a comparison to our results. We note that Mummadi et al. (2021) constitute concurrent unpublished work.

Meta Test-Time Training (MT3; Bartler et al., 2022) combine meta-learning, self-supervision and test-time training to adapt a model trained on clean CIFAR10 (Krizhevsky et al., 2009) to CIFAR10-C (Hendrycks & Dietterich, 2019). We added a detailed comparison to Bartler et al. (2022) in Appendix C.7.

The following papers also consider the setting of test-time adaptation, but are not used as direct baselines in our work, because they study other datasets or tasks. Azimi et al. (2022) show performance improvements when using test-time adaptation on video data. They show adaptation results for the popular test-time adaptation techniques of BN adaptation (Schneider et al., 2020), Test-Time Training (Sun et al., 2020) and test-time entropy minimization (Wang et al., 2021). MEMO (Zhang et al., 2021) maximizes the prediction consistency of different augmented copies regarding a given test sample. We do not consider the single-sample adaptation setting, and comparing our self-learning techniques to MEMO is not a fair setting for MEMO; unsurprisingly, techniques benefiting from multiple samples such as TENT (Wang et al., 2021) outperform MEMO. AdaContrast by Chen et al. (2022) combines pseudo-labeling with other techniques, such as self-supervised contrastive learning on the target domain, soft k-nearest neighbors voting to stabilize the pseudo-labels, as well as consistency and diversity regularization. A direct comparison with their results is difficult because they evaluate on VISDA-C and DomainNet, and so we would need to train our methods on either of the datasets and perform a full hyperparameter search for a fair comparison. We do have one point of comparison: in their paper, Chen et al. (2022) perform better than TENT (Wang et al., 2021). However, we note that Chen et al. (2022) used the default hyperparameters of TENT and did not perform hyperparameter tuning on the new dataset, thus, we would expect the performance of properly tuned TENT to be better than reported in the paper. We expect the additional changes of AdaContrast to further improve upon our simple self-learning baselines. Our work is conceptually similar to virtual adversarial domain adaptation in the fine-tuning phase of DIRT-T (Shu et al., 2018). In contrast to DIRT-T, our objective is simpler and we scale the approach to considerably larger datasets on ImageNet scale. Iwasawa & Matsuo (2021) propose a test-time adaptation algorithm for the task of domain generalization based on computing the distance of each test sample and pseudo-prototypes for each class. Kumar et al. (2020) study the setting of self-learning for gradual domain adaptation. They find that self-learning works better if the data distribution changes slowly. The gradual domain adaptation setting differs from ours: instead of a gradual shift over time, we focus on a fixed shift at test time.

**Self-learning for domain adaptation** Xie et al. (2020b) introduce "In-N-Out" which uses auxiliary information to boost both in- and out-of-distribution performance. AdaMatch (Berthelot et al., 2021) builds upon FixMatch (Sohn et al., 2020) and can be used for the tasks of unsupervised domain adaptation, semi-supervised learning and semi-supervised domain adaptation as a general-purpose algorithm. Prabhu et al. (2021) propose SENTRY, an algorithm based on judging the predictive consistency of samples from the target domain under different image transformations. Zou et al. (2019) show that different types of confidence regularization can improve the performance of self-learning. A theoretically motivated framework

for self-learning in domain adaptation based on consistency regularization has been proposed by Wei et al. (2020) and then extended by Cai et al. (2021).

The main differences from these works to ours are that they 1) utilize both source and target data during training (i.e., the classical UDA setup) whereas we only require access to unlabeled target data (source-free setup), 2) train their models from scratch whereas we adapt pretrained checkpoints to the unlabeled target data, and 3) are oftentimes more complicated (also in terms of the number of hyperparameters) than our approach due to using more than one term in the objective function. We would like to highlight that utilizing source data should always result in better performance compared to not using source data. Our contribution is to show that self-learning can still be very beneficial with a small compute budget and no access to source data. Our setup targets "deployed systems", e.g., a self-driving car or a detection algorithm in a production line which adapts to the distribution shift "on-the-fly" and cannot (or should not) be retrained from scratch for every new domain shift.

**Model selection**   Gulrajani & Lopez-Paz (2021) show that model selection for hyperparameter tuning is non-trivial for the task of domain generalization, and propose model selection criteria under which models should be selected for this task. Following their spirit, we identify a model selection criterion for test-time adaptation, and rigorously use it in all our experiments. We outperform state-of-the-art techniques which did not disclose their hyperparameter selection protocols.

## 3   Self-learning for Test-Time Adaptation

Tsypkin (1968) defines self-learning as "learning when there is no external indication concerning the correctness of the response of the automatic system to the presented patterns", and thus, we use this term as a superset of different variants of pseudo-labeling and entropy minimization to highlight that these methods can be used for adaptation to unlabeled data. Different versions of self-learning have been used in both unsupervised domain adaptation (French et al., 2018; Shu et al., 2018), self-supervised representation learning (Caron et al., 2021), and in semi-supervised learning (Xie et al., 2020a). In a typical self-learning setting, a *teacher* network $\mathbf{f}^t$ trained on the source domain predicts labels on the target domain. Then, a *student* model $\mathbf{f}^s$ is fine-tuned on the predicted labels.

In the following, let $\mathbf{f}^t(\mathbf{x})$ denote the logits for sample $\mathbf{x}$ and let $p^t(j|\mathbf{x}) \equiv \sigma_j(\mathbf{f}^t(\mathbf{x}))$ denote the probability for class $j$ obtained from a softmax function $\sigma_j(\cdot)$. Similarly, $\mathbf{f}^s(\mathbf{x})$ and $p^s(j|\mathbf{x})$ denote the logits and probabilities for the student model $\mathbf{f}^s$. For all techniques, one can optionally only admit samples where the probability $\max_j p^t(j|\mathbf{x})$ exceeds some threshold. We consider three popular variants of self-learning: Pseudo-labeling with hard or soft labels, as well as entropy minimization.

**Hard Pseudo-Labeling (Lee, 2013; Galstyan & Cohen, 2007).** We generate labels using the teacher and train the student on pseudo-labels $i$ using the softmax cross-entropy loss,

$$\ell_H(\mathbf{x}) := -\log p^s(i|\mathbf{x}), \quad i = \mathrm{argmax}_j\, p^t(j|\mathbf{x}) \tag{1}$$

Usually, only samples with a confidence above a certain threshold are considered for training the student. We test several thresholds but note that thresholding means discarding a potentially large portion of the data which leads to a performance decrease in itself. The teacher is updated after each epoch.

**Soft Pseudo-Labeling (Lee, 2013; Galstyan & Cohen, 2007).** In contrast to the hard pseudo-labeling variant, we here train the student on class probabilities predicted by the teacher,

$$\ell_S(\mathbf{x}) := -\sum_j p^t(j|\mathbf{x}) \log p^s(j|\mathbf{x}). \tag{2}$$

Soft pseudo-labeling is typically not used in conjunction with thresholding, since it already incorporates the certainty of the model. The teacher is updated after each epoch.

**Entropy Minimization (ENT; Grandvalet & Bengio, 2004; Wang et al., 2021).** This variant is similar to soft pseudo-labeling, but we no longer differentiate between a teacher and student network. It

corresponds to an "instantaneous" update of the teacher. The training objective becomes

$$\ell_E(\mathbf{x}) := -\sum_j p^s(j|\mathbf{x}) \log p^s(j|\mathbf{x}). \tag{3}$$

Intuitively, self-learning with entropy minimization leads to a sharpening of the output distribution for each sample, making the model more confident in its predictions.

**Robust Pseudo-Labeling (RPL).** Virtually all introduced self-learning variants use the softmax cross-entropy classification objective. However, the softmax cross-entropy loss has been shown to be sensitive to label noise (Zhang & Sabuncu, 2018; Zhang et al., 2017). In the setting of domain adaptation, inaccuracies in the teacher predictions and, thus, the labels for the student, are inescapable, with severe repercussions for training stability and hyperparameter sensitivity as we show in the results.

As a straight-forward solution to this problem, we propose to replace the cross-entropy loss by a robust classification loss designed to withstand certain amounts of label noise (Ghosh et al., 2017; Song et al., 2020; Shu et al., 2020; Zhang & Sabuncu, 2018). A popular candidate is the *Generalized Cross Entropy (GCE)* loss which combines the noise-tolerant Mean Absolute Error (MAE) loss (Ghosh et al., 2017) with the CE loss. We only consider the hard labels and use the robust GCE loss as the training loss for the student,

$$i = \operatorname{argmax}_j p^t(j|\mathbf{x}), \quad \ell_{GCE}(\mathbf{x}, i) := q^{-1}(1 - p^s(i|\mathbf{x})^q), \tag{4}$$

with $q \in (0, 1]$. For the limit case $q \to 0$, the GCE loss approaches the CE loss and for $q = 1$, the GCE loss is the MAE loss (Zhang & Sabuncu, 2018). We test updating the teacher both after every update step of the student (RPL) and once per epoch (RPL$^{\text{ep}}$).

**Adaptation parameters.** Following Wang et al. (2021), we only adapt the affine scale and shift parameters $\gamma$ and $\beta$ following the batch normalization layers (Ioffe & Szegedy, 2015) in most of our experiments. We verify that this type of adaptation works better than full model adaptation for large models in an ablation study in Section 6.

**Additional regularization in self-learning** Different regularization terms have been proposed as a means to stabilize entropy minimization. Niu et al. (2022) propose an anti-forgetting weight regularization term, Liang et al. (2020); Mummadi et al. (2021); Chen et al. (2022) add a diversity regularizer, and Chen et al. (2022) use an additional consistency regularizer. These methods show improved performance with these regularization terms over simple entropy minimization, but also introduce additional hyperparameters, the tuning of which significantly increases compute requirements. In this work, we do not experiment with additional regularization, as the main point of our analysis is to show that pure self-learning is effective at improving the performance over the unadapted model across model architectures/sizes and pre-training schemes. For practitioners, we note that regularization terms can further improve the performance if the new hyperparameters are tuned properly.

## 4 Experiment design

**Datasets.** ImageNet-C (IN-C; Hendrycks & Dietterich, 2019) contains corrupted versions of the 50 000 images in the ImageNet validation set. There are fifteen test and four hold-out corruptions, and there are five severity levels for each corruption. The established metric to report model performance on IN-C is the mean Corruption Error (mCE) where the error is normalized by the AlexNet error, and averaged over all corruptions and severity levels, see Eq. 20, Appendix C.1. ImageNet-R (IN-R; Hendrycks et al., 2020a) contains 30 000 images with artistic renditions of 200 classes of the ImageNet dataset. ImageNet-A (IN-A; Hendrycks et al., 2019) is composed of 7500 unmodified real-world images on which standard ImageNet-trained ResNet50 (He et al., 2016b) models yield chance level performance. CIFAR10 (Krizhevsky et al., 2009) and STL10 (Coates et al., 2011) are small-scale image recognition datasets with 10 classes each, and training sets of 50 000/5000 images and test sets of 10 000/8000 images, respectively. The digit datasets MNIST (Deng, 2012) and MNIST-M (Ganin et al., 2016) both have 60 000 training and 10 000 test images.

Table 1: Self-learning decreases the error on ImageNet-scale robustness datasets. Robust pseudo-labeling generally outperforms entropy minimization.

| mCE [%] on IN-C test ($\searrow$) | number of parameters | w/o adapt | w/ adapt ($\Delta$) RPL | w/ adapt ($\Delta$) ENT |
|---|---|---|---|---|
| ResNet50 vanilla (He et al., 2016b) | $2.6 \times 10^7$ | 76.7 | 50.5 (-26.2) | 51.6 (-25.1) |
| ResNet50 DAug+AM (Hendrycks et al., 2020a) | $2.6 \times 10^7$ | 53.6 | 41.7 (-11.9) | 42.6 (-11.0) |
| DenseNet161 vanilla (Huang et al., 2017) | $2.8 \times 10^7$ | 66.4 | 47.0 (-19.4) | 47.7 (-18.7) |
| ResNeXt101$_{32 \times 8d}$ vanilla (Xie et al., 2017) | $8.8 \times 10^7$ | 66.6 | 43.2 (-23.4) | 44.3 (-22.3) |
| ResNeXt101$_{32 \times 8d}$ DAug+AM (Hendrycks et al., 2020a) | $8.8 \times 10^7$ | 44.5 | 34.8 (-9.7) | 35.5 (-9.0) |
| ResNeXt101$_{32 \times 8d}$ IG-3.5B (Mahajan et al., 2018) | $8.8 \times 10^7$ | 51.7 | 40.9 (-10.8) | 40.8 (-10.9) |
| EfficientNet-L2 Noisy Student (Xie et al., 2020a) | $4.8 \times 10^8$ | 28.3 | **22.0** (-6.3) | 23.0 (-5.3) |
| top1 error [%] on IN-R ($\searrow$) | | | | |
| ResNet50 vanilla (He et al., 2016b) | $2.6 \times 10^7$ | 63.8 | 54.1 (-9.7) | 56.1 (-7.7) |
| EfficientNet-L2 Noisy Student (Xie et al., 2020a) | $4.8 \times 10^8$ | 23.5 | **17.4** (-6.1) | 19.7 (-3.8) |
| top1 error [%] on ImageNet-A ($\searrow$) | | | | |
| EfficientNet-L2 Noisy Student (Xie et al., 2020a) | $4.8 \times 10^8$ | 16.5 | **14.8** (-1.7) | 15.5 (-1.0) |

**Hyperparameters.** The different self-learning variants have a range of hyperparameters such as the learning rate or the stopping criterion. Our goal is to give a realistic estimation on the performance to be expected in practice. To this end, we optimize hyperparameters for each variant of pseudo-labeling on a hold-out set of IN-C that contains four types of image corruptions ("speckle noise", "Gaussian blur", "saturate" and "spatter") with five different strengths each, following the procedure suggested in Hendrycks & Dietterich (2019). We refer to the hold-out set of IN-C as our *dev* set. On the small-scale datasets, we use the hold-out set of CIFAR10-C for hyperparameter tuning. On all other datasets, we use the hyperparameters obtained on the hold-out sets of IN-C (for large-scale datasets) or CIFAR10-C (on small-scale datasets).

**Models for ImageNet-scale datasets.** We consider five popular model architectures: ResNet50 (He et al., 2016b), DenseNet161 (Huang et al., 2017), ResNeXt101 (Xie et al., 2017), EfficientNet-L2 (Tan & Le, 2019), and the Vision Transformer (ViT; Dosovitskiy et al., 2021) (see Appendix B.1 for details on the used models). For ResNet50, DenseNet and ResNeXt101, we include a simple *vanilla* version trained on ImageNet only. For ResNet50 and ResNeXt101, we additionally include a state-of-the-art robust version trained with DeepAugment and Augmix (DAug+AM; Hendrycks et al., 2020a)[2]. For the ResNeXt model, we also include a version that was trained on 3.5 billion weakly labeled images (IG-3.5B; Mahajan et al., 2018). For EfficientNet-L2 we select the current state of the art on IN-C which was trained on 300 million images from JFT-300M (Chollet, 2017; Hinton et al., 2014) using a noisy student-teacher protocol (Xie et al., 2020a). Finally, for the ViT, we use the model pretrained with DINO (Caron et al., 2021). We validate the ImageNet and IN-C performance of all considered models and match the originally reported scores (Schneider et al., 2020). For EfficientNet-L2, we match ImageNet top-1 accuracy up to 0.1% points, and IN-C up to 0.6% points mCE.

**Models for CIFAR10/ MNIST-scale datasets.** For CIFAR10-C experiments, we use three WideResNets (WRN; Zagoruyko & Komodakis, 2016): the first one is trained on clean CIFAR10 and has a depth of 28 and a width of 10, the second one is trained with CIFAR10 with the AugMix protocol (Hendrycks et al., 2020b) and has a depth of 40 and a width of 2, and the third one has a depth of 26 layers, and is pre-trained on clean CIFAR10 using the default training code from `https://github.com/kuangliu/pytorch-cifar`. We used this code-base to also train the ResNet18 and the ResNet50 models on CIFAR10. The remaining small-scale models are trained with UDA methods. We propose to regard any UDA method which requires joint training with source and target data as a pre-training step, similar to regular pre-training on ImageNet, and use self-learning on top of the final checkpoint. We consider two popular UDA methods: self-supervised domain adaptation (UDA-SS; Sun et al., 2019) and Domain-Adversarial Training of Neural Networks (DANN; Ganin et al., 2016). In UDA-SS, the authors seek to align the representations of both domains by performing an auxiliary self-supervised task on both domains simultaneously. In all UDA-SS experiments, we use a WideResNet with a depth of 26 and a width of 16. In DANN, the authors learn a domain-invariant embedding

---

[2]see leaderboard at `github.com/hendrycks/robustness`

Table 2: Self-learning decreases the error on small-scale datasets, for models pre-trained using data augmentation and unsupervised domain adaptation. Entropy minimization outperforms robust pseudo-labeling.

| top1 error [%] on CIFAR10-C ($\searrow$) | number of parameters | w/o adapt | w/ adapt ($\Delta$) RPL | w/ adapt ($\Delta$) ENT |
|---|---|---|---|---|
| WRN-28-10 vanilla (Zagoruyko & Komodakis, 2016) | $3.6 \times 10^7$ | 26.5 | 13.7 (-12.8) | 13.3 (-13.2) |
| WRN-40-2 AM (Hendrycks et al., 2020b) | $2.2 \times 10^6$ | 11.2 | 9.0 (-2.2) | 8.5 (-2.7) |
| WRN-26-1-GN (Bartler et al., 2022) | $1.5 \times 10^6$ | 18.6 | 18.0 (-0.6) | 18.4 (0.2) |
| WRN-26-1-BN (Zagoruyko & Komodakis, 2016) | $1.5 \times 10^6$ | 25.8 | 15.1 (-10.7) | 13.1 (-12.7) |
| WRN-26-16 vanilla (Zagoruyko & Komodakis, 2016) | $9.3 \times 10^7$ | 24.2 | 11.8 (-12.4) | 11.2 (-13.0) |
| WRN-26-16 UDA-SS (Sun et al., 2019) | $9.3 \times 10^7$ | 27.7 | 18.2 (-9.5) | 16.7 (-11.0) |
| WRN-26-16 DANN (Ganin et al., 2016) | $9.3 \times 10^7$ | 29.7 | 28.6 (-1.1) | 28.5 (-1.2) |
| UDA CIFAR10→STL10, top1 error on target [%]($\searrow$) | | | | |
| WRN-26-16 UDA-SS (Sun et al., 2019) | $9.3 \times 10^7$ | 28.7 | 22.9 (-5.8) | 21.8 (-6.9) |
| WRN-26-16 DANN (Ganin et al., 2016) | $9.3 \times 10^7$ | 25.0 | 24.0 (-1.0) | 23.9 (-1.1) |
| UDA MNIST→MNIST-M, top1 error on target [%]($\searrow$) | | | | |
| WRN-26-16 UDA-SS (Sun et al., 2019) | $9.3 \times 10^7$ | 4.8 | 2.4 (-2.4) | 2.0 (-2.8) |
| WRN-26-2 DANN (Ganin et al., 2016) | $1.5 \times 10^6$ | 11.4 | 5.2 (-6.2) | 5.1 (-6.3) |

by optimizing a minimax objective. For all DANN experiments except for MNIST→MNIST-M, we use the same WRN architecture as above. For the MNIST→MNIST-M experiment, the training with the larger model diverged and we used a smaller WideResNet version with a width of 2. We note that DANN training involves optimizing a minimax objective and is generally harder to tune.

## 5 Self-learning universally improves models

Self-learning is a powerful learning scheme, and in the following section we show that it allows to perform test-time adaptation on robustified models, models obtained with large-scale pre-training, as well as (already) domain adapted models across a wide range of datasets and distribution shifts. Our main results on large-scale and small-scale datasets are shown in Tables 1 and 2. These summary tables show final results, and all experiments use the hyperparameters we determined separately on the dev set. The self-learning loss function, i.e. soft- or hard-pseudo-labeling / entropy minimization / robust pseudo-labeling, is a hyperparameter itself, and thus, in Tables 1 and 2, we show the overall best results. Results for the other loss functions can be found in Section 6 and in Appendix C.

**Self-learning successfully adapts ImageNet-scale models across different model architectures on IN-C, IN-A and IN-R (Table 1)**. We adapt the vanilla ResNet50, ResNeXt101 and DenseNet161 models to IN-C and decrease the mCE by over 19 percent points in all models. Further, self-learning works for models irrespective of their size: Self-learning substantially improves the performance of the ResNet50 and the ResNext101 trained with DAug+AM, on IN-C by 11.9 and 9.7 percent points, respectively. Finally, we further improve the current state of the art model on IN-C—the EfficientNet-L2 Noisy Student model—and report a new state-of-the-art result of 22% mCE (which corresponds to a top1 error of 17.1%) on this benchmark with test-time adaptation (compared to 28% mCE without adaptation).

Self-learning is not limited to the distribution shifts in IN-C like compression artefacts or blur. On IN-R, a dataset with renditions, self-learning improves both the vanilla ResNet50 and the EfficientNet-L2 model, the latter of which improves from 23.5% to a new state of the art of 17.4% top-1 error. For a vanilla ResNet50, we improve the top-1 error from 63.8% (Hendrycks et al., 2020a) to 54.1%. On IN-A, adapting the EfficientNet-L2 model using self-learning decreases the top-1 error from 16.5% (Xie et al., 2020a) to 14.8% top-1 error, again constituting a new state of the art with test-time adaptation on this dataset. Self-learning can also be used in an online adaptation setting, where the model continually adapts to new samples on IN-C in Fig. 7(i) or IN-R Fig. 7(ii), Appendix C.10.

Adapting a ResNet50 on IN-A with RPL increases the error from 0% to 0.13% (chance level: 0.1%). Thus, an unadapted ResNet50 has 0% accuracy on IN-A by design and this error "is increased" to chance-level with

self-learning. Since all labels on ImageNet-A are wrong by design, predicting wrong labels as pseudo-labels does not lead to improvements beyond restoring chance-level performance.

The finding that self-learning can be effective across model architectures has also been made by Wang et al. (2021) who show improved adaptation performance on CIFAR100-C for architectures based on self-attention (Zhao et al., 2020) and equilibrium solving (Bai et al., 2020), and also by Mummadi et al. (2021) who showed adaptation results for TENT and TENT+ which combines entropy minimization with a diversity regularizer for a DenseNet121 (Huang et al., 2017), a MobileNetV2 (Sandler et al., 2018), a ResNeXt50 (Xie et al., 2017), and a robust model trained with DAug+AM on IN-C and IN-R.

The improvements of self-learning are very stable: In Table 30, Appendix C.9, we show the averaged results across three different seeds for a ResNet50 model adapted with ENT and RPL. The unbiased std is roughly two orders of magnitude lower than any improvement we report in our paper, showcasing the robustness of our results to random initialization.

**Self-learning improves robustified and domain adapted models on small-scale datasets (Table 2).** We test common domain adaptation techniques like DANN (Ganin et al., 2016) and UDA-SS (Sun et al., 2019), and show that self-learning is effective at further tuning such models to the target domain. We suggest to view unsupervised source/target domain adaptation as a step comparable to pre-training under corruptions, rather than an adaptation technique specifically tuned to the target set—indeed, we can achieve error rates using, e.g., DANN + target adaptation previously only possible with source/target based pseudo-labeling, across different common domain adaptation benchmarks.

For the UDA-SS experiments, we additionally trained a vanilla version with the same architecture using the widely used training code for CIFAR10 at `https://github.com/kuangliu/pytorch-cifar` (1.9k forks, 4.8k stars), and find that the vanilla trained model performs better both with and without adaptation compared to the UDA-SS model. We think that the UDA-SS model would need hyperparameter tuning; we did not perform any tuning for this model, especially because the authors provided scripts with hyperparameters they found to be optimal for different setups. In addition, the clean accuracy of the vanilla model on CIFAR10 (96.5%) is much higher than the average clean accuracy of the UDA-SS model (82.6%), which may explain or imply generally higher robustness under distribution shift (Miller et al., 2021). The finding that self-learning is more effective in the vanilla model compared to the UDA-SS model points towards the hypothesis that the network weights of the vanilla model trained on the source distribution are sufficiently general and can be tuned successfully using only the affine BN statistics, while the weights of the UDA-SS model are already co-adapted to both the source and the target distribution, and thus, self-learning is less effective.

Self-learning also decreases the error on CIFAR10-C of the Wide ResNet model trained with AugMix (AM, Hendrycks et al., 2020b) and reaches a new state of the art on CIFAR10-C of 8.5% top1 error with test-time adaptation.

**Self-learning also improves large pre-trained models (Table 3).** Unlike BatchNorm adaptation (Schneider et al., 2020), we show that self-learning transfers well to models pre-trained on a large amount of unlabeled data: self-learning decreases the mCE on IN-C of the ResNeXt101 trained on 3.5 billion weakly labeled samples (IG-3.5B, Mahajan et al., 2018) from 51.7% to 40.9%.

**Self-learning outperforms other test-time adaptation techniques on IN-C (Tables 4 and 5).** The main point of our paper is showing that self-learning is effective across datasets, model sizes and pretraining methods. Here, we analyze whether our simple techniques can compete with other state-of-the-art adaptation methods. Overall, we find that self-learning outperforms several state-of-the-art techniques, but underperforms in some cases, especially when self-learning is combined with other techniques.

By rigorous and fair model selection, we are able to improve upon TENT (Wang et al., 2021), and find that RPL performs better than entropy minimization on IN-C. We also compare to BN adaptation (Schneider et al., 2020), and find that 1) self-learning further improves the performance upon BN adapt, and 2) self-learning improves performance of a model pretrained on a large amount of data (Mahajan et al., 2018) (Table 3) which is a setting where BN adapt failed.

Table 3: Unlike batch norm adaptation, self-learning adapts large-scale models trained on external data.

| mCE, test [%] ($\searrow$) | w/o adapt | BN adapt | RPL |
|---|---|---|---|
| ResNeXt101 vanilla | 66.6 | 56.8 | 43.2 |
| ResNeXt101 IG-3.5B | 51.7 | 51.8 | **40.9** |

Table 4: Self-learning outperforms other test-time-adaptation techniques on IN-C.

| mCE [%] on IN-C test ($\searrow$) | w/o adapt | BN Adapt | TENT | EATA(lifelong) | RPL | ENT |
|---|---|---|---|---|---|---|
| ResNet50 | 76.7 | 62.2 | 53.5 | 51.2 | **50.5** | 51.6 |

Table 5: Self-learning can further be improved when combining it with other techniques.

| | literature results | | our results | | |
|---|---|---|---|---|---|
| | w/o adapt | w/ adapt | w/o adapt | w/ adapt (RPL) | w/ adapt (ENT) |
| top1 error [%] on IN-C test, sev. 5($\searrow$) | | | | | |
| TTT, ResNet18 (Sun et al., 2020) | 86.6 | 66.3 | 85.4 | **61.9** | 62.8 |
| SLR, ResNet50 (Mummadi et al., 2021) | 82.0 | (46.9) | 82.0 | 54.6 | 54.7 |
| top1 error [%] on CIFAR10-C ($\searrow$) | | | | | |
| MT3, WRN-26-1-GN (Bartler et al., 2022) | 35.7 | 24.4 | 18.6 | 18.4 | **18.0** |
| TTT+++, ResNet50 (Liu et al., 2021) | 29.1 | **9.8** | 24.9 | 14.6 | 12.4 |
| BUFR, ResNet18 (Eastwood et al., 2022) | 42.4 | **10.6** | 25.5 | 13.4 | 12.9 |

Table 6: Vision Transformers can be adapted with self-learning.

| | w/o adapt | w/ adapt affine layers | w/ adapt bottleneck layers | w/ adapt lin. layers | w/ adapt all weights |
|---|---|---|---|---|---|
| mCE on IN-C test [%] ($\searrow$) | | | | | |
| ViT-S/16 | 62.3 | 51.8 | 46.8 | 45.2 | **43.5** |

ENT and RPL outperform the recently published EATA method (Niu et al., 2022). In EATA, high entropy samples are excluded from optimization, and a regularization term is added to prevent the model from forgetting the source distribution. EATA requires tuning of two additional parameters: the threshold for high entropy samples to be discarded and the regularization trade-off parameter $\beta$.

RPL, ENT and simple hard PL outperform TTT (Sun et al., 2020); in particular, note that TTT requires a special loss function at training time, while our approach is agnostic to the pre-training phase. A detailed comparison to TTT is included in Appendix C.6. Mummadi et al. (2021) (SLR) is unpublished work and performs better than ENT and RPL on the highest severity of IN-C. SLR is an extension of entropy minimization where the entropy minimization loss is replaced with a version to ensure non-vanishing gradients of high confidence samples, as well as a diversity regularizer; in addition, a trainable module is prepended to the network to partially undo the distribution shift. Mummadi et al. (2021) introduce the hyperparameters $\delta$ as the trade-off parameter between their two losses, as well as $\kappa$ as the momentum in their diversity regularization term. The success of SLR over ENT and RPL shows the promise of extending self-learning methods by additional objectives, and corroborates our findings on the effectiveness of self-learning.

We also compare our approach to Meta Test-Time Training (MT3, Bartler et al., 2022), which combines meta-learning, self-supervision and test-time training for test-time adaptation. We find that both ENT and RPL perform better than MT3: using the same architecture as Bartler et al. (2022), our best error on CIFAR10-C is 18.0% compared to their best result of 24.4%. When exchanging GroupNorm layers (Wu & He, 2018) for BN layers, the error further reduces to 13.1% (Table 11). We thus find that self-learning is more effective when adapting affine BN parameters instead of GN layers, which is consistent with the findings in Schneider et al. (2020). We included a detailed comparison to Bartler et al. (2022) in Appendix C.7.

TTT+++ (Liu et al., 2021) outperforms both ENT and RPL on CIFAR10-C. Since Liu et al. (2021) do not report results on IN-C, it is impossible to judge whether their gains would generalize, although they do report much better results compared to TENT on Visda-C, so TTT+++ might also be effective on IN-C. Similar to TTT, TTT+++ requires a special loss function during pretraining and thus, cannot be used as an out-of-the-box adaptation technique on top of any pretrained checkpoint.

Table 7: RPL (ENT) performs better on IN-C (CIFAR10-C).

| | mCE, IN-C dev | | | err, C10-C |
| | ResNet50 | ResNeXt-101 | EffNet-L2 | WRN-40 |
|---|---|---|---|---|
| ENT | $50.0 \pm 0.04$ | 43.0 | 22.2 | **8.5** |
| RPL | **48.9** $\pm 0.02$ | **42.0** | **21.3** | 9.0 |

Table 8: RPL performs best without a threshold.

| threshold | 0.0 | 0.5 | 0.9 |
|---|---|---|---|
| mCE on IN-C dev [%] | | | |
| no adapt | 69.5 | | |
| soft PL | 60.1 | | |
| hard PL | 53.8 | 51.9 | 52.4 |
| RPL | **49.7** | 49.9 | 51.8 |

Bottom-Up Feature Restoration (BUFR; Eastwood et al., 2022) outperforms self-learning on CIFAR10-C. The authors note that BUFR is applicable to dataset shifts with measurement shifts which stem from measurement artefacts, but not applicable to more complicated shifts where learning new features would be necessary.

**Self-supervised methods based on self-learning allow out-of-the-box test-time adaptation (Table 6).** The recently published DINO method (Caron et al., 2021) is another variant of self-supervised learning that has proven to be effective for unsupervised representation learning. At the core, the method uses soft pseudo-labeling. Here, we test whether a model trained with DINO on the source dataset can be test-time adapted on IN-C using DINO to further improve out-of-distribution performance. We highlight that we specifically test the self-supervised DINO objective for its practicality as a test-time adaptation method, and did not switch the DINO objective for ENT or RPL to do test-time adaptation. Since the used model is a vision transformer model, we test different choices of adaptation parameters and find considerable performance improvements in all cases, yielding an mCE of 43.5% mCE at a parameter count comparable to a ResNet50 model. For adapting the affine layers, we follow Houlsby et al. (2019).

## 6 Understanding test-time adaptation with self-learning

In the following section, we show ablations and interesting insights of using self-learning for test-time adaptation. If not specified otherwise, all ablations are run on the hold-out corruptions of IN-C (our dev set) with a vanilla ResNet50.

**Robust pseudo-labeling outperforms entropy minimization on large-scale datasets while the reverse is true on small-scale datasets (Table 7).** We find that robust pseudo-labeling consistently improves over entropy minimization on IN-C, while entropy minimization performs better on smaller scale data (CIFAR10, STL10, MNIST). The finding highlights the importance of testing both algorithms on new datasets. The improvement is typically on the order of one percent point.

**Robust pseudo-labeling allows usage of the full dataset without a threshold (Table 8).** Classical hard labeling needs a confidence threshold (T) for best performance, thereby reducing the dataset size, while best performance for RPL is reached for full dataset training with a threshold T of 0.0. We corroborate the results of Wang et al. (2021) who showed that TENT outperforms standard hard labeling with a threshold on the highest severity of CIFAR10-C and CIFAR100-C. We show that this result transfers to IN-C, for a variety of thresholds and pseudo-labeling variants.

**Short update intervals are crucial for fast adaptation (Table 9).** Having established that RPL generally performs better than soft- and hard-labeling, we vary the update interval for the teacher. We find that instant updates are most effective. In entropy minimization, the update interval is instant per default.

**Adaptation of only affine layers is important in CNNs (Table 10).** On IN-C, adapting only the affine parameters after the normalization layers (i.e., the rescaling and shift parameters $\beta$ and $\gamma$) works better on a ResNet50 architecture than adapting all parameters or only the last layer. We indicate the number of adapted parameters in brackets. Note that for Vision Transformers, full model adaptation works better than affine adaptation (see Table 6). We also noticed that on convolutional models with a smaller parameter count like ResNet18, full model adaptation is possible. Wang et al. (2021) also used the affine BN parameters for test-time adaptation with TENT, and report that last layer optimization can improve performance but degrades with further optimization. They suggest that full model optimization does not improve performance

Table 9: RPL performs best with instantaneous updates (ResNet50).

| Update interval (RPL) | w/o adapt | none | epoch | instant |
|---|---|---|---|---|
| mCE, IN-C dev [%] | 69.5 | 54.0 | 49.7 | **49.2** |

Table 10: RPL performs best when affine BN parameters are adapted (ResNet50).

| Mechanism | w/o adapt | last layer | full model | affine |
|---|---|---|---|---|
| mCE, IN-C dev [%] | 69.5 | 60.2 | 51.5 | **48.9** |
| adapted parameters | 0 | 2M | 22.6M | 5.3k |

Table 11: Self-learning works better in models with BN layers compared to models with GN layers, although un-adapted models with GN are more stable under distribution shift compared to models with BN layers.

| top1 error [%] on CIFAR10-C ($\searrow$) | number of parameters | w/o adapt | w/ adapt ($\Delta$) RPL | w/ adapt ($\Delta$) ENT |
|---|---|---|---|---|
| WRN-26-1-BN (Zagoruyko & Komodakis, 2016) | $1.5 \times 10^6$ | 25.8 | 15.1 (-10.7) | 13.1 (-12.7) |
| WRN-26-1-GN (Bartler et al., 2022) | $1.5 \times 10^6$ | 18.6 | 18.4 (-0.2) | 18.0 (-0.6) |
| mCE [%] on IN-C test ($\searrow$) | | | | |
| ResNet50 BigTransfer (Kolesnikov et al., 2020) | $2.6 \times 10^7$ | 55.0 | 54.4 (-0.6) | 56.4 (+1.4) |

Table 12: Our expanded analysis confirms hyperparameter choices from the literature.

| Method | short updates | adapt affine params | use BN instead of GN | adapt to full dataset |
|---|---|---|---|---|
| TENT (Wang et al., 2021) | ✓ | ✓ | ✓ | ✓ |
| EATA (Niu et al., 2022) | ✓ | ✓ | ✓ | ✗ |
| SRL (Mummadi et al., 2021) | ✓ | ✓ | ✓ | ✓ |
| MEMO (Zhang et al., 2021) | ✗ | ✓ | ✓ | ✗ |
| RPL/ENT(ours) | ✓ | ✓ | ✓ | ✓ |

at all. In contrast, we find gains with full model adaptation, but stronger gains with adaptation of only affine parameters.

**Affine BN parameters work better for test-time adaptation compared to GN parameters. (Table 11).** Schneider et al. (2020) showed that models with batch normalization layers are less robust to distribution shift compared to models with group normalization (Wu & He, 2018) layers. However, after adapting BN statistics, the adapted model outperformed the non-adapted GN model. Here, we show that these results also hold for test-time adaptation when adapting a model with GN or BN layers. We show that a WideResNet-26-1 (WRN-26-1) vanilla model with BN layers pretrained on clean CIFAR10 has a much higher error on CIFAR10-C than the same model with GN layers, but it has a much lower error after adaptation. The full results for the WRN-26-1 model can be found in Appendix C.7. Further, we test the pretrained BigTransfer (Kolesnikov et al., 2020) models which have GN layers, and find only small improvements with RPL , and no improvements with ENT. There are no pretrained weights released for the BigTransfer models which have BN layers, thus, a comparison similar to the WRN-26-1 model is not possible. A more detailed discussion on our BigTransfer results as well as a hyperparameter selection study can be found in Appendix E.2.

**Hyperparameters obtained on corruption datasets transfer well to real world datasets.** When evaluating models, we select the hyperparameters discussed above (the learning rate and the epoch used for early stopping are the most critical ones) on the dev set (full results in Appendix C.2). We note that this technique transfers well to IN-R and -A, highlighting the practical value of corruption robustness datasets for adapting models on real distribution shifts.

**Learning rate and number of training epochs are important hyperparameters** We tune the learning rate as well as the number of training epochs for all models, except for the EfficientNet-L2 model where we only train for one epoch due to computational constraints. In Tables 18, 19, 20, and 21 in Appendix C.2, we show that both ENT and RPL collapse after a certain number of epochs, showing the inherent instability of pseudo-labeling. While Wang et al. (2021) trained their model only for one epoch with one learning rate,

Table 13: Self-learning leads to an increased ECE compared to the unadapted model.

| adaptation | IN-C full | IN-C sev 1 | IN-C sev 2 | IN-C sev 3 | IN-C sev 4 | IN-C sev 5 |
|---|---|---|---|---|---|---|
| w/o adapt | $2.3 \pm 0.8$ | $2.3 \pm 0.3$ | $2.1 \pm 0.3$ | $2.1 \pm 0.3$ | $2.2 \pm 0.4$ | $2.9 \pm 1.6$ |
| RPL | $10.6 \pm 7.3$ | $6.6 \pm 0.5$ | $7.9 \pm 1.5$ | $8.9 \pm 2.1$ | $11.2 \pm 3.3$ | $18.7 \pm 12.6$ |
| ENT | $10.6 \pm 7.3$ | $6.6 \pm 0.5$ | $7.9 \pm 1.5$ | $8.9 \pm 2.1$ | $11.2 \pm 3.3$ | $18.7 \pm 12.6$ |

we rigorously perform a full hyperparameter selection on a hold-out set (our dev set), and use the optimal hyperparameters on all tested datasets. We believe that this kind of experimental rigor is essential in order to be able to properly compare methods.

**Our analysis confirms hyperparameter choices from the literature** Having searched over a broad space of hyperparameters and self-learning algorithms, we identified ENT and RPL as the best performing variants across different model sizes, architectures and pretraining techniques. We summarize the most important hyperparameter choices in Table 12 and compare them to those used in the literature when applying self-learning for test-time adaptation. Wang et al. (2021) identified that on CIFAR-C, entropy minimization outperforms hard PL, and found that affine adaptation of BN parameters works better than full model adaptation, and adapted to the full dataset since TENT does not have a threshold in contrast to hard PL. EATA (Niu et al., 2022) and SRL (Mummadi et al., 2021) are extensions of TENT, and thus, followed their hyperparameter choices. EATA does not adapt to the full dataset as they do not adapt to very similar samples or samples with high entropy values. MEMO (Zhang et al., 2021) adapts to a single sample and adapts all model weights. It would be interesting to study whether the performance of MEMO can be improved when adapting only affine BN parameters.

**Pure self-learning hurts calibration, (Table 13)** Eastwood et al. (2022) show that approaches based on entropy minimization and pseudo-labeling hurt calibration due to the objective of confidence maximization. We corroborate their results and report the Expected Calibration Error (ECE; Naeini et al., 2015) when adapting a vanilla ResNet50 model with RPL and ENT. We report the mean ECE (lower is better) and standard deviation across corruptions (and severities) below. We observe that both methods increase ECE compared to the unadapted model. The increase in ECE is higher for more severe corruptions which can be explained by a successively stronger distribution shift compared to the source dataset.

**Self-learning leads to slightly decreased accuracy on the source dataset (Table 14)** To judge the forgetting effect on the source distribution, we calculated the accuracy on clean ImageNet for our adapted ENT and RPL checkpoints. We note that success of self-learning can partially be attributed to correcting the BN statistics of the vanilla model with respect to the distribution shift (Schneider et al., 2020). Thus, we only wish to examine the effect of fine-tuning of the affine BN parameters to the target distribution with respect to the source distribution. Thus, we again correct the mismatched statistics to the source dataset when calculating the accuracy.

We report the mean and standard deviation across corruptions (and severities) below. We observe that both ENT and RPL lead to a decrease in performance on the source dataset, an effect which has also been observed by Niu et al. (2022). The decrease in performance is higher for more severe corruptions which can be explained by a increasingly stronger distribution shift compared to the source dataset. We find that the effect of forgetting is less pronounced in RPL compared to ENT.

Additional experiments and ablation studies, as well as detailed results for all models and datasets can be found in Appendix C. We discuss additional proof-of-concept implementations on the WILDS benchmark (Koh et al., 2021), BigTransfer (BiT; Chen et al., 2020b) models and on self-learning based UDA models in Appendix E. On WILDS, self-learning is effective for the Camelyon17 task with a systematic shift between train, validation and test sets (each set is comprised of different hospitals), while self-learning fails to improve on tasks with mixed domains, such as on the RxRx1 and the FMoW tasks.These results support our claim that self-learning is effective, while showing the important limitation when applied to more diverse shifts.

Table 14: Self-learning leads to slightly decreased accuracy on the source dataset (clean IN).

| model | top1 accuracy on ImageNet val [%] |
|---|---|
| w/o adapt | 74.2 |
| RPL adapted to IN-C (avg over corruptions, sev. 1) | $74.7 \pm 0.6$ |
| RPL adapted to IN-C (avg over corruptions, sev. 2) | $73.9 \pm 0.9$ |
| RPL adapted to IN-C (avg over corruptions, sev. 3) | $73.0 \pm 1.3$ |
| RPL adapted to IN-C (avg over corruptions, sev. 4) | $71.8 \pm 1.8$ |
| RPL adapted to IN-C (avg over corruptions, sev. 5) | $69.8 \pm 3.0$ |
| ENT adapted to IN-C (avg over corruptions, sev. 1) | $74.3 \pm 0.6$ |
| ENT adapted to IN-C (avg over corruptions, sev. 2) | $73.2 \pm 1.1$ |
| ENT adapted to IN-C (avg over corruptions, sev. 3) | $72.3 \pm 1.7$ |
| ENT adapted to IN-C (avg over corruptions, sev. 4) | $70.7 \pm 2.3$ |
| ENT adapted to IN-C (avg over corruptions, sev. 5) | $68.0 \pm 3.9$ |

## 7 A simple model of stability in self-learning

We observed that different self-learning schemes are optimal for small-scale vs. large-scale datasets and varying amount of classes. We reconsider the used loss functions, and unify them into

$$\ell(\mathbf{x}) = -\sum_j \sigma_j \left( \frac{\mathbf{f}^t(\mathbf{x})}{\tau_t} \right) \log \left( \sigma_j \left( \frac{\mathbf{f}^s(\mathbf{x})}{\tau_s} \right) \right),$$

$$\mathbf{f}^t(\mathbf{x}) = \begin{cases} \mathbf{f}(\mathbf{x}), & \text{entropy minimization} \\ \text{sg}(\mathbf{f}(\mathbf{x})), & \text{pseudo-labeling.} \end{cases}$$

(5)

where we introduced student and teacher temperature $\tau_s$ and $\tau_t$ as parameters in the softmax function and the stop gradient operation sg. To study the learning dynamics, we consider a linear student network $\mathbf{f}^s = \mathbf{w}^s \in \mathbb{R}^d$ and a linear teacher network $\mathbf{f}^t = \mathbf{w}^t \in \mathbb{R}^d$ which are trained on $N$ data points $\{\mathbf{x}_i\}_{i=1}^N$ with a binary cross-entropy loss function $\mathcal{L}$ defined as

$$\mathcal{L} = -\sum_{i=1}^N \ell(\mathbf{x}_i) = -\sum_{i=1}^N \left( \sigma_t(\mathbf{x}_i^\top \mathbf{w}^t) \log \sigma_s(\mathbf{x}_i^\top \mathbf{w}^s) + \sigma_t(-\mathbf{x}_i^\top \mathbf{w}^t) \log \sigma_s(-\mathbf{x}_i^\top \mathbf{w}^s) \right),$$

where $\sigma_t(z) = \dfrac{1}{1 + e^{-z/\tau_t}}$ and $\sigma_s(z) = \dfrac{1}{1 + e^{-z/\tau_s}}.$

(6)

With stop gradient, student and teacher evolve in time according to

$$\dot{\mathbf{w}}^s = -\nabla_{\mathbf{w}^s} \mathcal{L}\left(\mathbf{w}^s, \mathbf{w}^t\right), \quad \dot{\mathbf{w}}^t = \alpha(\mathbf{w}^s - \mathbf{w}^t),$$

(7)

where $\alpha$ is the learning rate of the teacher. Without stop gradient, student and teacher are set equal to each other (following the instant updates we found to perform empirically best), and they evolve as

$$\dot{\mathbf{w}} = -\nabla_{\mathbf{w}} \mathcal{L}(\mathbf{w}), \text{ where } \mathbf{w}^s = \mathbf{w}^t = \mathbf{w}.$$

(8)

We restrict the theoretical analysis to the time evolution of the components of $\mathbf{w}^{s,t}$ in direction of two data points $\mathbf{x}_k$ and $\mathbf{x}_l$, $y_k^{s,t} \equiv \mathbf{x}_k^\top \mathbf{w}^{s,t}$ and $y_l^{s,t} \equiv \mathbf{x}_l^\top \mathbf{w}^{s,t}$. All other components $y_i^{s,t}$ with $i \neq k, l$ are neglected to reduce the dimensionality of the equation system. It turns out that the resulting model captures the neural network dynamics quite well despite the drastic simplification of taking only two data points into account (see Figure 2 for a comparison of the model vs. self-learning on the CIFAR-C dataset). We obtain the dynamics:

$$\text{with stop gradient: } \dot{y}_k^s = -\mathbf{x}_k^\top \nabla_{\mathbf{w}^s} \left( \ell(\mathbf{x}_k) + \ell(\mathbf{x}_l) \right), \quad \dot{y}_l^s = -\mathbf{x}_l^\top \nabla_{\mathbf{w}^s} \left( \ell(\mathbf{x}_k) + \ell(\mathbf{x}_l) \right),$$

$$\dot{y}_k^t = \alpha(y_k^t - y_k^s), \quad \dot{y}_l^t = \alpha(y_l^t - y_l^s),$$

(9)

$$\text{without stop gradient: } \dot{y}_k = -\mathbf{x}_k^\top \nabla_{\mathbf{w}} \left( \ell(\mathbf{x}_k) + \ell(\mathbf{x}_l) \right), \quad \dot{y}_l = -\mathbf{x}_l^\top \nabla_{\mathbf{w}} \left( \ell(\mathbf{x}_k) + \ell(\mathbf{x}_l) \right).$$

With this setup in place, we can derive

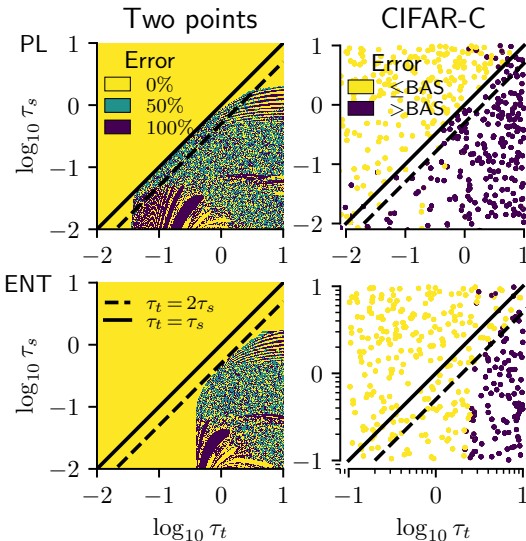

Figure 2: For the two point model, we show accuracy, and for the CIFAR10-C simulation, we show improvement (yellow) vs. degradation (purple) over the non-adapted baseline (BAS). An important convergence criterion for pseudo-labeling (top row) and entropy minimization (bottom row) is the ratio of student and teacher temperatures; it lies at $\tau_s = \tau_t$ for PL, and $2\tau_s = \tau_t$ for ENT. Despite the simplicity of the two-point model, the general convergence regions transfer to CIFAR10-C.

**Proposition 7.1** (Collapse in the two-point model). *The student and teacher networks $\mathbf{w}_s$ and $\mathbf{w}_t$ trained with stop gradient do not collapse to the trivial representation $\forall \mathbf{x} : \mathbf{x}^\top \mathbf{w}^s = 0, \mathbf{x}^\top \mathbf{w}^t = 0$ if $\tau_s > \tau_t$. The network $\mathbf{w}$ trained without stop gradient does not collapse if $\tau_s > \tau_t/2$.   Proof. see § A.1.* □

We validate the proposition on a simulated two datapoint toy dataset, as well as on the CIFAR-C dataset (Figure 2). In general, the size and location of the region where collapse is observed in the simulated model also depends on the initial conditions, the learning rate and the optimization procedure. An in depth discussion, as well as additional simulations are given in Appendix A.

Entropy minimization with standard temperatures ($\tau_s = \tau_t = 1$) and hard pseudo-labeling ($\tau_t \to 0$) are hence stable. The two-point learning dynamics vanish for soft pseudo-labeling with $\tau_s = \tau_t$, suggesting that one would have to analyze a more complex model with more data points. While this does not directly imply that the learning is unstable at this point, we empirically observe that both entropy minimization and hard labeling outperform soft-labeling in practice.

The finding aligns with empirical work: For instance, Caron et al. (2021) fixed $\tau_s$ and varied $\tau_t$ during training, and empirically found an upper bound for $\tau_t$ above which the training was no longer stable. It also aligns with our findings suggesting that hard-labeling tends to outperform soft-labeling approaches, and soft-labeling performs best when selecting lower teacher temperatures.

In practice, the result suggests that *student temperatures should always exceed the teacher temperatures for pseudo-labeling*, and *student temperatures should always exceed half the teacher temperature for entropy minimization*, which narrows the search space for hyperparameter optimization considerably.

## 8    Adapting models on a wider range of distribution shifts reveals limitations of robustification and adaptation methods

Robustness datasets on ImageNet-scale have so far been limited to a few selected domains (image corruptions in IN-C, image renditions in IN-R, difficult images for ResNet50 classifiers in IN-A). In order to test our approach on a wider range of complex distribution shifts, we re-purpose the dataset from the Visual Domain Adaptation Challenge 2019 (DomainNet; Saenko et al., 2019) as an additional robustness benchmark.

Table 15: Self-learning decreases the top1 error on IN-D domains with strong initial performance, but fails to improve performance on challenging domains.

| domain
adapt
model | Real | | Painting | | Clipart | | Sketch | | Infograph | | Quickdraw | | ImageNet |
|---|---|---|---|---|---|---|---|---|---|---|---|---|---|
| | w/o | w/ | w/o | w/ | w/o | w/ | w/o | w/ | w/o | w/ | w/o | w/ | w/o |
| EffNet-L2 Noisy Student | 29.2 | **27.9** | 42.7 | **40.9** | 45.0 | **37.9** | 56.4 | **51.5** | **77.9** | 94.3 | **98.4** | 99.4 | 11.6 |
| ResNet50 DAug+AM | 39.2 | 36.5 | 58.7 | 53.4 | 68.4 | 57.0 | 75.2 | 61.3 | 88.1 | 83.2 | 98.2 | 99.1 | 23.3 |
| ResNet50 vanilla | 40.1 | 37.3 | 65.1 | 57.8 | 76.0 | 63.6 | 82.0 | 73.0 | 89.6 | 85.1 | 99.2 | 99.8 | 23.9 |

**Creation of ImageNet-D** The original DomainNet dataset comes with six image styles: "Clipart", "Real", "Infograph", "Painting", "Quickdraw" and "Sketch", and has 345 classes in total, out of which 164 overlap with ImageNet. We map these 164 DomainNet classes to 463 ImageNet classes, e.g., for an image from the "bird" class in DomainNet, we accept all 39 bird classes in ImageNet as valid predictions. ImageNet also has ambiguous classes, e.g., it has separate classes for "cellular telephone" and "dial phone". For these cases, we accept both predictions as valid. In this sense, the mapping from DomainNet to ImageNet is a one-to-many mapping. We refer to the smaller version of DomainNet that is now compatible with ImageNet-trained models as ImageNet-D (IN-D). The benefit of IN-D over DomainNet is this re-mapping to ImageNet classes which allows robustness researchers to easily benchmark on this dataset, without the need of re-training a model (as is common in UDA). We show example images from IN-D in Table 15. The detailed evaluation protocol on IN-D, our label-mapping procedure from DomainNet to ImageNet along with justifications for our design choices and additional analysis are outlined in Appendix D.

The most similar robustness dataset to IN-D is IN-R which contains renditions of ImageNet classes, such as art, cartoons, deviantart, graffiti, embroidery, graphics and others. The benefit of IN-D over IN-R is that in IN-D, the images are separated according to the domain allowing for studying of systematic domain shifts, while in IN-R, the different domains are not distinguished. ImageNet-Sketch (Wang et al., 2019) is a dataset similar to the "Sketch" domain of IN-D.

**More robust models perform better on IN-D.** To test whether self-learning is helpful for more complex distribution shifts, we adapt a vanilla ResNet50, several robust IN-C models and the EfficientNet-L2 Noisy Student model on IN-D. We use the same hyperparameters we obtained on IN-C dev for all our IN-D experiments[3]. We show our main results in Table 15. Comparing the performance of the vanilla ResNet50 model to its robust DAug+AM variant, we find that the DAug+AM model performs better on all domains, with the most significant gains on the "Clipart", "Painting" and "Sketch" domains. We show detailed results for all domains and all tested models in Appendix D.3, along with results on IN-C and IN-R for comparison. We find that the best performing models on IN-D are also the strongest ones on IN-C and IN-R which indicates good generalization capabilities of the techniques combined for these models, given the large differences between the three considered datasets. The Spearman's rank correlation coefficient between IN-C and IN-D (averaged over all domains) is 0.54, and 0.73 between IN-R and IN-D. Thus, the errors on IN-R are strongly correlated to errors on IN-D which can be explained by the similarity of IN-D and IN-R. We show Spearman's rank correlation cofficients for the individual domains versus IN-C/IN-R in Fig. 9 in Appendix D.5, and find correlation values above 0.8 between IN-R and IN-D for all domains except for the "Real" domain where the coefficient is almost zero. Further, we find that even the best models perform 20 to 30 percentage points worse on IN-D compared to their performance on IN-C or IN-R, indicating that IN-D might be a more challenging benchmark.

**All models struggle with some domains of IN-D.** The EfficientNet-L2 Noisy Student model obtains the best results on most domains. However, we note that the overall error rates are surprisingly high compared to the model's strong performance on the other considered datasets (IN-A: 14.8% top-1 error, IN-R: 17.4% top-1

---

[3]In regards to hyperparameter selection, we performed a control experiment where we selected hyperparameters with leave-one-out cross validation—this selection scheme actually performed worse than IN-C parameter selection (see Appendix D.2).

error, IN-C: 22.0% mCE). Even on the "Real" domain closest to clean ImageNet where the EfficientNet-L2 model has a top-1 error of 11.6%, the model only reaches a top-1 error of 29.2%. Self-learning decreases the top-1 error on all domains except for "Infograph" and "Quickdraw". We note that both domains have very high error rates from the beginning and thus hypothesize that the produced pseudo-labels are of low quality.

**Error analysis on IN-D.** We investigate the errors a ResNet50 model makes on IN-D by analyzing the most frequently predicted classes for different domains to reveal systematic errors indicative of the encountered distribution shifts. We find most errors interpretable: the classifier assigns the label "comic book" to images from the "Clipart" or "Painting" domains, "website" to images from the "Infograph" domain, and "envelope" to images from the "Sketch" domain. Thus, the classifier predicts the domain rather than the class. We find no systematic errors on the "Real" domain which is expected since this domain should be similar to ImageNet. Detailed results on the most frequently predicted classes for different domains can be found in Fig. 9, Appendix D.5.

**IN-D should be used as an additional robustness benchmark.** While the error rates on IN-C, -R and -A are at a well-acceptable level for our largest EfficientNet-L2 model after adaptation, IN-D performance is consistently worse for all models. We propose to move from isolated benchmark settings like IN-R (single domain) to benchmarks more common in domain adaptation (like DomainNet) and make IN-D publicly available as an easy to use dataset for this purpose.

## 9    Best practices and evaluation in test-time adaptation

Based on our results as well as our discussion on previous work, we arrive at several proposals on how test-time adaptation should be evaluated in future work to ensure scientific rigor:

1. **Cross-validation:** We propose using the hold-out set of IN-C for model selection of all relevant hyperparameters, and then using these hyperparameters for testing on different datasets.

2. **Comparison to simple baselines:** With proper hyperparameter tuning, very simple baselines can perform on par with sophisticated approaches. This insight is also discussed by Gulrajani & Lopez-Paz (2021) for the setting of domain generalization and by Rusak et al. (2020) for robustness to common corruptions.

3. **Using more robust models:** Test-time adaptation can further improve upon robust models, which were pre-trained with more data or with UDA, or using protocols to increase robustness. A test-time adaptation method will be much more relevant to practitioners if it can improve upon the most robust model they can find for their task.

4. **Important hyperparameters:** We identify several important hyperparameters which affect the final performance in a crucial way, and thus, should be tuned to ensure fair comparisons:
   - **Number of adaptation epochs and learning rate**: The final performance crucially depends on both of these parameters for all models and all methods that we have studied.
   - **Adaptation parameters** While adaptation of affine batch normalization parameters works best for adaptation of CNNs, full adaptation performs best for ViTs. Therefore, it is important to benchmark test-time adaptation for different model architectures and adaptation parameters.

## 10    Conclusion

We evaluated and analysed how self-learning, an essential component in many unsupervised domain adaptation and self-supervised pre-training techniques, can be applied for adaptation to both small and large-scale image recognition problems common in robustness research. We demonstrated new state-of-the-art adaptation results with the EfficientNet-L2 model on the benchmarks ImageNet-C, -R, and -A, and introduced a new benchmark dataset (ImageNet-D) which remains challenging even after adaptation. Our theoretical analysis shows the influence of the temperature parameter in the self-learning loss function on the training stability and provides guidelines how to choose a suitable value. Based on our extensive experiments, we formulate best practices for future research on test-time adaptation. Across the large diversity of (systematic) distribution

shifts, architectures and pre-training methods we tested in this paper, we found that self-learning almost universally improved test-time performance. An important limitation of current self-learning methods is the observed instability over longer adaptation time frames. While we mitigate this issue through model selection (and show its robustness across synthetic and natural distribution shifts), this might not universally hold across distribution shifts encountered in practice. Concurrent work, e.g. Niu et al. (2022) tackle this problem through modifications of self-learning algorithms, and we think this direction will be important to continue to explore in future work. That being said, we hope that our results encourage both researchers and practitioners to *experiment with self-learning if their data distribution shifts.*

**Reproducibility Statement**    We attempted to make our work as reproducible as possible: We mostly used pre-trained models which are publicly available and we denoted the URL addresses of all used checkpoints; for the checkpoints that were necessary to retrain, we report the Github directories with the source code and used an official or verified reference implementation when available. We report all used hyperparameters in the Appendix and released the code.

**Software and Data**    Code for reproducing results of this paper is available at `https://github.com/bethgelab/robustness`.

**Author Contributions**    StS and ER conceived the project with suggestions from MB and WB. ER ran initial experiments. StS performed large scale experiments on ImageNet-C,R,A with support from PG, ER and WB. ER performed domain adaptation and CIFAR experiments with suggestions from StS. ER implemented all baselines that have been included as comparisons in the manuscript. GP ran DINO adaptation, WILDS, and self-ensembling experiments with suggestions from ER & StS. ER developed the ImageNet-D dataset with suggestions from StS and WB, and performed the large scale experiments on ImageNet-D. LE and StS developed the theoretical model, ER contributed the CIFAR-C experiments. WB, ER and StS wrote the initial manuscript with help from all authors. ER wrote an extensive related work section. ER reviewed and edited the manuscript in the course of the author-reviewer discussion period.

**Acknowledgements**    We thank Roland S. Zimmermann, Yi Zhu, Raghavan Manmatha, Matthias Kümmerer, Matthias Tangemann, Bernhard Schölkopf, Justin Gilmer, Shubham Krishna, Julian Bitterwolf, Berkay Kicanaoglu and Mohammadreza Zolfaghari for helpful discussions on pseudo labeling and feedback on our paper draft. We thank Yasser Jadidi and Alex Smola for support in the setup of our compute infrastructure. We thank the anonymous reviewers from TMLR and our Action Editor for their constructive feedback.

We thank the International Max Planck Research School for Intelligent Systems (IMPRS-IS) for supporting E.R. and St.S.; St.S. acknowledges his membership in the European Laboratory for Learning and Intelligent Systems (ELLIS) PhD program. This work was supported by the German Federal Ministry of Education and Research (BMBF) through the Tübingen AI Center (FKZ: 01IS18039A), by the Deutsche Forschungsgemeinschaft (DFG) in the priority program 1835 under grant BR2321/5-2 and in the SFB 1233, Robust Vision: Inference Principles and Neural Mechanisms (TP3), project number: 276693517. WB acknowledges financial support via an Emmy Noether Grant funded by the German Research Foundation (DFG) under grant no. BR 6382/1-1 and via the Open Philantropy Foundation funded by the Good Ventures Foundation. WB is a member of the Machine Learning Cluster of Excellence, EXC number 2064/1 – Project number 390727645.

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

# A  A two-point model of self-learning

## A.1  Proof of Proposition 7.1

**Learning dynamics with stop gradient.**  Computing the stop gradient evolution defined in equation 7 explicitly yields

$$
\dot{\mathbf{w}}^s = -\nabla_{\mathbf{w}^s}\mathcal{L} = \frac{1}{\tau_s}\sum_{i=1}^N \left(\sigma_t(\mathbf{x}_i^\top\mathbf{w}^t)\sigma_s(-\mathbf{x}_i^\top\mathbf{w}^s) - \sigma_t(-\mathbf{x}_i^\top\mathbf{w}^t)\sigma_s(\mathbf{x}_i^\top\mathbf{w}^s)\right)\mathbf{x}_i
$$
$$
\dot{\mathbf{w}}^t = \alpha(\mathbf{w}^s - \mathbf{w}^t)
$$
(10)

The second equality uses the well-known derivative of the sigmoid function, $\partial_z\sigma(z) = \sigma(z)\sigma(-z)$.

The equation system of $2d$ nonlinear, coupled ODEs for $\mathbf{w}^s \in \mathbb{R}^d$ and $\mathbf{w}^t \in \mathbb{R}^d$ in equation 10 is analytically difficult to analyze. Instead of studying the ODEs directly, we act on them with the data points $\mathbf{x}_k^\top$, $k = 1, \ldots, N$, and investigate the dynamics of the components $\mathbf{x}_k^\top\mathbf{w}^{s,t} \equiv y_k^{s,t}$:

$$
\dot{y}_k^s = \frac{1}{\tau_s}\sum_{i=1}^N \left(\mathbf{x}_i^\top\mathbf{x}_k\right)\left(\sigma_t(y_i^t)\sigma_s(-y_i^s) - \sigma_t(-y_i^t)\sigma_s(y_i^s)\right)
$$
$$
\dot{y}_k^t = \alpha(y_k^s - y_k^t).
$$
(11)

The learning rate of each mode $y_k^s$ is scaled by $(\mathbf{x}_k^\top\mathbf{x}_i)$ which is much larger for $i = k$ than for $i \neq k$ in high-dimensional spaces. In the two-point approximation, we consider only the two (in absolute value) largest terms $i = k, l$ for a given $k$ in the sum in equation 11. Any changes that $y_k^{s,t}(t)$ and $y_l^{s,t}(t)$ might induce in other modes $y_i^{s,t}(t)$ are neglected, and so we are left with only four ODEs:

$$
\begin{aligned}
\dot{y}_k^s =& \frac{1}{\tau_s}\|\mathbf{x}_k\|^2\left(\sigma_t(y_k^t)\sigma_s(-y_k^s) - \sigma_t(-y_k^t)\sigma_s(y_k^s)\right) \\
&+ \frac{1}{\tau_s}(\mathbf{x}_k^\top\mathbf{x}_l)\left(\sigma_t(y_l^t)\sigma_s(-y_l^s) - \sigma_t(-y_l^t)\sigma_s(y_l^s)\right), \\
\dot{y}_l^s =& \frac{1}{\tau_s}\|\mathbf{x}_l\|^2\left(\sigma_t(y_l^t)\sigma_s(-y_l^s) - \sigma_t(-y_l^t)\sigma_s(y_l^s)\right) \\
&+ \frac{1}{\tau_s}(\mathbf{x}_k^\top\mathbf{x}_l)\left(\sigma_t(y_k^t)\sigma_s(-y_k^s) - \sigma_t(-y_k^t)\sigma_s(y_k^s)\right) \\
\dot{y}_k^t =& \alpha(y_k^s - y_k^t), \ \dot{y}_l^t = \alpha(y_l^s - y_l^t).
\end{aligned}
$$
(12)

The fixed points of equation 12 satisfy

$$
\dot{y}_k^s = \dot{y}_l^s = \dot{y}_k^t = \dot{y}_l^t = 0.
$$
(13)

For $\alpha > 0$, requiring $\dot{y}_k^t = \dot{y}_l^t = 0$ implies that $y_k^s = y_k^t$ and $y_l^s = y_l^t$. For $\tau_s = \tau_t$, the two remaining equations $\dot{y}_k^s = \dot{y}_l^s = 0$ vanish automatically so that there are no non-trivial two-point learning dynamics. For $\tau_s \neq \tau_t$, there is a fixed point at $y_k^{s,t} = y_l^{s,t} = 0$ since at this point, each bracket in equation 12 vanishes individually:

$$
\sigma_t(y_{k,l})\sigma_s(-y_{k,l}) - \sigma_s(-y_{k,l})\sigma_t(y_{k,l})\Big|_{y_{k,l}=0} = \frac{1}{4} - \frac{1}{4} = 0.
$$
(14)

At the fixed point $y_k^{s,t} = y_l^{s,t} = 0$, $\mathbf{w}^s$ and $\mathbf{w}^t$ are orthogonal to both $\mathbf{x}_k$ and $\mathbf{x}_l$ and hence classification fails. If this fixed point is stable, $\mathbf{w}^s$ and $\mathbf{w}^t$ will stay at the fixed point once they have reached it, i.e. the model collapses. The fixed point is stable when all eigenvalues of the Jacobian $J$ of the ODE system equation 12 evaluated at $y_k^{s,t} = y_l^{s,t} = 0$ are negative. Two eigenvalues $\lambda_\pm$ are always negative, whereas the two other

eigenvalues $\tilde{\lambda}_\pm$ are positive if $\tau_s > \tau_t$:

$$
J\Big|_{y_k^{s,t}=y_l^{s,t}=0} = \begin{pmatrix} \frac{-\|\mathbf{x}_k\|^2}{4\tau_s^2} & \frac{-(\mathbf{x}_k^\top \mathbf{x}_l)}{4\tau_s^2} & \frac{\|\mathbf{x}_k\|^2}{4\tau_s\tau_t} & \frac{(\mathbf{x}_k^\top \mathbf{x}_l)}{4\tau_s\tau_t} \\ \frac{-(\mathbf{x}_k^\top \mathbf{x}_l)}{4\tau_s^2} & \frac{-\|\mathbf{x}_l\|^2}{4\tau_s^2} & \frac{(\mathbf{x}_k^\top \mathbf{x}_l)}{4\tau_s\tau_t} & \frac{\|\mathbf{x}_l\|^2}{4\tau_s\tau_t} \\ \alpha & 0 & -\alpha & 0 \\ 0 & \alpha & 0 & -\alpha \end{pmatrix},
$$

$$
\lambda_\pm = -\frac{1}{16\tau_s^2}\left(A_\pm + \sqrt{A_\pm^2 + 32\alpha\tau_s^2 B(\tau_s/\tau_t - 1)}\right) < 0, \tag{15}
$$

$$
\tilde{\lambda}_\pm = \frac{1}{16\tau_s^2}\left(-A_\pm + \sqrt{A_\pm^2 + 32\alpha\tau_s^2 B(\tau_s/\tau_t - 1)}\right) > 0 \text{ if } \tau_s > \tau_t, \text{ where}
$$

$$
A_\pm = 8\alpha\tau_s^2 \pm B, \quad B = \|\mathbf{x}_k\| + \|\mathbf{x}_l\| + \sqrt{(\|\mathbf{x}_k\| - \|\mathbf{x}_l\|)^2 + 4(\mathbf{x}_k^\top \mathbf{x}_l)^2}
$$

To sum up, training with stop gradient and $\tau_s > \tau_t$ avoids a collapse of the two-point model to the trivial representation $y_k^{s,t} = y_l^{s,t} = 0$ since the fixed point is not stable in this parameter regime.

**Learning dynamics without stop gradient** Without stop gradient, we set $\mathbf{w}^t = \mathbf{w}^s \equiv \mathbf{w}$ which leads to an additional term in the gradient:

$$
\dot{\mathbf{w}} = -\nabla_\mathbf{w}\mathcal{L} = \frac{1}{\tau_s}\sum_{i=1}^N \left(\sigma_t(\mathbf{x}_i^\top \mathbf{w})\sigma_s(-\mathbf{x}_i^\top \mathbf{w}) - \sigma_t(-\mathbf{x}_i^\top \mathbf{w})\sigma_s(\mathbf{x}_i^\top \mathbf{w})\right)\mathbf{x}_i
$$

$$
+ \frac{1}{\tau_t}\sum_{i=1}^N \sigma_t(\mathbf{x}_i^\top \mathbf{w})\sigma_t(-\mathbf{x}_i^\top \mathbf{w})\underbrace{\left(\log\sigma_s(\mathbf{x}_i^\top \mathbf{w}) - \log\sigma_s(-\mathbf{x}_i^\top \mathbf{w})\right)}_{=\log\left((1+e^{y_i/\tau_s})/(1+e^{-y_i/\tau_s})\right)=y_i/\tau_s}\mathbf{x}_i. \tag{16}
$$

As before, we focus on the evolution of the two components $y_k = \mathbf{w}^\top \mathbf{x}_k$ and $y_l = \mathbf{w}^\top \mathbf{x}_l$.

$$
\dot{y}_k = \|\mathbf{x}_k\|^2\left(\frac{1}{\tau_s}\left(\sigma_t(y_k)\sigma_s(-y_k) - \sigma_t(-y_k)\sigma_s(y_k)\right) + \frac{1}{\tau_t}\sigma_t(y_k)\sigma_t(-y_k)y_k\right)
$$

$$
+ (\mathbf{x}_k^\top \mathbf{x}_l)\left(\frac{1}{\tau_s}\left(\sigma_t(y_l)\sigma_s(-y_l) - \sigma_t(-y_l)\sigma_s(y_l)\right) + \frac{1}{\tau_s\tau_t}\sigma_t(y_l)\sigma_t(-y_l)y_l\right)
$$

$$
\dot{y}_l = \|\mathbf{x}_l\|^2\left(\frac{1}{\tau_s}\left(\sigma_t(y_l)\sigma_s(-y_l) - \sigma_t(-y_l)\sigma_s(y_l)\right) + \frac{1}{\tau_t}\sigma_t(y_l)\sigma_t(-y_l)y_l\right) \tag{17}
$$

$$
+ (\mathbf{x}_k^\top \mathbf{x}_l)\left(\frac{1}{\tau_s}\left(\sigma_t(y_k)\sigma_s(-y_k) - \sigma_t(-y_k)\sigma_s(y_k)\right) + \frac{1}{\tau_s\tau_t}\sigma_t(y_k)\sigma_t(-y_k)y_k\right)
$$

There is a fixed point at $y_k = y_l = 0$ where each bracket in equation 17 vanishes individually,

$$
\frac{1}{\tau_s}\left(\sigma_t(y_{k,l})\sigma_s(-y_{k,l}) - \sigma_t(-y_{k,l})\sigma_s(y_{k,l})\right) + \frac{1}{\tau_s\tau_t}\sigma_t(y_{k,l})\sigma_t(-y_{k,l})y_{k,l}\Big|_{y_{k,l}} = 0. \tag{18}
$$

The Jacobian of the ODE system in equation 17 and its eigenvalues evaluated at the fixed point are given by

$$
J\Big|_{y_k=y_l=0} = \begin{pmatrix} \frac{\|\mathbf{x}_k\|^2}{4\tau_s}\left(\frac{2}{\tau_t} - \frac{1}{\tau_s}\right) & \frac{(\mathbf{x}_k^\top \mathbf{x}_l)}{4\tau_s}\left(\frac{2}{\tau_t} - \frac{1}{\tau_s}\right) \\ \frac{(\mathbf{x}_k^\top \mathbf{x}_l)}{4\tau_s}\left(\frac{2}{\tau_t} - \frac{1}{\tau_s}\right) & \frac{\|\mathbf{x}_l\|^2}{4\tau_s}\left(\frac{2}{\tau_t} - \frac{1}{\tau_s}\right) \end{pmatrix}
$$

$$
\lambda_{1,2} = \frac{1}{8\tau_s}\left(\frac{2}{\tau_t} - \frac{1}{\tau_s}\right)\left(\pm\underbrace{\sqrt{\|\mathbf{x}_k\|^4 + \|\mathbf{x}_l\|^4 - 2\|\mathbf{x}_k\|^2\|\mathbf{x}_l\|^2 + 4(\mathbf{x}_k^\top \mathbf{x}_l)^2}}_{\leq \|\mathbf{x}_k\|^2 + \|\mathbf{x}_l\|^2} + \underbrace{\|\mathbf{x}_k\|^2 + \|\mathbf{x}_l\|^2}_{}\right). \tag{19}
$$

$$\underbrace{\phantom{xxxxxxxxxxxxxxxxxxxxxxxxxxxxxxxxxxxxxxxxxxxxxxxxxxxxxxxxxxx}}_{\geq 0 \text{ with equality if } \mathbf{x}_k = \pm\mathbf{x}_l}$$

Hence the fixed point is unstable when $\tau_s > \tau_t/2$ and thus the model without stop gradient does not collapse onto $y_k = y_l = 0$ in this regime, concluding the proof.

## A.2 Simulation of the two-point model

For visualization purposes in the main paper, we set $\mathbf{w}^s = \mathbf{w}^t = [0.5, 0.5]^\top$ and train the model using instant gradient updates on the dataset with points $\mathbf{x}_1 = [1, 0]$ and $\mathbf{x}_2 = [0, -1]$ using SGD with learning rate 0.1 and momentum 0.9. We varied student and teacher temperatures on a log-scale with 250 points from $10^{-3}$ to 10. Qualitatively similar results can be obtained without momentum training, at higher learning rates (most likely due to the implicit learning rate scaling introduced by the momentum term).

Note that the temperature scales for observing the collapse effect depend on the learning rate, and the exact training strategy—lower learning rates can empirically prevent the model from collapsing and shift the convergence region. The result in Figure 2 will hence depend on the exact choice of learning rate (which is currently not considered in our continuous time evolution theory), while the predicted region without collapse is robust to details of the optimization.

To visualize the impact of different hyperparameters, we show variants of the two point model with different learning rates using gradient descent with (Figure 3) and without momentum (Figure 4), and with different start conditions (Figure 5), which all influence the regions where the model degrades, but not the stable regions predicted by our theory.

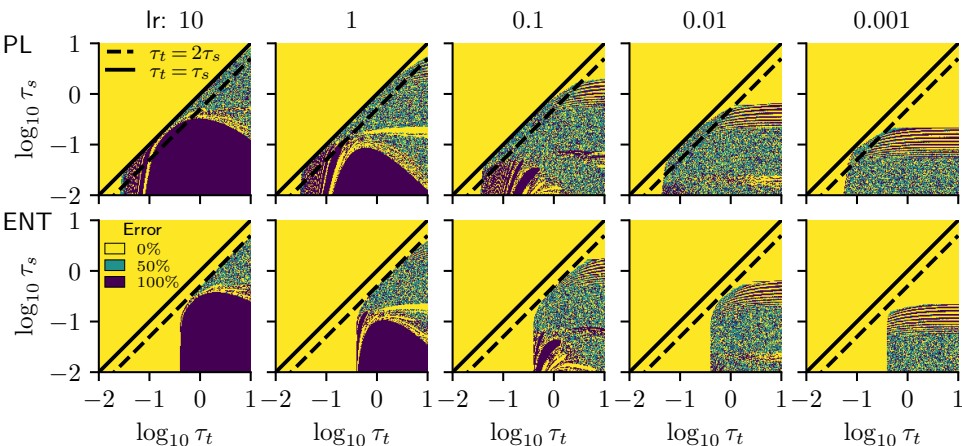

Figure 3: Entropy minimization (top) Training two point model with momentum 0.9 and different learning rates with initialization $\mathbf{w}^s = \mathbf{w}^t = [0.5, 0.5]^\top$.

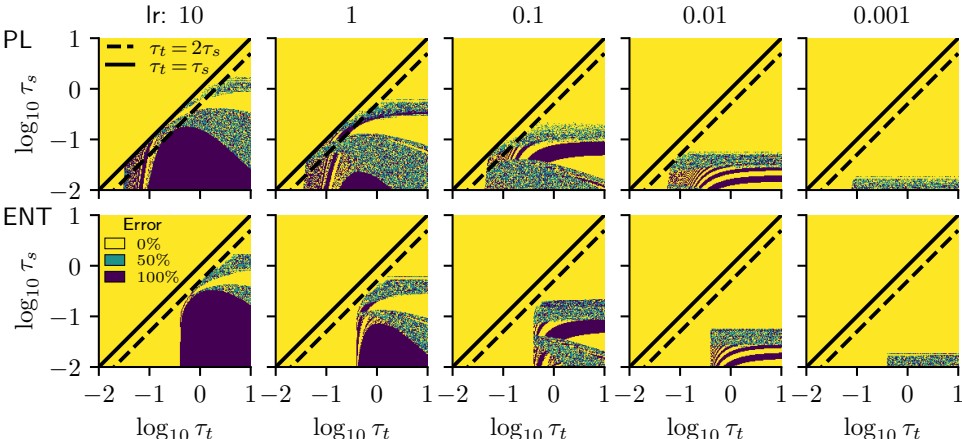

Figure 4: Training a two point model without momentum and different learning rates with initialization $\mathbf{w}^s = \mathbf{w}^t = [0.5, 0.5]^\top$. Note that especially for lower learning rates, longer training would increase the size of the collapsed region.

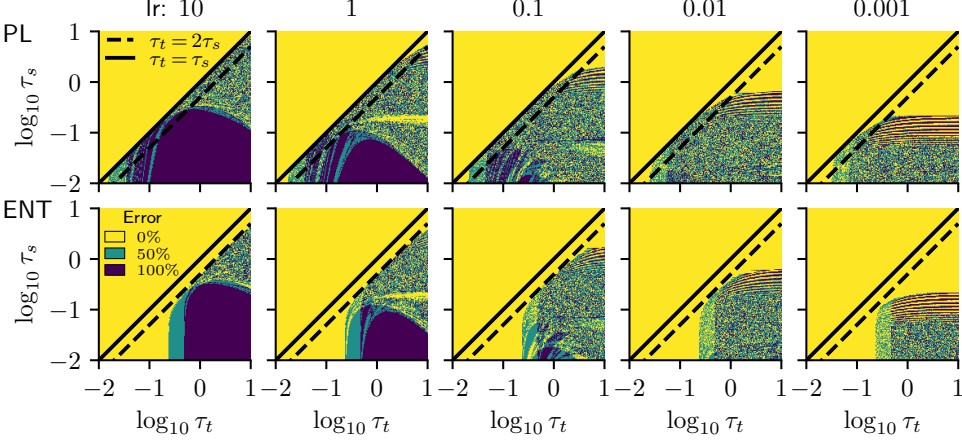

Figure 5: Training a two point model with momentum 0.9 and different learning rates with initialization $\mathbf{w}^s = \mathbf{w}^t = [0.6, 0.3]^\top$.

## B  Additional information on used models

### B.1  Details on all hyperparameters we tested for different models

For all models except EfficientNet-L2, we adapt the batch norm statistics to the test domains following (Schneider et al., 2020). We do not expect significant gains for combining EfficientNet-L2 with batch norm adaptation: as demonstrated in (Schneider et al., 2020), models trained with large amounts of weakly labeled data do not seem to benefit from batch norm adaptation.

**ResNet50 models (IN-C)**  We use a vanilla ResNet50 model and compare soft- and hard-labeling against entropy minimization and robust pseudo-labeling. To find optimal hyperparameters for all methods, we perform an extensive evaluation and test (i) three different adaptation mechanisms (ii) several learning rates $1.0 \times 10^{-4}$, $1.0 \times 10^{-3}$, $1.0 \times 10^{-2}$ and $5.0 \times 10^{-2}$, (iii) the number of training epochs and (iv) updating the teacher after each epoch or each iteration. For all experiments, we use a batch size of 128. The hyperparameter search is performed on IN-C dev. We then use the optimal hyperparameters to evaluate the methods on the IN-C test set.

**ResNeXt101 models**  The ResNeXt101 model is considerably larger than the ResNet50 model and we therefore limit the number of ablation studies we perform for this architecture. Besides a baseline, we include a state-of-the-art robust version trained with DeepAugment+Augmix (DAug+AM, Hendrycks et al., 2020a) and a version that was trained on 3.5 billion weakly labeled images (IG-3.5B, Mahajan et al., 2018). We only test the two leading methods on the ResNeXt101 models (ENT and RPL). We vary the learning rate in same interval as for the ResNet50 model but scale it down linearly to account for the smaller batch size of 32. We only train the affine batch normalization parameters because adapting only these parameters leads to the best results on ResNet50 and is much more resource efficient than adapting all model parameters. Again, the hyperparameter search is performed only on the development corruptions of IN-C. We then use the optimal hyperparameters to evaluate the methods on the IN-C test set.

**EfficientNet-L2 models**  The current state of the art on IN, IN-C, IN-R and IN-A is an EfficientNet-L2 trained on 300 million images from JFT-300M (Chollet, 2017; Hinton et al., 2014) using a noisy student-teacher protocol (Xie et al., 2020a). We adapt this model for only one epoch due to resource constraints. During the hyperparameter search, we only evaluate three corruptions on the IN-C development set[4] and test the learning rates $4.6 \times 10^{-2}$, $4.6 \times 10^{-3}$, $4.6 \times 10^{-4}$ and $4.6 \times 10^{-5}$. We use the optimal hyperparameters to evaluate ENT and RPL on the full IN-C test set (with all severity levels).

**UDA-SS models**  We trained the models using the scripts from the official code base at github.com/yueatsprograms/uda_release. We used the provided scripts for the cases: (a) source: CIFAR10, target: STL10 and (b) source: MNIST, target: MNIST-M. For the case (c) source: CIFAR10, target: CIFAR10-C, we used the hyperparameters from case (a) since this case seemed to be the closest match to the new setting. We think that the baseline performance of the UDA-SS models can be further improved with hyperparameter tuning.

**DANN models**  To train models with the DANN-method, we used the PyTorch implementation of this paper at https://github.com/fungtion/DANN_py3. The code base only provides scripts and hyperparameters for the case (b) source: MNIST, target: MNIST-M. For the cases (a) and (c), we used the same optimizer and trained the model for 100 epochs. We think that the baseline performance of the DANN models can be further improved with hyperparameter tuning.

**Preprocessing**  For IN, IN-R, IN-A and IN-D, we resize all images to $256 \times 256$ px and take the center $224 \times 224$ px crop. The IN-C images are already rescaled and cropped. We center and re-scale the color values with $\mu_{RGB} = [0.485, 0.456, 0.406]$ and $\sigma_{RGB} = [0.229, 0.224, 0.225]$. For the EfficientNet-L2, we follow the

---

[4]We compare the results of computing the dev set on the 1, 3 and 5 severities versus the 1, 2, 3, 4 and 5 severities on our ResNeXt101 model in the Supplementary material.

procedure in Xie et al. (2020a) and rescale all inputs to a resolution of $507 \times 507$ px and then center-crop them to $475 \times 475$ px.

## B.2 Full list of used models

**ImageNet scale models**   ImageNet trained models (ResNet50, DenseNet161, ResNeXt) are taken directly from torchvision (Marcel & Rodriguez, 2010). The model variants trained with DeepAugment and AugMix augmentations (Hendrycks et al., 2020b;a) are taken from https://github.com/hendrycks/imagenet-r. The weakly-supervised ResNeXt101 model is taken from the PyTorch Hub. For EfficientNet (Tan & Le, 2019), we use the PyTorch re-implementation available at https://github.com/rwightman/gen-efficientnet-pytorch. This is a verified re-implementation of the original work by Xie et al. (2020a). We verify the performance on ImageNet, yielding a 88.23% top-1 accuracy and 98.546% top-5 accuracy which is within 0.2% points of the originally reported result (Xie et al., 2020a). On ImageNet-C, our reproduced baseline achieves 28.9% mCE vs. 28.3% mCE originally reported by Xie et al. (2020a). As noted in the re-implementation, this offset is possible due to minor differences in the pre-processing. It is possible that our adaptation results would improve further when applied on the original codebase by Xie *et al.*.

**Small scale models**   We train the UDA-SS models using the original code base at github.com/yueatsprograms/uda_release, with the hyperparameters given in the provided bash scripts. For our DANN experiments, we use the PyTorch implementation at github.com/fungtion/DANN_py3. We use the hyperparameters in the provided bash scripts.

The following Table 16 contains all models we evaluated on various datasets with references and links to the corresponding source code.

Table 16: Model checkpoints used for our experiments.

| Model | Source |
|---|---|
| WideResNet(28,10) (Croce et al., 2020) | https://github.com/RobustBench/robustbench/tree/master/robustbench |
| WideResNet(40,2)+AugMix (Croce et al., 2020) | https://github.com/RobustBench/robustbench/tree/master/robustbench |
| ResNet50 (He et al., 2016b) | https://github.com/pytorch/vision/tree/master/torchvision/models |
| ResNeXt101, 32×8d (He et al., 2016b) | https://github.com/pytorch/vision/tree/master/torchvision/models |
| DenseNet (Huang et al., 2017) | https://github.com/pytorch/vision/tree/master/torchvision/models |
| ResNeXt101, 32×8d (Xie et al., 2017) | https://pytorch.org/hub/facebookresearch_WSL-Images_resnext/ |
| ResNet50+DeepAugment+AugMix (Hendrycks et al., 2020a) | https://github.com/hendrycks/imagenet-r |
| ResNext101 (Hendrycks et al., 2020a) | https://github.com/hendrycks/imagenet-r |
| ResNext101 32×8d IG-3.5B (Mahajan et al., 2018) | https://github.com/facebookresearch/WSL-Images/blob/master/hubconf.py |
| Noisy Student EfficientNet-L2 (Xie et al., 2020a) | https://github.com/rwightman/gen-efficientnet-pytorch |
| ViT-S/16 (Caron et al., 2021) | https://github.com/facebookresearch/dino |

## C    Detailed and additional Results on IN-C

### C.1    Definition of the mean Corruption Error (mCE)

The established performance metric on IN-C is the mean Corruption Error (mCE), which is obtained by normalizing the model's top-1 errors with the top-1 errors of AlexNet across the C=15 test corruptions and S=5 severities:

$$\text{mCE(model)} = \frac{1}{C} \sum_{c=1}^{C} \frac{\sum_{s=1}^{S} \text{err}_{c,s}^{\text{model}}}{\sum_{s=1}^{S} \text{err}_{c,s}^{\text{AlexNet}}}. \tag{20}$$

The AlexNet errors used for normalization are shown in Table 17.

| Category | Corruption | top1 error |
|---|---|---|
| Noise | Gaussian Noise | 0.886428 |
| | Shot Noise | 0.894468 |
| | Impulse Noise | 0.922640 |
| Blur | Defocus Blur | 0.819880 |
| | Glass Blur | 0.826268 |
| | Motion Blur | 0.785948 |
| | Zoom Blur | 0.798360 |
| Weather | Snow | 0.866816 |
| | Frost | 0.826572 |
| | Fog | 0.819324 |
| | Brightness | 0.564592 |
| | Contrast | 0.853204 |
| Digital | Elastic Transform | 0.646056 |
| | Pixelate | 0.717840 |
| | JPEG Compression | 0.606500 |
| Hold-out Noise | Speckle Noise | 0.845388 |
| Hold-out Digital | Saturate | 0.658248 |
| Hold-out Blur | Gaussian Blur | 0.787108 |
| Hold-out Weather | Spatter | 0.717512 |

Table 17: AlexNet top1 errors on ImageNet-C

### C.2    Detailed results for tuning epochs and learning rates

We tune the learning rate for all models and the number of training epochs for all models except the EfficientNet-L2. In this section, we present detailed results for tuning these hyperparameters for all considered models. The best hyperparameters that we found in this analysis, are summarized in Table 22.

Table 18: mCE in % on the IN-C dev set for ENT and RPL for different numbers of training epochs when adapting the affine batch norm parameters of a ResNet50 model.

| criterion | ENT | | | RPL | | |
|---|---|---|---|---|---|---|
| lr | $10^{-4}$ | $10^{-3}$ | $10^{-2}$ | $10^{-4}$ | $10^{-3}$ | $10^{-2}$ |
| epoch | | | | | | |
| 0 | 60.2 | 60.2 | 60.2 | 60.2 | 60.2 | 60.2 |
| 1 | 54.3 | **50.0** | 72.5 | 57.4 | 51.1 | 52.5 |
| 2 | 52.4 | 50.9 | 96.5 | 55.8 | 49.6 | 57.4 |
| 3 | 51.5 | 51.0 | 112.9 | 54.6 | 49.2 | 64.2 |
| 4 | 51.0 | 52.4 | 124.1 | 53.7 | 49.0 | 71.0 |
| 5 | 50.7 | 53.5 | 131.2 | 52.9 | **48.9** | 76.3 |

Table 19: mCE ($\searrow$) in % on the IN-C dev set for different learning rates for EfficientNet-L2. We favor $q = 0.8$ over $q = 0.7$ due to slightly improved robustness to changes in the learning rate in the worst case error setting.

| lr (4.6 $\times$) | base | $1 \times 10^{-3}$ | $1 \times 10^{-4}$ | $1 \times 10^{-5}$ | $1 \times 10^{-6}$ |
|---|---|---|---|---|---|
| ENT | 25.5 | 87.8 | 25.3 | **22.2** | 24.1 |
| RPL$_{q=0.7}$ | 25.5 | 60.3 | 21.3 | 23.3 | n/a |
| RPL$_{q=0.8}$ | 25.5 | 58.2 | **21.4** | 23.4 | n/a |

Table 22: The best hyperparameters for all models that we found on IN-C. For all models, we fine-tune only the affine batch normalization parameters and use $q = 0.8$ for RPL. The small batchsize for the EfficientNet model is due to hardware limitations.

| Model | Method | Learning rate | batch size | number of epochs |
|---|---|---|---|---|
| vanilla ResNet50 | ENT | $1 \times 10^{-3}$ | 128 | 1 |
| vanilla ResNet50 | RPL | $1 \times 10^{-3}$ | 128 | 5 |
| vanilla ResNeXt101 | ENT | $2.5 \times 10^{-4}$ | 128 | 1 |
| vanilla ResNeXt101 | RPL | $2.5 \times 10^{-4}$ | 128 | 4 |
| IG-3.5B ResNeXt101 | ENT | $2.5 \times 10^{-4}$ | 128 | 4 |
| IG-3.5B ResNeXt101 | RPL | $2.5 \times 10^{-3}$ | 128 | 2 |
| DAug+AM ResNeXt101 | ENT | $2.5 \times 10^{-4}$ | 128 | 1 |
| DAug+AM ResNeXt101 | RPL | $2.5 \times 10^{-4}$ | 128 | 4 |
| EfficientNet-L2 | ENT | $4.6 \times 10^{-5}$ | 8 | 1 |
| EfficientNet-L2 | RPL | $4.6 \times 10^{-4}$ | 8 | 1 |

Table 20: mCE in % on IN-C dev for entropy minimization for different learning rates and training epochs for ResNeXt101. (div.=diverged)

| ENT lr 2.5 × epoch | Baseline | | | IG-3.5B | | | DAug+AM | | |
|---|---|---|---|---|---|---|---|---|---|
| | 1e-4 | 1e-3 | 5e-3 | 1e-4 | 1e-3 | 5e-3 | 1e-4 | 1e-3 | 5e-3 |
| BASE | 53.6 | 53.6 | 53.6 | 47.4 | 47.4 | 47.4 | 37.4 | 37.4 | 37.4 |
| 1 | **43.0** | 92.2 | div. | 40.9 | 40.4 | 58.6 | **35.4** | 46.4 | div. |
| 2 | 44.8 | 118.4 | div. | 39.8 | 41.5 | 69.5 | 35.5 | 90.8 | div. |
| 3 | 45.4 | 131.9 | div. | 39.3 | 42.6 | 76.1 | 35.5 | 122.5 | div. |
| 4 | 46.7 | div. | div. | **39.1** | 44.2 | 84.3 | 35.6 | 133.8 | div. |

Table 21: mCE in % on IN-C dev for robust pseudo-labeling for different learning rates and training epochs for ResNeXt101. (div.=diverged)

| RPL lr 2.5× epoch | Baseline | | | IG-3.5B | | | DAug+AM | | |
|---|---|---|---|---|---|---|---|---|---|
| | 1e-4 | 1e-3 | 5e-3 | 1e-4 | 1e-3 | 5e-3 | 1e-4 | 1e-3 | 5e-3 |
| BASE | 53.6 | 53.6 | 53.6 | 47.4 | 47.4 | 47.4 | 37.4 | 37.4 | 37.4 |
| 1 | 43.4 | 51.3 | div. | 45.0 | 39.9 | 43.6 | 35.3 | 35.1 | 79.1 |
| 2 | 42.3 | 63.2 | div. | 43.4 | **39.3** | 48.2 | 34.9 | 35.6 | 121.2 |
| 3 | 42.0 | 72.6 | div. | 42.4 | 39.4 | 52.9 | 34.7 | 40.1 | 133.5 |
| 4 | **42.0** | 72.6 | div. | 42.4 | 39.4 | 52.9 | **34.7** | 40.1 | 133.5 |

## C.3 Detailed results for all IN-C corruptions

We outline detailed results for all corruptions and models in Table 23. Performance across the severities in the dataset is depicted in Figure 6. All detailed results presented here are obtained by following the model selection protocol outlined in the main text.

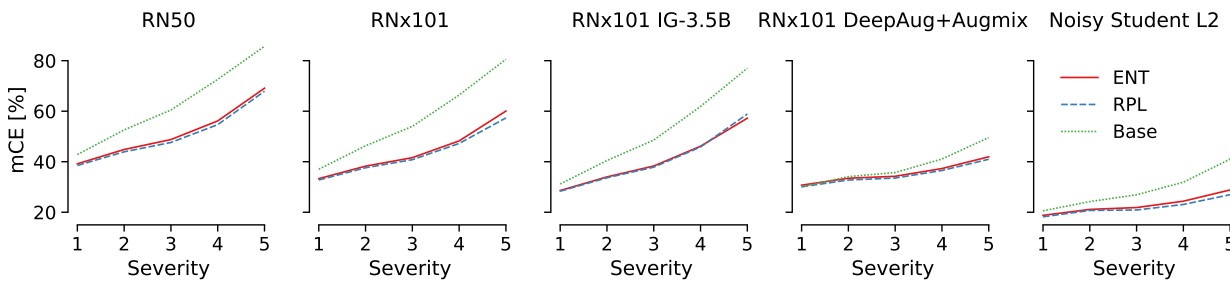

Figure 6: Severity-wise mean corruption error (normalized using the *average* AlexNet baseline error for each corruption) for ResNet50 (RN50), ResNext101 (RNx101) variants and the Noisy Student L2 model. Especially for more robust models (DeepAugment+Augmix and Noisy Student L2), most gains are obtained across higher severities 4 and 5. For weaker models, the baseline variant (Base) is additionally substantially improved for smaller corruptions.

Table 23: Detailed results for each corruption along with mean corruption error (mCE) as reported in Table 2 in the main paper. We show (unnormalized) top-1 error rate averaged across 15 test corruptions along with the mean corruption error (mCE: which is normalized). Hyperparameter selection for both ENT and RPL was carried out on the dev corruptions as outlined in the main text. Mismatch in baseline mCE for EfficientNet-L2 can be most likely attributed to pre-processing differences between the original tensorflow implementation Xie et al. (2020a) and the PyTorch reimplementation we employ. We start with slightly weaker baselines for ResNet50 and ResNext101 than Schneider et al. (2020): ResNet50 and ResNext101 results are slightly worse than previously reported results (typically 0.1% points) due to the smaller batch size of 128 and 32. Smaller batch sizes impact the quality of re-estimated batch norm statistics when computation is performed on the fly Schneider et al. (2020), which is of no concern here due to the large gains obtained by pseudo-labeling.

| | gauss | shot | impulse | defocus | glass | motion | zoom | snow | frost | fog | bright | contrast | elastic | pixelate | jpeg | **mCE** |
|---|---|---|---|---|---|---|---|---|---|---|---|---|---|---|---|---|
| **ResNet50** | | | | | | | | | | | | | | | | |
| Baseline (Schneider et al., 2020) | | | | | | | | | | | | | | | | 62.2 |
| Baseline (ours) | 57.2 | 59.5 | 60.0 | 61.4 | 62.3 | 51.3 | 49.5 | 54.6 | 54.1 | 39.3 | 29.1 | 46.7 | 41.4 | 38.2 | 41.8 | 62.8 |
| ENT | 45.5 | 45.5 | 46.8 | 48.4 | 48.7 | 40.0 | 40.3 | 42.0 | 46.6 | 33.2 | 28.1 | 42.4 | 35.2 | 32.2 | 35.1 | 51.6 |
| RPL | 44.2 | 44.4 | 45.5 | 47.0 | 47.4 | 38.8 | 39.2 | 40.7 | 46.2 | 32.5 | 27.7 | 42.7 | 34.6 | 31.6 | 34.4 | 50.5 |
| **ResNeXt101 Baseline** | | | | | | | | | | | | | | | | |
| Baseline (Schneider et al., 2020) | | | | | | | | | | | | | | | | 56.7 |
| Baseline (ours) | 52.8 | 54.1 | 54.0 | 55.4 | 56.8 | 46.7 | 46.6 | 48.5 | 49.4 | 36.6 | 25.4 | 42.8 | 37.8 | 32.5 | 36.7 | 56.8 |
| ENT | 40.5 | 39.5 | 41.4 | 41.6 | 43.0 | 34.1 | 34.5 | 35.0 | 39.4 | 28.5 | 24.0 | 33.8 | 30.3 | 27.2 | 30.5 | 44.3 |
| RPL | 39.4 | 38.9 | 39.8 | 40.3 | 41.0 | 33.4 | 33.8 | 34.6 | 38.7 | 28.0 | 23.7 | 31.4 | 29.8 | 26.8 | 30.0 | 43.2 |
| **ResNeXt101 IG-3.5B** | | | | | | | | | | | | | | | | |
| Baseline (Schneider et al., 2020) | | | | | | | | | | | | | | | | 51.6 |
| Baseline (ours) | 50.7 | 51.5 | 53.1 | 54.2 | 55.5 | 45.5 | 44.7 | 41.7 | 42.0 | 28.1 | 20.1 | 33.8 | 35.4 | 27.8 | 33.9 | 51.8 |
| ENT | 38.6 | 38.3 | 40.4 | 41.4 | 41.5 | 33.8 | 33.6 | 32.2 | 34.6 | 24.1 | 19.7 | 26.3 | 27.6 | 24.2 | 27.9 | 40.8 |
| RPL | 39.1 | 39.2 | 40.8 | 42.1 | 42.4 | 33.7 | 33.5 | 31.8 | 34.7 | 23.9 | 19.6 | 26.1 | 27.5 | 23.8 | 27.5 | 40.9 |
| **ResNeXt101 DeepAug+Augmix** | | | | | | | | | | | | | | | | |
| Baseline (Schneider et al., 2020) | | | | | | | | | | | | | | | | 38.0 |
| Baseline (ours) | 30.0 | 30.0 | 30.2 | 32.9 | 35.5 | 28.9 | 31.9 | 33.3 | 32.8 | 29.5 | 22.6 | 28.4 | 31.2 | 23.0 | 26.5 | 38.1 |
| ENT | 28.7 | 28.5 | 29.0 | 29.8 | 30.9 | 26.9 | 28.0 | 29.3 | 30.5 | 26.2 | 23.2 | 26.3 | 28.5 | 23.7 | 26.0 | 35.5 |
| RPL | 28.1 | 27.8 | 28.3 | 29.1 | 30.1 | 26.3 | 27.4 | 28.8 | 29.8 | 25.9 | 22.7 | 25.6 | 27.9 | 23.2 | 25.4 | 34.8 |
| **Noisy Student L2** | | | | | | | | | | | | | | | | |
| Baseline (Xie et al., 2020a) | | | | | | | | | | | | | | | | 28.3 |
| Baseline (ours) | 21.6 | 22.0 | 20.5 | 23.9 | 40.5 | 19.8 | 23.2 | 22.8 | 26.9 | 21.0 | 15.2 | 21.2 | 24.8 | 17.9 | 18.6 | 28.9 |
| ENT | 18.5 | 18.7 | 17.4 | 18.8 | 23.4 | 16.9 | 18.8 | 17.1 | 19.6 | 16.8 | 14.1 | 16.6 | 19.6 | 15.8 | 16.5 | 23.0 |
| RPL | 17.8 | 18.0 | 17.0 | 18.1 | 21.4 | 16.4 | 17.9 | 16.4 | 18.7 | 15.7 | 13.6 | 15.6 | 19.2 | 15.0 | 15.6 | 22.0 |

### C.4 Detailed results for the CIFAR10-C and UDA adaptation

Table 24: Detailed results for each corruption along with mean error on CIFAR10-C as reported in Table 2 in the main paper.

| | gauss | shot | impulse | defocus | glass | motion | zoom | snow | frost | fog | bright | contrast | elastic | pixelate | jpeg | **avg** |
|---|---|---|---|---|---|---|---|---|---|---|---|---|---|---|---|---|
| WRN-28-10 vanilla | | | | | | | | | | | | | | | | |
| Baseline | 53.0 | 41.2 | 44.7 | 18.5 | 49.0 | 22.3 | 24.4 | 18.1 | 25.0 | 11.2 | 6.7 | 17.4 | 16.2 | 28.0 | 22.4 | 26.5 |
| BN adapt | 20.8 | 17.6 | 22.7 | 8.1 | 28.4 | 10.9 | 9.2 | 14.2 | 13.0 | 8.7 | 6.8 | 8.5 | 13.5 | 12.1 | 21.0 | 14.4 |
| ENT | 18.5 | 15.9 | 20.6 | 7.8 | 25.5 | 10.6 | 8.5 | 13.1 | 12.3 | 8.3 | 6.9 | 8.0 | 12.6 | 11.1 | 18.9 | 13.3 |
| RPL | 19.1 | 21.4 | 16.3 | 8.1 | 26.4 | 8.9 | 10.9 | 13.7 | 12.9 | 6.9 | 13.1 | 19.7 | 8.2 | 11.4 | 8.7 | 13.7 |
| WRN-40-2 AM | | | | | | | | | | | | | | | | |
| Baseline | 19.1 | 14.0 | 13.3 | 6.3 | 17.1 | 7.9 | 7.0 | 10.4 | 10.6 | 8.5 | 5.9 | 9.7 | 9.2 | 16.8 | 11.9 | 11.2 |
| BN adapt | 14.1 | 11.9 | 13.9 | 7.2 | 17.6 | 8.7 | 7.9 | 10.8 | 10.6 | 9.0 | 6.8 | 9.0 | 10.9 | 10.1 | 14.0 | 10.8 |
| ENT | 10.8 | 9.1 | 10.9 | 6.0 | 13.4 | 7.2 | 6.3 | 8.4 | 7.8 | 7.1 | 5.7 | 7.1 | 9.2 | 7.4 | 11.2 | 8.5 |
| RPL | 11.5 | 11.6 | 9.6 | 6.2 | 14.2 | 6.5 | 7.4 | 8.8 | 8.2 | 6.0 | 9.5 | 11.9 | 7.9 | 8.0 | 7.6 | 9.0 |
| WRN-26-16 UDA-SS | | | | | | | | | | | | | | | | |
| Baseline | 26.0 | 24.7 | 19.3 | 22.4 | 56.2 | 32.4 | 32.1 | 31.7 | 31.2 | 26.6 | 15.8 | 20.4 | 26.3 | 21.5 | 28.9 | 27.7 |
| BN adapt | 20.5 | 19.0 | 15.6 | 13.5 | 43.1 | 19.4 | 18.3 | 23.1 | 21.2 | 16.2 | 12.8 | 14.1 | 20.9 | 16.7 | 23.4 | 19.9 |
| ENT | 16.9 | 16.7 | 12.3 | 11.3 | 37.6 | 15.6 | 14.8 | 18.3 | 18.2 | 13.4 | 10.8 | 11.9 | 17.9 | 14.4 | 20.9 | 16.7 |
| RPL | 18.1 | 17.1 | 13.2 | 11.9 | 41.5 | 17.3 | 16.1 | 20.4 | 19.1 | 14.5 | 11.8 | 12.7 | 18.8 | 18.1 | 22.6 | 18.2 |
| WRN-26-16 vanilla | | | | | | | | | | | | | | | | |
| Baseline | 50.8 | 46.9 | 39.3 | 15.9 | 44.2 | 20.8 | 18.8 | 14.9 | 17.8 | 4.8 | 13.6 | 20.4 | 19.0 | 25.0 | 10.0 | 24.2 |
| BN adapt | 18.6 | 20.4 | 15.6 | 6.1 | 24.9 | 7.4 | 8.3 | 11.6 | 10.8 | 5.3 | 11.4 | 18.9 | 6.8 | 9.9 | 6.8 | 12.2 |
| ENT | 16.7 | 18.4 | 13.9 | 5.7 | 23.0 | 6.8 | 7.9 | 10.8 | 9.9 | 5.0 | 10.8 | 17.3 | 6.4 | 9.1 | 6.4 | 11.2 |
| RPL | 16.1 | 18.0 | 13.6 | 6.6 | 23.2 | 8.7 | 9.3 | 11.9 | 11.1 | 6.4 | 11.7 | 17.0 | 7.4 | 9.0 | 7.2 | 11.8 |

Table 25: Detailed results for the UDA methods reported in Table 2 of the main paper.

| | Baseline | BN adapt | RPL | ENT |
|---|---|---|---|---|
| UDA CIFAR10→STL10, top1 error on target [%]($\searrow$) | | | | |
| WRN-26-16 UDA-SS | 28.7 | 24.6 | 22.9 | 21.8 |
| WRN-26-16 DANN | 25.0 | 25.0 | 24.0 | 23.9 |
| UDA MNIST→MNIST-M, top1 error on target [%]($\searrow$) | | | | |
| WRN-26-16 UDA-SS | 4.8 | 3.9 | 2.4 | 2.0 |
| WRN-26-2 DANN | 11.4 | 6.2 | 5.2 | 5.1 |

### C.5 Ablation over the hyperparameter $q$ for RPL

For RPL, we must choose the hyperparameter $q$. We performed an ablation study over $q$ and show results in Table 26, demonstrating that RPL is robust to the choice of $q$, with slight preference to higher values. Note: In the initial parameter sweep for this paper, we only compared $q = 0.7$ and $q = 0.8$. Given the result in Table 26, it could be interesting to re-run the models in Table 1 of the main paper with $q = 0.9$, which could yield another (small) improvement in mCE.

Table 26: ImageNet-C dev set mCE in %, vanilla ResNet50, batch size 96. We report the best score across a maximum of six adaptation epochs.

| q | 0.5 | 0.6 | 0.7 | 0.8 | 0.9 |
|---|---|---|---|---|---|
| mCE (dev) | 49.5 | 49.3 | 49.2 | 49.2 | 49.1 |

### C.6 Self-learning outperforms Test-Time Training (Sun et al., 2020)

Sun et al. (2020) use a ResNet18 for their experiments on ImageNet and only evaluate their method on severity 5 of IN-C. To enable a fair comparison, we trained a ResNet18 with both hard labeling and RPL and compare the efficacy of both methods to Test-Time Training in Table 27. For both hard labeling and RPL,

we use the hyperparameters we found for the vanilla ResNet50 model and thus, we expect even better results for hyperparameters tuned on the vanilla ResNet18 model and following our general hyperparameter search protocol.

While all methods (self-learning and TTT) improve the performance over a simple vanilla ResNet18, we note that even the very simple baseline using hard labeling already outperfoms Test-Time Training; further gains are possible with RPL. The result highlights the importance of simple baselines (like self-learning) when proposing new domain adaptation schemes. It is likely that many established DA techniques more complex than the basic self-learning techniques considered in this work will even further improve over TTT and other adaptation approaches developed exclusively in robustness settings.

We report better accuracy numbers achieved with self-learning compared to online TTT, but note that they adapt to a single test sample while we make use of batches of data. Due to the single-image approach, they utilize GN instead of BN, and this may explain the performance gap to a certain degree as we show that BN is more effective for test-time adaptation compared to GN, even though the un-adapted BN models are less robust compared to the un-adapted GN models, as also noted by Sun et al. (2020). Further, Sun et al. (2020) optimize all model parameters while we only optimize the affine BN parameters which works better.

Table 27: Comparison of hard-pseudo labeling and robust pseudo-labeling to Test-Time Training Sun et al. (2020): Top-1 error for a ResNet18 and severity 5 for all corruptions. Simple hard pseudo-labeling already outperforms TTT, robust pseudo labeling over multiple epochs yields additional gains.

| | gauss | shot | impulse | defocus | glass | motion | zoom | snow | frost | fog | bright | contrast | elastic | pixelate | jpeg | **Avg** |
|---|---|---|---|---|---|---|---|---|---|---|---|---|---|---|---|---|
| vanilla ResNet18 | 98.8 | 98.2 | 99.0 | 88.6 | 91.3 | 88.8 | 82.4 | 89.1 | 83.5 | 85.7 | 48.7 | 96.6 | 83.2 | 76.9 | 70.4 | 85.4 |
| Test-Time Training | 73.7 | 71.4 | 73.1 | 76.3 | 93.4 | 71.3 | 66.6 | 64.4 | 81.3 | 52.4 | 41.7 | **64.7** | 55.7 | 52.2 | 55.7 | 66.3 |
| hard PL, (1 epoch) | 73.2 | 70.8 | 73.6 | 76.5 | 75.6 | 63.9 | 56.1 | 59.0 | **65.9** | 48.4 | 39.7 | 85.2 | 50.4 | 47.0 | 51.5 | 62.5 |
| ENT(1 epoch) | 72.8 | 69.8 | 73.2 | 77.2 | 75.7 | 63.1 | 55.5 | 58.0 | 68.1 | 48.0 | 39.8 | 92.7 | 49.6 | 46.4 | 51.3 | 62.8 |
| RPL (4 epochs) | **71.3** | **68.3** | **71.7** | **76.2** | **75.6** | **61.5** | **54.4** | **56.9** | 67.1 | **47.3** | **39.3** | 93.2 | **48.9** | **45.7** | **50.4** | **61.9** |

### C.7 Comparison to Meta Test-Time Training (Bartler et al., 2022)

Bartler et al. (2022) report an error of 24.4% as their top result on CIFAR10-C which is far worse than our top results of 8.5% with an AugMix trained model, or 13.3% with a vanilla trained model, but a different architecture: they used a WRN-26-1 while we report results with a WRN-40-2. Thus, to make the comparison more fair, we tested our approach on their model architecture.

A direct comparison is not straight-forward since Bartler et al. (2022) trained their models using Keras while our code-base is in PyTorch. Therefore, we first trained a baseline model in PyTorch on clean CIFAR10 using their architecture with the standard and widely used CIFAR10 training code available at https://github.com/kuangliu/pytorch-cifar (1.9k forks, 4.8k stars); the baseline test accuracy on clean CIFAR10 using the architecture of Bartler et al. (2022) is at 94.3%. As a second step, we adapted the baseline model with ENT and RPL on CIFAR10-C. Please see Table 28 for our results. BN adaptation is not possible for the model used by Bartler et al. (2022) since they use GroupNorm instead of BatchNorm. Due to the usage of GroupNorm, it is not evident whether the affine GroupNorm parameters should be adapted, or all model parameters. Thus, we tested both, and report the results for both adaptation mechanisms.

We find that adaptation of affine GN layers works better than full model adaptation, consistent with our results for the adaptation of BN layers. As the gains due to ENT and RPL seem lower than for our WRN-40-2 architecture, we hypothesize that the issue lies in the GN layers as the presence of GN instead of BN layers is the main difference between our WRN-40-2 and the new WRN-26-1 model. Therefore, we trained another WRN-26-1 model with BN layers instead of GN, and adapted this model using BN, ENT and RPL. The baseline accuracy on clean CIFAR10 of WRN-26-1-BN is 95.04%. Indeed, we find that adapting the model with BN instead of GN layers leads to much larger gains, which is consistent with findings by Schneider et al. (2020) who found that BN adaptation outperforms non-adapted models trained with GN. Thus, we conclude

that self-learning techniques work better with models which have BN layers and less well with models with GN layers. For both model types (with GN or with BN layers), ENT works better than RPL which is consistent with our other results on small-scale datasets.

Overall, our best result for the model architecture used by Bartler et al. (2022) (after replacing GN layers with BN layers) et al. is 13.1% which is much lower than their best result of 24.4%. Even when using GN layers, our best top-1 error is 18.0% which is significantly lower than the best result of Bartler et al. (2022).

Table 28: Detailed results for our comparison to MT3 (Bartler et al., 2022)

| | gauss | shot | impulse | defocus | glass | motion | zoom | snow | frost | fog | bright | contrast | elastic | pixelate | jpeg | **avg** |
|---|---|---|---|---|---|---|---|---|---|---|---|---|---|---|---|---|
| WRN-26-1-GN vanilla | | | | | | | | | | | | | | | | |
| Baseline (Bartler et al., 2022) | 50.5 | 47.2 | 56.1 | 23.7 | 51.7 | 24.3 | 26.3 | 25.6 | 34.4 | 28.1 | 13.5 | 25.0 | 27.4 | 55.8 | 29.8 | 34.6 |
| adapted (Bartler et al., 2022) | 30.1 | 29.5 | 41.8 | 15.6 | 33.7 | 22.8 | 18.7 | 20.2 | 18.8 | 24.1 | 13.8 | 22.4 | 23.7 | 27.6 | 22.7 | 24.4 |
| WRN-26-1-GN vanilla | | | | | | | | | | | | | | | | |
| Baseline [ours] | 39.1 | 32.0 | 30.2 | 10.9 | 31.9 | 13.5 | 13.3 | 14.1 | 16.7 | 10.6 | 7.3 | 9.4 | 14.1 | 16.1 | 20.6 | 18.6 |
| ENT [GN layers, ours] | 41.6 | 30.7 | 32.8 | 9.7 | 30.9 | 11.8 | 11.4 | 13.5 | 14.9 | 10.0 | 7.2 | 8.9 | 13.2 | 13.3 | 19.8 | 18.0 |
| RPL [GN layers, ours] | 39.3 | 31.6 | 30.6 | 10.6 | 31.6 | 13.0 | 12.6 | 14.0 | 16.1 | 10.4 | 7.3 | 9.2 | 13.9 | 15.4 | 20.4 | 18.4 |
| ENT [full model, ours] | 61.2 | 46.0 | 37.6 | 9.1 | 30.2 | 11.1 | 10.4 | 13.1 | 14.4 | 10.0 | 7.2 | 8.7 | 13.1 | 11.9 | 19.7 | 20.3 |
| RPL [full model, ours] | 54.1 | 37.3 | 33.0 | 10.2 | 31.5 | 11.9 | 12.4 | 13.6 | 15.3 | 7.3 | 13.6 | 20.4 | 9.2 | 13.6 | 10.2 | 19.6 |
| WRN-26-1-BN vanilla | | | | | | | | | | | | | | | | |
| Baseline [ours] | 55.7 | 44.8 | 43.1 | 15.5 | 44.2 | 20.5 | 21.1 | 17.2 | 20.7 | 6.6 | 15.6 | 21.3 | 23.6 | 24.7 | 11.8 | 25.8 |
| BN adapt [ours] | 28.0 | 28.9 | 24.2 | 10.5 | 31.5 | 12.7 | 14.0 | 18.4 | 18.1 | 9.4 | 17.1 | 26.9 | 13.2 | 15.3 | 12.7 | 18.7 |
| ENT [BN layers, ours] | 17.9 | 19.6 | 14.9 | 8.1 | 23.2 | 9.1 | 10.4 | 13.1 | 13.2 | 7.2 | 13.2 | 18.0 | 9.9 | 10.5 | 8.9 | **13.1** |
| RPL [BN layers, ours] | 21.3 | 22.0 | 17.9 | 9.3 | 26.6 | 10.5 | 11.8 | 15.1 | 15.0 | 7.8 | 14.7 | 20.8 | 12.1 | 11.8 | 10.4 | 15.1 |

## C.8 Effect of batch size and linear learning rate scaling

How is self-learning performance affected by batch size constraints? We compare the effect of different batch sizes and linear learning rate scaling. In general, we found that affine adaptation experiments on ResNet50 scale can be run with batch size 128 on a Nvidia V100 GPU (16GB), while only batch size 96 experiments are possible on RTX 2080 GPUs.

The results in Table 29 show that for a ResNet50 model, higher batch size yields a generally better performance.

Table 29: ImageNet-C dev set mCE for various batch sizes with linear learning rate scaling. All results are computed for a vanilla ResNet50 model using RPL with $q = 0.8$, reporting the best score across a maximium of six adaptation epochs.

| batch size | 16 | 32 | 64 | 80 | 96 | 128 |
|---|---|---|---|---|---|---|
| learning rate ($\times 10^{-3}$) | 0.125 | 0.250 | 0.500 | 0.625 | 0.750 | 1 |
| dev mCE | 53.8 | 51.0 | 49.7 | 49.3 | 49.2 | 48.9 |

## C.9 Performance over different seeds in a ResNet50 on ImageNet-C

To limit the amount of compute, we ran RPL and ENT for our vanilla ResNet50 model three times with the optimal hyperparameters. The averaged results, displayed as "mean (unbiased std)" are:

Table 30: ImageNet-C performance for three seeds on a ResNet50 for ENT and RPL.

| ResNet50 + self-learning | mCE on IN-C dev [%] | mCE on IN-C test [%] |
|---|---|---|
| ENT | 50.0 (0.04) | 51.6 (0.04) |
| RPL | 48.9 (0.02) | 50.5 (0.03) |

### C.10 Self-learning as continuous test-test adaptation

We test our method on continuous test-time adaptation where the model adapts to a continuous stream of data from the same domain. In Fig. 7, we display the error of the Noisy Student L2 model while it is being adapted to ImageNet-C and ImageNet-R. The model performance improves as the model sees more data from the new domain. We differentiate continuous test-time adaptation from the online test-time adaptation setting (Zhang et al., 2021) where the model is adapted to each test sample individually, and reset after each test sample.

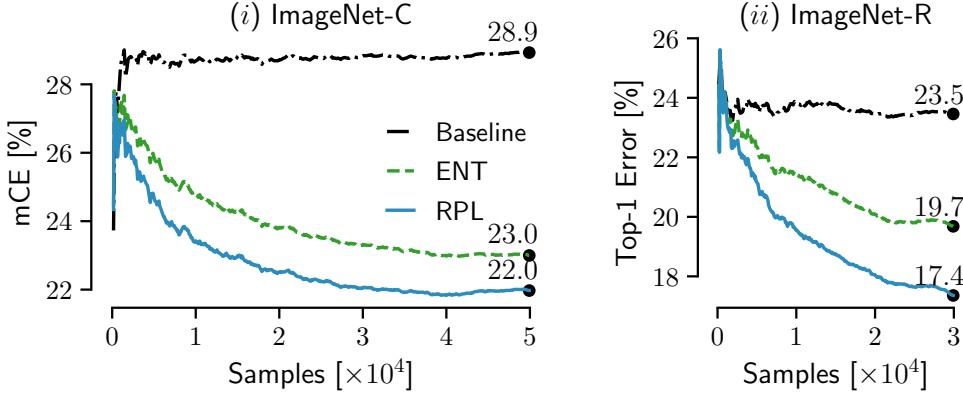

Figure 7: Evolution of error during online adaptation for EfficientNet-L2.

Table 31: Statistics of one-to-many mappings from IN-D to ImageNet.

| Number of IN classes one IN-D class is mapped to | 1 | 2 | 3 | 4 | 5 | 6 | 7 | 8 | 13 | 28 | 39 | 132 |
|---|---|---|---|---|---|---|---|---|---|---|---|---|
| Frequency of these mappings | 102 | 32 | 13 | 3 | 5 | 1 | 1 | 2 | 1 | 2 | 1 | 1 |

# D    Detailed and additional Results on IN-D

## D.1    Detailed protocol for label mapping from DomainNet to ImageNet

The mapping was first done by comparing the class labels in DomainNet and the synset labels on ImageNet. Afterwards, the resulting label maps were cleaned manually, because simply comparing class label strings resulted in imperfect matches. For example, images of the class "hot dog" in DomainNet were mapped to the class "dog" in IN. Another issue is that IN synset labels of different animal species do not contain the animal name in the text label, e.g., the class "orangutan, orang, orangutang, Pongo pygmaeus" does not contain the word "monkey" and we had to add this class to the hierarchical class "monkey" manually. We verified the mappings by investigating the class-confusion matrix of the true DomainNet class and the predicted ImageNet classes remapped to DomainNet on the "Real" domain, and checked that the predictions lay on the main diagonal, indicating that ImageNet classes have not been forgotten. The statistics for the mappings are shown in Table 31. Most IN-D classes (102) are mapped to one single ImageNet class. A few IN-D classes are mapped to more than 20 ImageNet classes: the IN-D classes "monkey" and "snake" are mapped to 28 ImageNet monkey and snake species classes, the IN-D class "bird" is mapped to 39 ImageNet bird species classes, and the IN-D class "dog" is mapped to 132 ImageNet dog breed classes.

We have considered max-pooling predictions across all sub-classes according to the ImageNet class hierarchy following the approach of Taori et al. (2020) for Youtube-BB and ImageNet-Vid (Recht et al., 2019). However, Radford et al. (2021) note that the resulting mappings are sometimes "much less than perfect", thus, we decided to clean the mappings ourselves to increase the mappings' quality. The full dictionary of the mappings will be released alongside the code.

## D.2    Evaluation protocol on IN-D

The domains in IN-D differ in terms of their difficulty for the studied models. Therefore, to calculate an aggregate score, we propose normalizing the error rates by the error achieved by AlexNet on the respective domains to calculate the mean error, following the approach in Hendrycks & Dietterich (2019) for IN-C. This way, we obtain the aggregate score mean Domain Error (mDE) by calculating the mean over different domains,

$$\text{DE}_d^f = \frac{E_d^f}{E_d^{\text{AlexNet}}}, \qquad \text{mDE} = \frac{1}{D}\sum_{d=1}^{D} E_d^f, \tag{21}$$

where $E_d^f$ is the top-1 error of a classifier $f$ on domain $d$.

**Leave-one-out-cross-validation**    For all IN-D results we report in this paper, we chose the hyperparameters on the IN-C dev set. We tried a different model selection scheme on IN-D as a control experiment with "Leave one out cross-validation" (L1outCV): with a round-robin procedure, we choose the hyperparameters for the test domain on all other domains. We select the same hyperparameters as when tuning on the "dev" set: For the ResNet50 model, we select over the number of training epochs (with a maximum of 7 training epochs) and search for the optimal learning rate in the set [0.01, 0.001, 0.0001]. For the EfficientNet-L2 model, we train only for one epoch as before and select the optimal learning rate in the set $[4.6 \times 10^{-3},$ $4.6 \times 10^{-4}, 4.6 \times 10^{-5}, 4.6 \times 10^{-6}]$. This model selection leads to worse results both for the ResNet50 and the EfficientNet-L2 models, highlighting the robustness of our model selection process, see Table 32.

Table 32: mDE in % on IN-D for different model selection strategies.

| model | model selection | |
| --- | --- | --- |
| | L1outCV | IN-C dev |
| ResNet50 RPL$_{q=0.8}$ | 81.3 | 76.1 |
| ResNet50 ENT | 82.4 | 77.3 |
| EfficientNet-L2 ENT | 69.2 | 66.8 |
| EfficientNet-L2 RPL$_{q=0.8}$ | 69.1 | 67.2 |

### D.3 Detailed results for robust ResNet50 models on IN-D

We show detailed results for all models on IN-D for vanilla evaluation (Table 33) BN adaptation (Table 34), RPL$_{q=0.8}$ (Table 35) and ENT(Table 36). For RPL$_{q=0.8}$ and ENT, we use the same hyperparameters that we chose on our IN-C 'dev' set. This means we train the models for 5 epochs with RPL$_{q=0.8}$ and for one epoch with ENT.

We evaluate the pre-trained and public checkpoints of SIN (Geirhos et al., 2019), ANT (Rusak et al., 2020), ANT+SIN (Rusak et al., 2020), AugMix (Hendrycks et al., 2020b), DeepAugment (Hendrycks et al., 2020a) and DeepAug+Augmix (Hendrycks et al., 2020a) in the following tables.

Table 33: Top-1 error on IN-D in % as obtained by robust ResNet50 models. For reference, we also show the mCE on IN-C and the top-1 error on IN-R. See main test for model references.

| Model | Clipart | Infograph | Painting | Quickdraw | Real | Sketch | mDE | IN-C | IN-R |
| --- | --- | --- | --- | --- | --- | --- | --- | --- | --- |
| vanilla | 76.0 | 89.6 | 65.1 | 99.2 | 40.1 | 82.0 | 88.2 | 76.7 | 63.9 |
| SIN | 71.3 | 88.6 | 62.6 | 97.5 | 40.6 | 77.0 | 85.6 | 69.3 | 58.5 |
| ANT | 73.4 | 88.9 | 63.3 | 99.2 | 39.9 | 80.8 | 86.9 | 62.4 | 61.0 |
| ANT+SIN | 68.4 | 88.6 | 60.6 | 95.5 | 40.8 | 70.3 | 83.1 | 60.7 | 53.7 |
| AugMix | 70.8 | 88.6 | 62.1 | 99.1 | 39.0 | 78.5 | 85.4 | 65.3 | 58.9 |
| DeepAugment | 72.0 | 88.8 | 61.4 | 98.9 | 39.4 | 78.5 | 85.6 | 60.4 | 57.8 |
| DeepAug+Augmix | 68.4 | 88.1 | 58.7 | 98.2 | 39.2 | 75.2 | 83.4 | 53.6 | 53.2 |

Table 34: Top1 error on IN-D in % as obtained by state-of-the-art robust ResNet50 models and batch norm adaptation, with a batch size of 128. See main text for model references.

| Model | Clipart | Infograph | Painting | Quickdraw | Real | Sketch | mDE |
| --- | --- | --- | --- | --- | --- | --- | --- |
| vanilla | 70.2 | 88.2 | 63.5 | 97.8 | 41.1 | 78.3 | 80.2 |
| SIN | 67.3 | 89.7 | 62.2 | 97.2 | 44.0 | 75.2 | 79.6 |
| ANT | 69.2 | 89.4 | 63.0 | 97.5 | 42.9 | 79.5 | 80.7 |
| ANT+SIN | 64.9 | 88.2 | 60.0 | 96.8 | 42.6 | 73.0 | 77.8 |
| AugMix | 66.9 | 88.1 | 61.2 | 97.1 | 40.4 | 75.0 | 78.4 |
| DeepAugment | 66.6 | 89.7 | 60.0 | 97.2 | 42.5 | 75.1 | 78.8 |
| DeepAug+Augmix | 61.9 | 85.7 | 57.5 | 95.3 | 40.2 | 69.2 | 74.9 |

Table 35: Top-1 error on IN-D in % as obtained by state-of-the-art robust ResNet50 models and RPL$_{q=0.8}$. See main text for model references.

| Model | Clipart | Infograph | Painting | Quickdraw | Real | Sketch | mDE |
| --- | --- | --- | --- | --- | --- | --- | --- |
| vanilla | 63.6 | 85.1 | 57.8 | 99.8 | 37.3 | 73.0 | 76.1 |
| SIN | 60.8 | 86.4 | 56.0 | 99.0 | 37.8 | 67.0 | 76.8 |
| ANT | 63.4 | 86.3 | 57.7 | 99.2 | 37.7 | 71.0 | 78.1 |
| ANT+SIN | 61.5 | 86.4 | 56.8 | 97.0 | 39.0 | 67.1 | 76.1 |
| AugMix | 59.7 | 83.4 | 54.1 | 98.2 | 35.6 | 70.1 | 74.6 |
| DeepAugment | 58.1 | 84.6 | 53.3 | 99.0 | 36.2 | 64.2 | 74.8 |
| DeepAug+Augmix | 57.0 | 83.2 | 53.4 | 99.1 | 36.5 | 61.3 | 72.6 |

Table 36: Top-1 error on IN-D in % as obtained by state-of-the-art robust ResNet50 models and ENT. See main text for references to the used models.

| Model | Clipart | Infograph | Painting | Quickdraw | Real | Sketch | mDE |
|---|---|---|---|---|---|---|---|
| vanilla | 65.1 | 85.8 | 59.2 | 98.5 | 38.4 | 75.8 | 77.3 |
| SIN | 62.1 | 87.0 | 57.3 | 99.1 | 39.0 | 68.6 | 75.5 |
| ANT | 64.2 | 86.9 | 58.7 | 97.1 | 38.8 | 72.8 | 76.5 |
| ANT+SIN | 62.2 | 86.8 | 57.7 | 95.8 | 40.1 | 68.7 | 75.2 |
| AugMix | 60.2 | 84.6 | 55.8 | 97.6 | 36.8 | 72.0 | 74.4 |
| DeepAugment | 59.5 | 85.7 | 54.4 | 98.0 | 37.1 | 66.4 | 73.3 |
| DeepAug+Augmix | 58.4 | 84.3 | 54.7 | 98.5 | 38.1 | 63.6 | 72.7 |

Table 37: mDE on IN-D in % as obtained by robust ResNet50 models with a baseline evaluation, batch norm adaptation, $RPL_{q=0.8}$ and ENT. See main text for model references.

| | mDE on IN-D ($\searrow$) | | | |
|---|---|---|---|---|
| Model | Baseline | BN adapt | $RPL_{q=0.8}$ | ENT |
| vanilla | 88.2 | 80.2 | 76.1 | 77.3 |
| SIN | 85.6 | 79.6 | 76.8 | 75.5 |
| ANT | 86.9 | 80.7 | 78.1 | 76.5 |
| ANT+SIN | **83.1** | 77.8 | 76.1 | 75.2 |
| AugMix | 85.4 | 78.4 | 74.6 | 74.4 |
| DeepAugment | 85.6 | 78.8 | 74.8 | 73.3 |
| DeepAugment+Augmix | 83.4 | **74.9** | **72.6** | **72.7** |

The summary results for all models are shown in Table 37.

We show the top-1 error for the different IN-D domains versus training epochs for a vanilla ResNet50 in Fig. 8. We indicate the epochs 1 and 5 at which we extract the errors with dashed black lines.

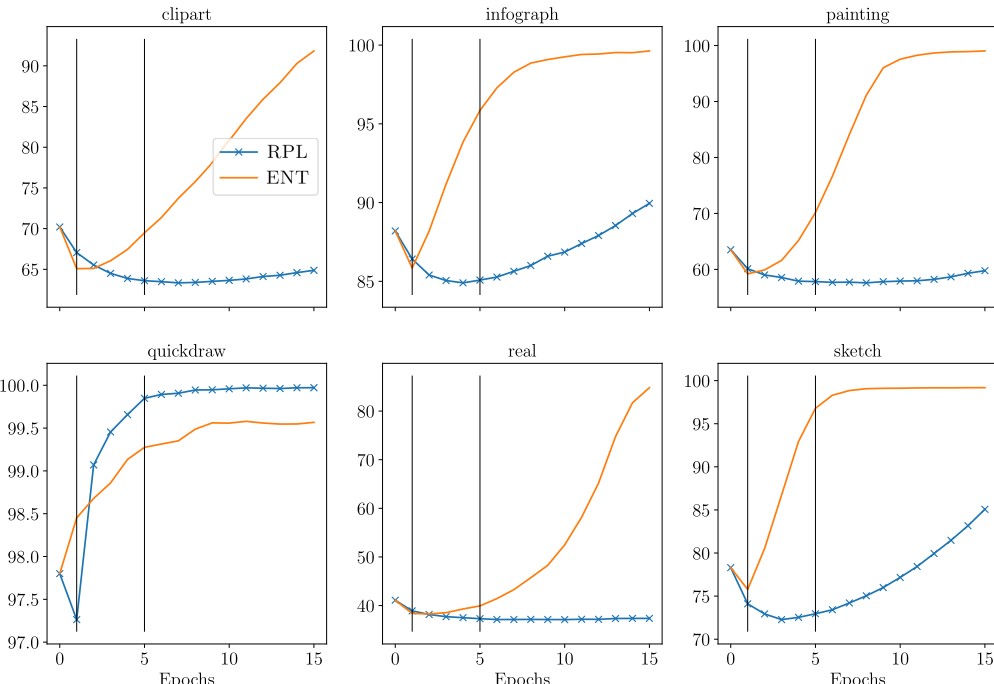

Figure 8: Top-1 error for the different IN-D domains for a ResNet50 and training with $RPL_{q=0.8}$ and ENT. We indicate the epochs at which we extract the test errors by the dashed black lines (epoch 1 for ENT and epoch 5 for $RPL_{q=0.8}$).

### D.4 Detailed results for the EfficientNet-L2 Noisy Student model on IN-D

We show the detailed results for the EfficientNet-L2 Noisy Student model on IN-D in Table 38.

Table 38: Top-1 error ($\searrow$) on IN-D in % for EfficientNet-L2

| Domain | Baseline | ENT | RPL |
|---|---|---|---|
| Clipart | 45.0 | 39.8 | **37.9** |
| Infograph | **77.9** | 91.3 | 94.3 |
| Painting | 42.7 | 41.7 | **40.9** |
| Quickdraw | **98.4** | 99.4 | 99.4 |
| Real | 29.2 | 28.7 | **27.9** |
| Sketch | 56.4 | **48.0** | 51.5 |
| mDE | 67.2 | **66.8** | 67.2 |

### D.5 Detailed results on the error analysis on IN-D

**Analysing frequently predicted classes** We analyze the most frequently predicted classes on IN-D by a vanilla ResNet50 and show the results in Fig. 9. The colors of the bars indicate whether the predicted class is part of the IN-D dataset: "blue" indicates that the class appear in the IN-D dataset, while "orange" means that the class is not present in IN-D.

We make several interesting observations: First, we find most errors interpretable: it makes sense that a ResNet50 assigns the label "comic book" to images from the "Clipart" or "Painting" domains, or "website" to images from the "Infograph" domain, or "envelope" to images from the "Sketch" domain. Second, on the hard domain "Quickdraw", the ResNet50 mostly predicts non-sensical classes that are not in IN-D, mirroring its almost chance performance on this domain. Third, we find no systematic errors on the "Real" domain which is expected since this domain should be similar to IN.

**Analyzing the correlation between the performance on IN-C/IN-R and IN-D** We show the Spearman's rank correlation coefficients for errors on ImageNet-D correlated to errors on ImageNet-R and ImageNet-C for robust ResNet50 models in Fig. 9. For this correlation analysis, we take the error numbers from Table 33. We find the correlation to be high between most domains in IN-D and IN-R which is expected since the distribution shift between IN-R and IN is similar to the distribution shift between IN-D and ImageNet. The only domain where the Spearman's rank correlation coefficient is higher for IN-C is the "Real" domain which can be explained with IN-C being closer to real-world data than IN-R. Thus, we find that the Spearman's rank correlation coefficient reflects the similarity between different datasets.

**Filtering predictions on IN-D that cannot be mapped to ImageNet** We perform a second analysis: We filter the predicted labels according to whether they can be mapped to IN-D and report the filtered top-1 errors as well as the percentage of filtered out inputs in Table 39. We note that for the domains "infograph" and "quickdraw", the ResNet50 predicts labels that cannot be mapped to IN-D in over 70% of all cases, highlighting the hardness of these two domains.

**Filtering labels and predictions on IN that cannot be mapped to ImageNet-D** To test for possible class-bias effects, we test the performance of a ResNet50 model on ImageNet classes that can be mapped to IN-D and report the results in Table 39.

First, we map IN labels to IN-D to make the setting as similar as possible to our experiments on IN-D and report the top-1 error (12.1%). This error is significantly lower compared to the top-1 error a ResNet50 obtains following the standard evaluation protocol (23.9%). This can be explained by the simplification of the task: While in IN there are 39 bird classes, these are all mapped to the same hierarchical class in IN-D. Therefore, the classes in IN-D are more dissimilar from each other than in IN. Additionally, there are only 164 IN-D classes compared to the 1000 IN classes, raising the chance level prediction.

Table 39: top-1 error on IN and different IN-D domains for different settings: left column: predicted labels that cannot be mapped to IN-D are filtered out, right column: percentage of filtered out labels.

| Dataset | top-1 error on filtered labels in % | percentage of rejected inputs |
|---|---|---|
| IN val | 13.4 | 52.7 |
| IN-D real | 17.2 | 27.6 |
| IN-D clipart | 59.0 | 59.0 |
| IN-D infograph | 59.3 | 74.6 |
| IN-D painting | 39.5 | 42.4 |
| IN-D quickdraw | 96.7 | 76.1 |
| IN-D sketch | 65.6 | 47.9 |

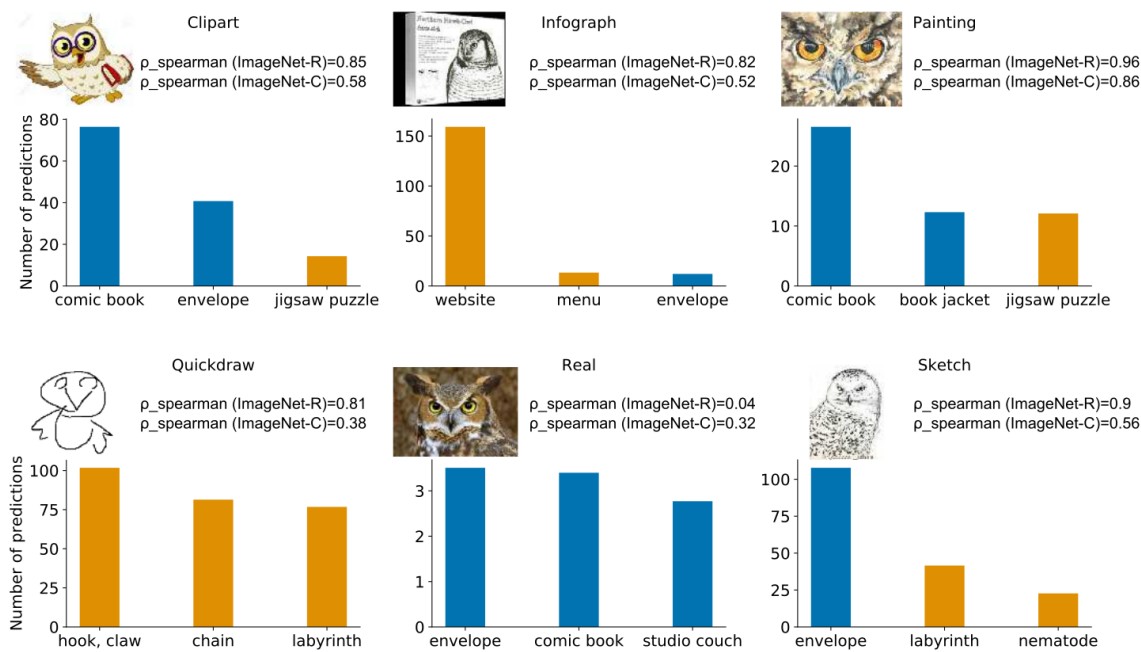

Figure 9: Systematic predictions of a vanilla ResNet50 on IN-D for different domains. The colors of the bars indicate whether the predicted class is part of the IN-D dataset: "blue" indicates that the class appear in the IN-D dataset, while "orange" means that the class is not present in IN-D. The errors on IN-R are strongly correlated to errors on IN-D for most domains, except for the "Real" domain.

If we further only accept predictions that can be mapped to IN-D, the top-1 error is slightly increased to 13.4%. In total, about 52.7% of all images in the IN validation set cannot be mapped to IN-D.

### D.6  Top-1 error on IN-D for AlexNet

We report the top-1 error numbers on different IN-D as achieved by AlexNet in Table 40. We used these numbers for normalization when calculating mDE.

Table 40: top-1 error on IN-D by AlexNet which was used for normalization.

| Dataset | top-1 error in % |
|---|---|
| IN-D real | 54.887 |
| IN-D clipart | 84.010 |
| IN-D infograph | 95.072 |
| IN-D painting | 79.080 |
| IN-D quickdraw | 99.745 |
| IN-D sketch | 91.189 |

# E  Additional experiments

## E.1  Beyond ImageNet classes: Self-learning on WILDS

The WILDS benchmark (Koh et al., 2021) is comprised of ten tasks to test domain generalization, subpopulation shift, and combinations thereof. In contrast to the setting considered here, many of the datasets in WILDS mix several 10s or 100s domains during test time.

The Camelyon17 dataset in WILDS contains histopathological images, with the labels being binary indicators of whether the central $32{\times}32$ region contains any tumor tissue; the domain identifies the hospital that the patch was taken from. Camelyon17 contains three different test splits with different domains and varying difficulty levels. For evaluation, we took the pretrained checkpoint from worksheets.codalab.org/worksheets/ 0x00d14c55993548a1823a710642f6d608 (camelyon17_erm_densenet121_seed0) for a DenseNet121 model (Huang et al., 2017) and verified the reported baseline performance numbers. We adapt the models using ENT or RPL for a maximum of 10 epochs using learning rates $\{3 \times 10^{-5}, 3 \times 10^{-4}, \ldots 3 \times 10^{-1}\}$. The best hyperparameter is selected according to OOD Validation accuracy.

The RxRx1 dataset in WILDS contains RGB images of cells obtained by fluorescent microscopy, with the labels indicating which of the 1,139 genetic treatments (including no treatment) the cells received; the domain identifies the batch in which the imaging experiment was run. The RxRx1 dataset contains three test splits, however, unlike Camelyon17, in all of the splits the domains are mixed. For evaluation, we took the pretrained checkpoint from worksheets.codalab.org/bundles/0x7d33860545b64acca5047396d42c0ea0 for a ResNet50 model and verified the reported baseline performance numbers. We adapt the models using ENT or RPL for a maximum of 10 epochs using base learning rates $\{6.25 \times 10^{-6}, 6.25 \times 10^{-5}, \ldots 6.25 \times 10^{-2}\}$, which are scaled to the admissible batch size for single GPU adaptation using linear scaling. The best hyperparameter is selected according to OOD Validation accuracy.

Table 41: Self-learning can improve performance on WILDS if a systematic shift is present—on Camelyon17, the ood validation and test sets are different hospitals, for example. On datasets like RxRx1 and FMoW, we do not see an improvement, most likely because the ood domains are shuffled, and a limited amount of images exist for each test domain.

| | Top-1 accuracy [%] | | |
| | Validation (ID) | Validation (OOD) | Test (OOD) |
| --- | --- | --- | --- |
| Camelyon17 | | | |
| Baseline | 81.4 | 88.7 | 63.1 |
| BN adapt | 97.8 (+16.4) | 90.9 (+2.2) | 88.0 (+24.9) |
| ENT | 97.6 (+16.2) | 92.7 (+4.0) | 91.6 (+28.5) |
| RPL | 97.6 (+16.2) | 93.0 (+4.3) | 91.0 (+27.9) |
| RxRx1 | | | |
| Baseline | 35.9 | 19.1 | 29.7 |
| BN adapt | 35.0 (-0.9) | 19.1 (0.0) | 29.4 (-0.3) |
| ENT | 34.8 (-1.1) | 19.2 (+0.1) | 29.4 (-0.3) |
| RPL | 34.8 (-1.1) | 19.2 (+0.1) | 29.4 (-0.3) |
| FMoW | | | |
| Baseline | 60.5 | 59.2 | 52.9 |
| BN adapt | 59.9 (-0.6) | 57.6 (-1.6) | 51.8 (-1.1) |
| ENT | 59.9 (-0.6) | 58.5 (-0.7) | 52.2 (-0.7) |
| RPL | 59.8 (-0.7) | 58.6 (-0.6) | 52.1 (-0.8) |

The FMoW dataset in WILDS contains RGB satellite images, with the labels being one of 62 building or land use categories; the domain specifies the year in which the image was taken and its geographical region (Africa, the Americas, Oceania, Asia, or Europe). The FMoW dataset contains four test splits for different time periods, for which all regions are mixed together. For evaluation, we took the pretrained checkpoint from //worksheets.codalab.org/bundles/0x20182ee424504e4a916fe88c91afd5a2 for a DenseNet121 model and verified the reported baseline performance numbers. We adapt the models using ENT or RPL for a maximum

of 10 epochs using learning rates $\{5.0 \times 10^{-6}, 5.0 \times 10^{-5}, \ldots 5.0 \times 10^{-2}\}$. The best hyperparameter is selected according to OOD Validation accuracy.

While we see improvements on Camelyon17, neither BN adaptation nor self-learning can improve performance on RxRx1 or FMoW. Initial experiments on PovertyMap and iWildsCam also do not show improvements with self-learning. We hypothesize that the reason lies in the mixing of the domains: Both BN adaptation and our self-learning methods work best on systematic domain shifts. These results support our claim that self-learning is effective, while showing the important limitation when applied to more diverse shifts.

### E.2 Small improvements on BigTransfer models with Group normalization layers

We evaluated BigTransfer models (Kolesnikov et al., 2020) provided by the timm library (Wightman, 2019). A difference to the ResNet50, ResNeXt101 and EfficientNet models is the use of group normalization layers, which might influence the optimal method for adaptation—for this evaluation, we followed our typical protocol as performed on ResNet50 models, and used affine adaptation.

For affine adaptation, a distilled BigTransfer ResNet50 model improves from 49.6 % to 48.4 % mCE on the ImageNet-C development set, and from 55.0 % to 54.4 % mCE on the ImageNet-C test set when using RPL ($q = 0.8$) for adaptation, at learning rate $7.5 \times 10^{-4}$ at batch size 96 after a single adaptation epoch. Entropy minimization did not further improve results on the ImageNet-C test set. An ablation over learning rates and epochs on the dev set is shown in Table 42, the final results are summarized in Table 43.

Table 42: mCE in % on the IN-C dev set for ENT and RPL for different numbers of training epochs when adapting the affine batch norm parameters of a BigTransfer ResNet50 model.

| criterion | ENT | | | RPL | | |
|---|---|---|---|---|---|---|
| lr, 7.5 × epoch | $10^{-5}$ | $10^{-4}$ | $10^{-3}$ | $10^{-5}$ | $10^{-4}$ | $10^{-3}$ |
| 0 | 49.63 | 49.63 | 49.63 | 49.63 | 49.63 | 49.63 |
| 1 | 49.44 | 50.42 | 52.59 | 49.54 | 48.89 | 48.95 |
| 2 | 49.26 | 50.27 | 56.47 | 49.47 | **48.35** | 50.77 |
| 3 | 49.08 | 52.18 | 60.06 | 49.39 | 48.93 | 51.45 |
| 4 | 48.91 | 52.03 | 60.50 | 49.31 | 50.01 | 51.53 |
| 5 | **48.80** | 51.97 | 62.91 | 49.24 | 49.96 | 51.34 |
| 6 | 48.83 | 52.10 | 62.96 | 49.16 | 49.71 | 51.19 |
| 7 | 48.83 | 52.10 | 62.96 | 49.16 | 49.71 | 51.19 |

Table 43: mCE in % on the IN-C dev set for ENT and RPL for different numbers of training epochs when adapting the affine batch norm parameters of a BigTransfer ResNet50 model.

| | dev mCE | test mCE |
|---|---|---|
| Baseline | 49.63 | 55.03 |
| ENT | 48.80 | 56.36 |
| RPL | **48.35** | **54.41** |

### E.3 Can Self-Learning improve over Self-Learning based UDA?

An interesting question is whether test-time adaptation with self-learning can improve upon self-learning based UDA methods. To investigate this question, we build upon French et al. (2018) and their released code base at github.com/Britefury/self-ensemble-visual-domain-adapt. We trained the Baseline models from scratch using the provided shell scripts with the default hyperparameters and verified the reported performance. For adaptation, we tested BN adaptation, ENT, RPL, as well as continuing to train in exactly the setup of French et al. (2018), but without the supervised loss. For the different losses, we adapt the models for a maximum of 10 epochs using learning rates $\{1 \times 10^{-5}, 1 \times 10^{-4}, \ldots, 1 \times 10^{-1}\}$.

Note that for this experiment, in contrast to any other result in this paper, **we purposefully do not perform proper hyperparameter selection based on a validation dataset**—instead we report the best accuracy across all tested epochs and learning rates to give an upper bound on the achievable performance for test-time adaptation.

As highlighted in Table 44, none of the four tested variants is able to meaningfully improve over the baseline, corroborating our initial hypothesis that self-learning within a full UDA setting is the optimal strategy, if dataset size and compute permits. On the other hand, results like the teacher refinement step in DIRT-T

(Shu et al., 2018) show that with additional modifications in the loss function, it might be possible to improve over standard UDA with additional adaptation at test time.

Table 44: Test-time adaptation marginally improves over self-ensembling.

| | Baseline | BN adapt | ENT | RPL | Self-ensembling loss |
|---|---|---|---|---|---|
| MNIST→SVHN | | | | | |
| MT+TF | 33.88 | 34.44 | 34.87 | 35.09 | 33.27 |
| MT+CT* | 32.62 | 34.11 | 34.25 | 34.21 | 33.36 |
| MT+CT+TF | 41.59 | 41.93 | 41.95 | 41.95 | 42.70 |
| MT+CT+TFA | 30.55 | 32.53 | 32.54 | 32.55 | 30.84 |
| SVHN-specific aug. | 97.05 | 96.82 | 96.91 | 96.87 | 97.12 |
| MNIST→USPS | | | | | |
| MT+TF | 98.01 | 97.91 | 97.96 | 97.91 | 98.16 |
| MT+CT* | 88.34 | 88.39 | 88.54 | 88.39 | 88.44 |
| MT+CT+TF | 98.36 | 98.41 | 98.41 | 98.41 | 98.50 |
| MT+CT+TFA | 98.45 | 98.45 | 98.45 | 98.45 | 98.61 |
| SVHN→MNIST | | | | | |
| MT+TF | 98.49 | 98.47 | 98.49 | 98.47 | 99.40 |
| MT+CT* | 88.34 | 88.36 | 88.36 | 88.36 | 89.36 |
| MT+CT+TF | 99.51 | 99.49 | 99.5 | 99.49 | 99.57 |
| MT+CT+TFA | 99.56 | 99.57 | 99.57 | 99.57 | 99.58 |
| SVHN-specific aug. | 99.52 | 99.49 | 99.5 | 99.49 | 99.65 |
| USPS→MNIST | | | | | |
| MT+TF | 92.79 | 92.62 | 92.62 | 92.66 | 93.08 |
| MT+CT* | 99.11 | 99.13 | 99.14 | 99.13 | 99.21 |
| MT+CT+TF | 99.41 | 99.42 | 99.45 | 99.42 | 99.52 |
| MT+CT+TFA | 99.48 | 99.54 | 99.57 | 99.54 | 99.54 |

# F    Software stack

We use different open source software packages for our experiments, most notably Docker (Merkel, 2014), scipy and numpy (Virtanen et al., 2020), GNU parallel (Tange, 2011), Tensorflow (Abadi et al., 2016), PyTorch (Paszke et al., 2017), timm (Wightman, 2019), Self-ensembling for visual domain adaptation (French et al., 2018), the WILDS benchmark (Koh et al., 2021), and torchvision (Marcel & Rodriguez, 2010).

