# OpenReview forum: "If your data distribution shifts, use self-learning"
_TMLR — Accepted by TMLR_

### Review · Reviewer_qrwv · 2022-08-21

**Summary Of Contributions:**

The authors performed a thorough evaluation of several self-learning techniques such as entropy minimization and pseudo-labeling for source-free domain adaptation performance on a range of datasets (ImageNet-C/R, CIFAR10/STL10, etc) with different network architecture choices (ResNet, DenseNet, WRN, etc). From the experimental results, they show that self-learning methods universally improves domain adaptation performance regardless of model choices and dataset size. They also provided abundant ablation studies on the properties of self-learning and a stability study of the technique. Finally, the authors suggest best practices of self-learning for real deployment.

**Broader Impact Concerns:**

No ethical concerns observed.

**Requested Changes:**

- Include medical data and test self-learning across hospitals;
- Perform repeated experiments especially for results that are indistinguishable;
- Show both  robust pseudo-labeling and entropy minimization results in Table 1 & 2;
- Compare self-learning results with invariant learning techniques;


**Strengths And Weaknesses:**

Strengths:
- This empirical study is well motivated that a thorough guideline/evaluation for the recently developed self-learning and test-time adaptation methods is desired;
- The paper is clear and overall well written;
- The paper included a wide range of domain shift datasets and network architectures;
- The authors performed rich ablation studies on the characteristics of the self-learning methods;

Weaknesses:
- The hospital example for the necessity of source-free domain adaptation in the introduction section is a great motivation for this study. However, there’s no medical data has been evaluated in the experiments section;
- Only single-run results have been reported in the paper. The comparison between baselines might be subject to randomness. Perhaps repeated experiments are necessary ;
- In Table 1 & 2, the authors only show robust pseudo-labeling or entropy minimization -based adaptation results. Perhaps showing results in both tables can further gives us insights of the performance of these two techniques on different scales of datasets;
- In Table 2, there’s only marginal improvements of the entropy minimization adaptation compared to the DANN baseline (29.7 -> 28.5; 25.0 -> 23.9). Does this suggest that the performance of self-learning is relatively upper bound by adversarial adaptation approaches?
- The authors only compare self-learning methods with adversarial adaptation and test-time adaptation baselines but not domain-invariant approaches such as invariant minimization risk (IRM);

---

> ### Author Response · Authors · 2022-09-16
> **Authors’ response to Reviewer qrwv (1/2)**
>
> Thanks a lot for your review and suggestions. We addressed three of your questions directly with a paper update and comment below.
> All changes in response to your suggestions are highlighted in **blue** in the manuscript, except for changes in the Tables due to formatting.
>
> Thanks for your excellent suggestions for improving the paper, and please let us know if your concerns are adequately addressed by our response.
>
> ###  “Include medical data and test self-learning across hospitals”
>
> We agree that making this connection more explicit is very worthwhile.
>
> In the current version of the paper, we already tested several medical image datasets included in the first version of WILDS. We show the results below, and in Table 35 in Appendix E.1. “Beyond ImageNet classes: Self-learning on WILDS”:
>
> | Top-1 accuracy [%]  | Validation (ID) | Validation (OOD) | Test (OOD)  |
> |--|--|--|--|
> |Camelyon17 |  | | |
> |Baseline| 81.4| 88.7| 63.1|
> |BN adapt| 97.8 (+16.4)| 90.9 (+2.2) |88.0 (+24.9)|
> |ENT| 97.6 (+16.2)| 92.7 (+4.0) |91.6 (+28.5)|
> |RPL |97.6 (+16.2) |93.0 (+4.3) |91.0 (+27.9)|
>
> Self-learning is very effective on the CAMELYON domain of WILDS (which is about whole-slide images of histological lymph node sections) which has **systematic** distribution shifts with different hospitals in train, test and validation. In contrast, other WILDS domains such as RxRx1 or FMoW mix datasets with different distribution shifts and thus, neither self-learning nor simple BN adaptation show improvements. This is expected since both BN adaptation and our self-learning methods work best on systematic domain shifts (for the case of BN adaptation, Schneider et al. (2020) tested this claim on a “mixed ImageNet-C” variant). These results support our claim that self-learning is effective, while showing the important limitation when applied to more diverse shifts.
>
> The WILDS results are currently in the Appendix due to the page limit. We will point to it more prominently and discuss the WILDS results in the main part in the following way:
>
> > “We discuss additional proof-of-concept implementations on the WILDS benchmark (Koh et al., 2021), BigTransfer (BiT; Chen et al., 2020) models and on self-learning based UDA models in Appendix E. On WILDS, self-learning is effective for the Camelyon17 task with a systematic shift between train, validation and test sets (each set is comprised of different hospitals), while self-learning fails to improve on tasks with mixed domains, such as on the RxRx1 and the FMoW tasks.These results support our claim that self-learning is effective, while showing the important limitation when applied to more diverse shifts.”
>
> Does this adequately address your suggestion to include self-learning on hospital data?
>
> ### “Perform repeated experiments especially for results that are indistinguishable”
>
> Thank you for this suggestion. We already included the unbiased std over three runs for ENT and RPL on IN-C dev and test in Table 24, Appendix C.8. “Performance over different seeds in a ResNet50 on ImageNet-C”:
>
> |ResNet50 + self-learning | mCE on IN-C dev [%]  | mCE on IN-C test [%] |
> |--|--|--|
> |ENT| 50.0 (0.04)| 51.6 (0.04)|
> |RPL |48.9 (0.02) |50.5 (0.03)|
>
> Thus, our self-learning results are very stable and the variance across seeds is two orders lower than any improvements which we demonstrate. This is expected: Compared to e.g. a result on the ImageNet validation set, our ImageNet-C results are already averaged across the 75 validation datasets (3.75m images vs. 50k in standard ImageNet evaluation). To limit the amount of compute, could you please point to specific results you would like to see results of multiple seeds for, if any?
>
> To stress the stability of self-learning, we added the following sentences to the main part: “The improvements of self-learning are very stable to random initialization and order of the samples during adaptation: In Table 24, Appendix C.8, we show the averaged results across three different seeds for a ResNet50 model adapted with ENT and RPL. The unbiased standard deviation is one to two orders of magnitude lower than any improvement we report in our paper, showcasing the significance of our results.”
>
> ### “Show both robust pseudo-labeling and entropy minimization results in Table 1 & 2”
>
> Thank you for the suggestion, we updated Tables 1+2 to show both ENT and RPL. Please refer to the revised manuscript.
>
> ### “In Table 2, there’s only marginal improvements of the entropy minimization adaptation compared to the DANN baseline. Does this suggest that the performance of self-learning is relatively upper bound by adversarial adaptation approaches?”
>
> To test this hypothesis, we looked at another shift between MNIST and MNIST-M (Table 2), where DANN achieved an error of 11.4% which was improved to an error of 5.1% by ENT. Hence, ENT more than halved the error on this dataset. For this reason, we do not think that DANN constitutes an upper bound on the performance achievable by self-learning.

---

> ### Author Response · Authors · 2022-09-16
> **Authors’ response to Reviewer qrwv (2/2)**
>
> ### “Compare self-learning results with invariant learning techniques”
>
> We did not consider invariant learning techniques because they generally do not outperform standard ERM in well-controlled experiments [1] and we already show that adapting ERM pre-trained models with self-learning meaningfully improves ERM performance.
>
> Furthermore, we do not see IRM as a competing approach to self-learning. Instead, what we try to show in our work is that self-learning is *complementary* to various pre-training techniques (incl. data augmentation, domain adaptation, self-supervised pretraining). Hence, even if a setting existed where IRM outperformed ERM, the question within the scope of our paper would be whether further adaptation of this IRM-pre-trained checkpoint would be  possible via self-learning.
>
> Given that IRM-like techniques are not yet popular on ImageNet scale/for model deployment, we did not include such a study in the paper. We are happy to discuss further if you would find such an effort worthwhile.
>
> [1] Gulrajani & Lopez-Paz, “In Search of Lost Domain Generalization”, ICLR 2020

---

### Review · Reviewer_4Aye · 2022-09-05

**Summary Of Contributions:**

The paper studies the problem of distribution discrepancy between training and testing. The technique of interest is self-learning such as entropy minimization and pseudo labeling. The paper's purpose is to demonstrate the effectiveness of using self-learning on the testing examples, which have no label information. The experiments show the promise of self-learning, which significantly error reduction on multiple datasets.

**Broader Impact Concerns:**

The proposed approach could introduce bias from the test data set. People usually execute actions to avoid bias during training, however, there is no known policy to restrict bias in the testing data. The model adapted to the biased testing data will lead to fairness issues if used subsequently on broader applications.

**Requested Changes:**

1. Improve the discussion on related work
2. Provide necessary comparison with related approaches
3. Provide a discussion on the model calibration of the unadapted and adapted models
4. Provide a discussion on the forgetting effect of the training distribution.

**Strengths And Weaknesses:**

# Strength
1. The paper addresses an important problem in machine learning, i.e., the distribution shift between training and testing.
2. The paper has a clear argument and successfully proved it, i.e., using self-learning for test-time adaptation.
3. The paper is well-written and easy to follow.
4. The experimental validation is sufficient to validate the core argument.

# Weakness
1. The paper seems to rebrand unsupervised learning or self-distillation as "self-learning". I am not sure if this is appropriate.
2. The idea of self-supervised learning for test-time adaptation does not seem novel. It is worthwhile to provide sufficient literature review on this topic. Some related work to consider (some do not seem cited by the paper):
    * Azimi et al. Self-supervised Test-time Adaptation on Video Data. 2021
    * Chen et al. Contrastive Test-Time Adaptation. 2022
    * Bartler et al. MT3: Meta Test-Time Training for Self-Supervised Test-Time Adaption. 2022
    * Sun et al. Test-Time Training with Self-Supervision for Generalization under Distribution Shifts. 2020
3. Experiments should compare with some related work
4. The calibration of the adapted model is not studied
5. The forgetting effect of the training distribution is not studied

---

> ### Author Response · Authors · 2022-09-23
> **Authors’ response to Reviewer 4Aye (1/4)**
>
> Thanks a lot for your review and suggestions. We addressed your questions directly with a paper update and comments below. All changes in the paper reflecting the implementation of your suggestions are shown in **green** to be found more easily, except for changes in the Tables due to formatting.
>
> Thanks for your excellent suggestions for improving the paper, and please let us know if your concerns are adequately addressed by our response.
>
> ### The paper seems to rebrand unsupervised learning or self-distillation as "self-learning"
>
> Thanks for raising this, this was not our intention. We were looking for an umbrella term including techniques like pseudo-labeling, entropy minimization, and also self-supervised learning. In this search for a good term, self-learning seemed suitable and has a longer history than other terms: Tsypkin (1968) defines self-learning as “learning when there is no external indication concerning the correctness of the response of the automatic system to the presented patterns” [1]. The term “self-learning” has then been popularized in the late 1980s in a number of works [2,3,4].
>
> However, it is not our intention to rebrand existing work. If you are aware of another umbrella term, we are happy to adapt the paper in this regard.
>
> To improve clarity, we added the following footnote to Section 1:
> > “We define self-learning as the superset of different pseudo-labeling variants and entropy minimization to highlight the fact these methods do not require ground truth labels for adaptation.”
>
> We also added the following sentence to Section 3:
> > ”Tsypkin defines self-learning as “learning when there is no external indication concerning the correctness of the response of the automatic system to the presented patterns”', and thus, we use this term as a superset of different variants of pseudo-labeling and entropy minimization to highlight that these methods can be used for adaptation to unlabeled data.”
>
> Please let us know whether the current changes adequately addressed your concerns about us using the word “self-learning” in this context.
>
> References:
> [1] Ya Tsypkin “Self-Learning-What Is It?”, 1968
> [2] Xu et al. “Fuzzy model identification and self-learning for dynamic systems” IEEE Transactions on Systems, Man, and Cybernetics 1987
> [3] Nguyen et al. “Neural networks for self-learning control systems” IEEE Control Systems Magazine 1990
> [4] Widrow et al. “The truck backer-upper: An example of self-learning in neural networks'' Advanced Neural Computers 1990
>
> ### Please add calibration results.
>
> Thank you for this suggestion. Eastwood et al. [1] show that approaches based on entropy minimization and pseudo-labeling hurt calibration due to the objective of confidence maximization. We are working on producing these results and will post them later during the discussion period, and add them to the revised manuscript. We expect both ENT and RPL to lead to decreased calibration compared to the unadapted models, and think that reporting these results is very important for completeness, given our extensive analysis of self-learning.
>
> [1] Eastwood et al. “Source-Free Adaptation to Measurement Shift via Bottom-Up Feature Restoration”, ICLR 2022
>
> ### Please add a discussion on the forgetting effect of the training distribution.
>
> We are working on producing these results and will post them later during the discussion period.

---

> ### Author Response · Authors · 2022-09-23
> **Authors’ response to Reviewer 4Aye (2/4)**
>
> ### Concerns regarding technical novelty.
>
> We would like to highlight that to the best of our knowledge, this paper is the first to show that a broad range of self-learning techniques (ENT, RPL, PL) work across a range of model architectures (ConvNets, ViT), sizes (RN50 to EffNet-L2), and are orthogonal to robustification (e.g., DeepAugment) and other pre-training schemes. We are not aware of any published works making explicit statements along any of these dimensions.
>
> We also aimed to improve the rigor in empirical evaluation compared to past works with our experimental protocol and a range of ablation studies that will have value for practitioners when applying self-learning---and indeed, **due**  to this experimental protocol, observe behaviors not evident from previous work. A variety of our insights go well beyond what has been known in the literature. To name just a few examples: (1) we show the collapse of the TENT model when training at the proposed learning rate for more than one epoch, cf. Table 13, which is an important finding for practitioners, (2) we use vision transformers and obtain very different results compared to ConvNets: while in ConvNets, the adaptation of affine BN parameters works best, in ViTs, full model adaptation performs best, (3) we show the sensitivity of entropy minimization to the temperature parameter, which we theoretically study and empirically validate, (4) we demonstrate that self-learning works across model sizes, architectures and datasets, and (5) we show that self-learning is orthogonal to various pre-training and UDA methods. We think that none of these statements are trivial without thorough and comprehensive experimental investigation.
>
> We want to again stress that our method provides new state-of-the art results across a range of tasks (e.g., EffNet-L2 on IN-C (top1 acc.): 82.9% [ours] vs 77.3% [before]), with well-controlled baselines, and careful hyperparameter selection. Indeed, we show that our simple methods outperform much more sophisticated approaches.
>
> Beyond empirical observations, our theory on the employed loss functions aligns with both our empirical findings and the hyperparameters selected in other empirical works (e.g., the selection of the temperature parameter in  DINO, TENT), and adds to the understanding of self-learning techniques.
>
> Finally, we have chosen TMLR as the venue for this paper, because TMLR values rigorous and careful experiments, and sets technical novelty at a lower priority compared to experimental rigor. The criteria for acceptance in TMLR are that “claims made in the submission [are] supported by accurate, convincing and clear evidence”, and that “at least some individuals in TMLR's audience [should] be interested in knowing the findings of this paper” [1]. We made our experimental protocols as rigorous as possible in order to provide results and guidelines which are interesting and useful to practitioners when applying self-learning.
>
> [1] https://jmlr.org/tmlr/editorial-policies.html

---

> ### Author Response · Authors · 2022-09-23
> **Authors’ response to Reviewer 4Aye (3/4) [Comparison to previous work 1/2]**
>
> ### Comparisons to prior work
>
> Thank you for your suggestions to include other related works to our paper. Following your suggestions and the suggestions of Reviewer iPbT, we have significantly expanded our related work section as well as Table 4 to compare to more test-time adaptation techniques.
>
> We included the papers you have suggested [3,4,5,6] as related work in our revised paper version, and added [6] as another baseline. Please see our discussion on the suggested papers below, and our revised version of the paper.
>
> #### Discussion on **Azimi et al. [4]**
>
> **Azimi et al. [4]** use the test-time adaptation techniques Batch Norm adaptation, TENT and Test Time Training on video data. We have already included all three techniques as a comparison in Table 4 of our main paper and show that 1) our properly tuned entropy minimization baseline outperforms TENT, and 2) our RPL baseline improves upon all three baselines. We have changed the wording in Tables 3+4 to say “RPL” instead of “self-learning” for clarity.
>
> We added the work by **Azimi et al.** to our related work section in the following way:
>
> > “Azimi et al. (2022) show performance improvements when using test-time adaptation on video data. They show adaptation results for the popular test-time adaptation techniques of BN adaptation (Schneider et al., 2020), Test-Time Training (Sun et al., 2020) and test-time entropy minimization (Wang et al., 2021).”
>
> #### Discussion on **Sun et al. [3]**
>
> As mentioned above, we compare to **Sun et al. [3]** in Table 4 of our main paper. In addition, we have dedicated a separate Appendix section to compare our approach with TTT, please see Appendix “C.6 Self-training outperforms contrastive Test-Time Training”. We found that even simple hard-labeling without hyperparameter tuning outperformed TTT, and further gains were possible with RPL. To improve clarity, we expanded our related work section to clearly state the baselines we compare to.
>
> #### Discussion on **Chen et al. [6]**
>
> AdaContrast by **Chen et al. [6]** combines pseudo-labeling with other techniques, such as self-supervised contrastive learning on the target domain, soft k-nearest neighbors voting to stabilize the pseudo-labels, as well as consistency and diversity regularization. A direct comparison with their results is difficult because they evaluate on  VISDA-C and DomainNet, and so we would need to train our methods on either of the datasets and perform a full hyperparameter search for a fair comparison. We do have one point of comparison: in their paper, Chen et al. [6] perform better than TENT [2]. However, we note that Chen et al. used the default hyperparameters of TENT and did not perform hyperparameter tuning on the new dataset, thus, we would expect the performance of properly tuned TENT to be better than reported in the paper. We expect the additional changes of AdaContrast to further improve upon our simple self-learning baselines. The focus of our paper was to study pure self-learning and understand under which conditions it performs best.
>
> We have added this discussion on AdaContrast to the related work section of our paper.

---

> ### Author Response · Authors · 2022-09-23
> **Authors’ response to Reviewer 4Aye (4/4) [Comparison to previous work 2/2]**
>
> ### Comparisons to prior work [continued]
>
> #### Discussion on **Bartler et al. [5]**
>
> **Bartler et al. [5]** report an error of 24.4% as their top result on CIFAR10-C which is far worse than our top results of 8.5% with an AugMix trained model, or 13.3% with a vanilla trained model. A part of this performance difference can be attributed to their use of a WRN-26-1 model while we report results with a WRN-40-2. Another important distinction between their model and ours is that they used GroupNorm (GN) layers while we have only looked at models with BatchNorm (BN) layers. Thus, to make the comparison more fair, we tested our approach on their model architecture.
>
> A direct comparison is not straight-forward since Bartler et al. trained their models using Keras while our code-base is in PyTorch. Therefore, we first trained a baseline model in PyTorch using their architecture with the standard and widely used CIFAR10 training code available at https://github.com/kuangliu/pytorch-cifar (1.9k forks, 4.8k stars); the baseline test accuracy on clean CIFAR10 using the architecture of Bartler et al. is at 94.3%. As a second step, we adapted the baseline model with ENT and RPL on CIFAR10-C. Please see the Table below for our results. BN adaptation is not possible for the model used by Bartler et al. since they use GN. Due to the usage of GN, it is not evident whether the affine GN parameters should be adapted, or all model parameters. Thus, we report results for both adaptation mechanisms.
>
> We find that adaptation of affine GN layers works better than full model adaptation, consistent with our results for the adaptation of affine BN layers. As the gains due to ENT and RPL seem lower than for our WRN-40-2 architecture, we hypothesize that the issue lies in the GN layers. Therefore, we trained another WRN-26-1 model with BN layers instead of GN, and adapted this model using BN, ENT and RPL. The baseline accuracy on clean CIFAR10 of WRN-26-1-BN is 95.04%. Indeed, we find that adapting the model with BN instead of GN layers leads to much larger gains, which is consistent with findings by Schneider et al. [1] who found that 1) models with GN are generally more robust to distribution shifts compared to models with BN, but 2) BN  adaptation still outperforms non-adapted models trained with GN. Thus, we conclude that self-learning techniques work better with models which have BN layers and less well with models with GN layers. For both model types (with GN or with BN layers), ENT works better than GCE which is consistent with our other results on small-scale datasets.
>
> Overall, our best result for the model architecture used by Bartler (after replacing GN layers with BN layers) et al. is **13.1%** which is much lower than their best result of 24.4%. Even when using GN layers, our best top-1 error is 18.0% which is significantly lower than the best result of Bartler et al.
>
> |Model| top-1 error on CIFAR10-C |
> |--|--|
> | WRN-26-1-GN vanilla|       |
> | Baseline  MT3 | 34.6  |
> | Adapted with MT3  | 24.4  |
> |  ||
> | WRN-26-1-GN vanilla|       |
> | Baseline [ours]  | 18.6  |
> | ENT (GN layers)| 18.0|
> | GCE (GN layers)  | 18.4  |
> | ENT (full model)  | 20.3  |
> | ENT (full model)| 19.6  |
> |  |    |
> | WRN-26-1-BN vanilla   | |
> | Baseline| 25.8 |
> | BN adapt| 18.7 |
> | ENT (BN layers)| **13.1** |
> | GCE (BN layers)|   15.1 |
>
> We added the work by **Bartler et al.** to our paper as following:
> We discuss the work by Bartler et al in our introduction and our related work sections.
> We added our results with their model architecture to our Table 2, and the model description to our paper section “Experiment design: Models for CIFAR10/ MNIST-scale datasets.”
> We include Bartler et al in Table 4 as a baseline we compare self-learning to.
> We added a paragraph to Section 6 discussing the benefits of adapting models with BN layers compared to models with GN layers and a new Table 10. There, we also included the results on adapted BigTransfer models which have GN layers from Appendix E.2.
> We created a new subsection in our Appendix (section C.7) which includes a full comparison to Bartler et al, as a well as a detailed discussion. We summarize the results and point to Appendix C.7. in the main part.
>
> #### References:
>
> - [1] Schneider et al. Improving robustness against common corruptions by covariate shift adaptation, 2020
> - [2] Wang et al. Tent: Fully Test-time Adaptation by Entropy Minimization, 2020
> - [3] Sun et al. Test-Time Training with Self-Supervision for Generalization under Distribution Shifts. 2020
> - [4] Azimi et al. Self-supervised Test-time Adaptation on Video Data, 2021
> - [5] Bartler et al. MT3: Meta Test-Time Training for Self-Supervised Test-Time Adaption. 2022
> - [6] Chen et al. Contrastive Test-Time Adaptation. 2022

---

> ### Author Response · Authors · 2022-09-28
> **Update: Additional results on calibration and forgetting due to self-learning**
>
> ### Please add calibration results.
>
> We have calculated the calibration results for RPL and added the following paragraph and Table to the manuscript.
>
> > “Eastwood et al. [1] show that approaches based on entropy minimization and pseudo-labeling hurt calibration due to the objective of confidence maximization. We corroborate their results and report the Expected Calibration Error (ECE; Naeini 2015) when adapting a vanilla ResNet50 model with RPL and ENT. We report the mean ECE (lower is better) and standard deviation across corruptions (and severities) below. We observe that both methods increase ECE compared to the unadapted model. The increase in ECE is higher for more severe corruptions which can be explained by a increasingly stronger distribution shift compared to the source dataset.”
>
> | adaptation| IN-C full | IN-C sev 1 | IN-C sev 2 | IN-C sev 3 | IN-C sev 4 | IN-C sev 5 |
> | --| -- | -- | -- |--|--|--|
> | w/o adapt | $2.3 \pm0.8$  | $ 2.3 \pm 0.3 $ | $ 2.1 \pm 0.3 $ | $ 2.1 \pm 0.3 $|$ 2.2 \pm 0.4 $|$ 2.9 \pm 1.6 $|
> |RPL| $ 9.6 \pm 7.1 $ | $ 6.1 \pm 0.4 $ |$ 7.1 \pm 1.2 $ | $ 8.0 \pm 1.7 $ | $ 9.8 \pm 2.7 $ |$ 17.1 \pm 12.8 $ |
> |ENT | $ 10.6 \pm 7.3 $ |$ 6.6 \pm 0.5 $ |$ 7.9 \pm 1.5 $ |$ 8.9 \pm 2.1 $ | $ 11.2 \pm 3.3 $|$ 18.7 \pm 12.6 $|
>
> [1] Eastwood et al. “Source-Free Adaptation to Measurement Shift via Bottom-Up Feature Restoration”, ICLR 2022
>
>
> ### Please evaluate the forgetting effect of self-learning on the source distribution
>
> Thank you for this suggestion. We added the following paragraph as well as a new Table to the revised manuscript.
>
> > “To judge the forgetting effect on the source distribution, we calculated the accuracy on clean ImageNet for our adapted ENT and RPL checkpoints. We note that success of self-learning can partially be attributed to correcting the Batch Normalization (BN) statistics of the vanilla model with respect to the distribution shift [2]. Thus, we only wish to examine the effect of fine-tuning of the affine BN parameters to the target distribution with respect to the source distribution. Thus, we again correct the mismatched statistics to the source dataset when calculating the accuracy.
>
> > We report the mean top1 accuracy and standard deviation across corruptions (and severities) below. We observe that both ENT and RPL lead to a decrease in performance on the source dataset, an effect which has also been observed by Niu et al. [3]. The decrease in performance is higher for more severe corruptions which can be explained by a successively stronger distribution shift compared to the source dataset. We find that the effect of forgetting is less pronounced in RPL compared to ENT.”
>
> | model | top1 accuracy on ImageNet val [%]|
> |--|--|
> |w/o adapt | 74.2 |
> |RPL adapted to IN-C (avg over corruptions and severities) | 72.7$\pm$ 2.4 |
> |RPL adapted to IN-C (avg over corruptions, sev. 1) | 74.7$\pm$ 0.6 |
> |RPL adapted to IN-C (avg over corruptions, sev. 2) | 73.9$\pm$ 0.9 |
> |RPL adapted to IN-C (avg over corruptions, sev. 3) | 73.0$\pm$ 1.3 |
> |RPL adapted to IN-C (avg over corruptions, sev. 4) | 71.8$\pm$ 1.8 |
> |RPL adapted to IN-C (avg over corruptions, sev. 5) | 69.8$\pm$ 3.0 |
> | | |
> |ENT adapted to IN-C (avg over corruptions and severities) |71.7$\pm$3.1 |
> |ENT adapted to IN-C (avg over corruptions, sev. 1) |74.3 $\pm$0.6|
> |ENT adapted to IN-C (avg over corruptions, sev. 2) | 73.2$\pm$ 1.1 |
> |ENT adapted to IN-C (avg over corruptions, sev. 3) | 72.3$\pm$ 1.7|
> |ENT adapted to IN-C (avg over corruptions, sev. 4) | 70.7$\pm$ 2.3 |
> |ENT adapted to IN-C (avg over corruptions, sev. 5) | 68.0 $\pm$ 3.9|
>
> Please let us know if we have adequately addressed your concerns.
>
> - [2]  Schneider et al. Improving robustness against common corruptions by covariate shift adaptation, 2020
> - [3] Niu et al. Efficient Test-Time Model Adaptation without Forgetting, ICML 2022

---

### Review · Reviewer_iPbT · 2022-09-14

**Summary Of Contributions:**

Self-learning, or the updating of a model from its own predictions, is an established technique for semi-supervised learning and domain adaptation.
In particular, entropy minimization and pseudo-labeling have recently been the focus of _source-free_ and _test-time_ adaptation methods, which adapt a model given only unlabeled target/test data without labeled source/train data.
This work provides a broad and informative empirical study of self-learning for adaptation across training schemes, model architectures, and various dataset shifts.
The presented study is of particular value due to (1) the scale of its models and focus on ImageNet-like data, (2) the variety of shifts including corruptions, different renditions, and adversarially filtered data, and (3) the emphasis on a clear, uniform, and rigorous experimental protocol for tuning and evaluation.
The self-learning techniques studied are pseudo-labeling (PL) and entropy minimization (ENT), as established by prior work, and a variant of pseudo-labeling that imports the generalized cross-entropy loss from prior work on robustness to label noise, which this work names Robust Pseudo-Labeling (RPL).
The new knowledge of these experiments is not in the (lack of) novelty of the losses, but in the broad and fair comparison of them in the setting of source-free adaptation.
Benchmarking underlines the finding that self-learning can achieve state of the art results for adaptation on CIFAR-10-C, IN-C, IN-R, IN-A.
This finding is worth demonstrating so thoroughly, and worth broadcasting to the community, in order to highlight how self-learning can be effective, simple, and robust to hyperparameters, as this has not necessarily been appreciated while more complex and computationally-intensive methods have been developed.
This work proposes a new adaptation dataset, IN-D, in addition to its empirical contributions measured on existing datasets.
IN-D is like the existing IN-R, but harder, which one could argue is needed as more progress is made on the existing IN variants.
Finally, a last methodological contribution is made with the emphasis placed on experimental protocol: this work tunes on a single development set (the held-out corruption types of IN-C) and then evaluates on all other shifts.
Exising papers on source-free adaptation suffer from a certain amount of diversity and hence incomparability in setup, so advocating for a single and practical setup is worthwhile.
In summary, this work extends and standardizes empirical knowledge of self-learning for adaptation and does so thoroughly, with good agreement between the claimed scope and completed experiments.

**Broader Impact Concerns:**

There are no ethical concerns that warrant a broader impact statement. This work is about a general approach for adapting visual recognition models to reduce generalization error, and it is not particularly suited to a specific positive or negative ethical purpose.
Nevertheless, the authors are encouraged to reconsider their advice to apply self-learning in practice, as it is not guaranteed to improve accuracy.
It may be obvious to those with experience in adaptation research, but a caution to experiment with a given shift before deployment would not be unreasonable.



**Requested Changes:**

**Critical Changes** for Correctness:

- Naming in Sec. 1 & 3: Please clarify "self-learning" as the superset of pseudo-labeling, entropy minimization, and RPL (or not). The current language is a bit confused, as entropy minimization is included in self-learning, but then the text and table captions (see Table 4) for instance then claims self-learning is better than entropy minimization.
- Crediting Related Work in Sec. 2: Please discuss the related test-time adaptation methods listed above that are absent from the related work. If these are not in scope, then please explain how this is so during the discussion phase.
- Attributing Techniques in Sec. 3 & 4: Please note the choice of parameters aligns with TENT, in updating the affine parameters of normalization layers, and not TTT/MEMO/others in the updating of all parameters.
- Attributing Results in Sec. 2, 4, and 6: Please identify which results are new and which are expansions of prior work when reporting results. For instance, the confirmations of short update intervals and affine parametrs for convnets are presented alone, but both were already shown (for fewer architectures, datasets, ...) by TENT. Furthermore, results on gradual adaptation are shown not just by Kumar 2020 but also TTT.

**Minor Changes** for Improvement:

- Sec. 3: Please consider a discussion of parameterization (that is, the choice of affine parameters or not in TENT) and regularization (that is, the information maximization/diversity regularizer in SHOT). These are important choices for the quality of adaptation, as this work confirms for affine parameterization.
- Sec. 4., Tables 1 & 2: Please consider reporting RPL and ENT side-by-side in both tables. This allows the reader to measure their difference, and would reinforce the point that self-learning generally works, even if one or the other may be best for a given evaluation.
- Sec. 2 Related Work: Please make a contribution statement or summarize the aspects on which related work are compared and contrasted. The immediate dive into papers does not orient the reader. Perhaps consider defining source-free and test-time at the beginning of this section or in the introduction.
- Sec. 7 Theory: Please clarify the connection of this model with the difference in small-scale vs. large-scale results or introduce
- Sec. 6, Table 9: Please specify the architecture for a more self-contained table.
- Sec. 7, Table 10: Please include the IN error rate to gauge the degree of shift.



**Strengths And Weaknesses:**

**Strengths**

- This is an informative collection of results on how well self-training/entropy minimization/pseudo-labeling can do and how to apply it well. The experiments cover
  - architectures for image classification that span ResNet, DenseNet, ResNeXt, EfficientNet, and ViTs;
  - pre-training on the standard ImageNet data, on larger-scale uncurated Instagram data, and by domain adaptation schemes;
  - parameterization of updates over all parameters, only affine parameters, or more variants for ViT architectures; and
  - hyperparameters in general like the learning rate and epochs for optimization and the update interval and other settings specifically for RPL.
- Self-learning does indeed work well, and the state-of-the-art results among adaptation methods are worth sharing, in particular for the ImageNet-scale experiments on IN-C/A/R/D.
- The breadth and rigor of the experiments is relevant and necessary: prior works only include a subset of models or shifts in their experiments and they tune and evaluate in varying ways.
  The suggestions for best practices (Sec. 9) are definite, justified, and computationally practical.
- The thoroughness of the experimentation and documentation for reproduction and extension are commendable (see appendices B, C, and D).
- The proposed IN-D dataset makes reasonable and justified design choices, as explained in the text and appendix D, following in the style of "renditions" as a type of shift (as done by IN-R).

**Weaknesses**

- The related works of TTT and Tent are incompletely or incorrectly summarized. This submission evidently delivers a much-expanded scope for experiments with entropy minimization and related self-learning techniques, but it should still appropriately summarize what came before.
    - For TTT, it is worth noting that it included results on gradual adaptation (across severities of IN-C), counter to the "future work" left in Sec. 2 of this work.
  In the results, TTT is likewise able to make use of the full dataset, which is for some reason underlined for RPL only in Sec. 6 of this work.
    - For TENT, the difference with ENT in this work is left unclear, and the existing experiments for TENT are not accurately summarized.
  When are ENT results the same or different than TENT? (For example, a row in Sec. C.4's Table 19 is labeled TENT, but tables in the main paper are labeled ENT.)
  While TENT experiments are summarized as a "single model [...] on ImageNet-C", TENT experimented with multiple models (ResNet-50, ResNet-26, the SAN attention model, and the implicit DEQ model) and on multiple datasets (CIFAR-10/100 corruption datasets, VisDA-C and GTA-CityScapes simulation-to-real datasets, and the digits datasets SVHN, MNIST, USPS).
- Prior test-time and self-learning schemes for updating models are absent from the related work and results: see [TTT++ NeurIPS'21](https://proceedings.neurips.cc/paper/2021/hash/b618c3210e934362ac261db280128c22-Abstract.html), [T3A NeurIPS'21](https://proceedings.neurips.cc/paper/2021/hash/1415fe9fea0fa1e45dddcff5682239a0-Abstract.html), [BUFR ICLR'22](https://arxiv.org/abs/2107.05446), [EATA ICML'22](https://arxiv.org/abs/2204.02610), and [confidence maximization with input transformation](https://arxiv.org/abs/2106.14999).
  These should at least be referenced, though all need not be compared against on ImageNet, where they do not report results at ImageNet scale (T3A, BUFR).
  Of course evaluating them on ImageNet would be a further contribution, but it is reasonable to rule them out as beyond the scope of this empirical study.
- (Minor) The theoretical component is vastly simplified from the experiments studied, although this is admitted by the text. The real issue is that is not an explanation for the highlighted observation that "different self-learning schemes are optimal for small-scale vs. large-scale datasets". This weakness is minor, as the theoretical content is a bonus on top of this primarily empirical work, but the purpose of the theory could be better indicated.
- (Minor) The proposed IN-D dataset (Sec. 8) is derived from the VisDA benchmark, so the degree of contribution is lessened. Nevertheless it is a contribution, simply because it is harder, but more importantly because it differs from IN-R in separating out the types of shift. This allows future research to compare single-target vs. multi-target adaptation by evaluating shifts separately or jointly.

**Summary**

This work _delivers on its claimed contributions_, and _needs only correct the gaps and errors in its coverage of related work_ to be ready for publication.
I expect it will serve as an informative publication for researchers in adaptation and robustness. The contributions are:

- experiments of substantial scope across datasets, model architectures and parameters, and shifts;
- a new to source-free adaptation loss, RPL, derived from the GCE from label noise robustness research;
- a new dataset, IN-D, derived from the VisDA dataset for UDA, but with a different protocol for source-free adaptation, and error rates to verify the shifts are difficult; and
- a rigorous experimental protocol that is strictly adhered to and is accompanied by clear advice on best practices.

**Miscellaneous Feedback and Questions**

- Sec. 3: Is "standard cross entropy" more precisely described as softmax cross entropy? There are other cross entropies, of course, but all of the deep classifiers in this study make softmax outputs.
- Sec. 4, Table 1: What is the result for with R-50 on IN-A? Does self-learning help or harm? This is of interest, as this dataset should be adversarial to self-learning in particular.
- Sec. 4, Table 2: It would be nice to match the architecture of the vanilla and UDA models, to assess how much UDA vs. TTA is helping. As it stands, the vanilla baseline is better than UDA.
- Sec. 5: For the ViT result, please clarify the loss. Is it RPL or ENT, or purely DINO?
- Sec. 6: Several classes of results were already demonstrated by TENT, and so perhaps should be identified as such? This is the case of "usage of the full dataset without a threshold", "short update intervals", and "adaptation of only affine parameters". To be clear, the confirmatory and broader evidence is valuable, but it is nevertheless not the first.
- Sec. 9 Conclusions: Can we recommend self-learning to practitioners without further understanding of stability and deployment data distributions? The topic is certainly of interest, but it may not be ready to deploy. A more nuanced statement could be appropriate.
- Sec. D.5: Consider discarding the results without the mapping of permissible classes. Leading the nonsensical/irrelevant predictions first is confusing.
- Sec. E.1 & E.2: The limitations on mixed shifts and architecture are worth pointing to from the main paper. Consider a footnote or pointer from the results section.
- Sec. E.3: How does this square with the digits results for TENT? To briefly mention them here, TENT shows improved test-time adaptation accuracy for 2/3 datasets, suggesting that models can perhaps do better by specializing to target than jointly training on source and target.

Optional Suggestions for Experiments

- ENT seems to do better on smaller models and datasets, such as on CIFAR-10-C and digit datasets. Is this specific to the data are architecture? How do ENT and RPL compare on R-18 on IN-C, for instance, to measure a smaller model on a larger dataset?
- How do RPL and ENT compare in the online and test-time setting? Most results are for the offline and source-free setting of multi-epoch optimization, which is fine, but the most efficient setting may also be of interest.

Trivial

- Sec. C.6: Why is TTT labeled contrastive? It is a self-supervised method with an auxiliary task, but it is not a contrastive learning method.
- Sec. D.3: Should GCE be RPL?
- Typo, pg. 2: "both in" is written for more than two sets

---

> ### Author Response · Authors · 2022-09-23
> **Authors’ response to Reviewer iPbT (1/7) [Critical changes]**
>
> First, we would like to thank Reviewer iPbT for the thorough and highly detailed review, many thoughtful suggestions and the appreciation of our work. We believe that incorporating the requested changes has significantly improved the paper, in particular with respect to our related work section and additional relevant baselines.
> We addressed your questions with a paper update and comments below. All changes in response to your suggestions are highlighted in **red** in the manuscript, except for changes in the Tables due to formatting. We would love to hear your opinion on the changes, and whether your concerns have been resolved.
>
> ## Responses to “Critical Changes”:
>
> ### “Clarify “self-learning” as the superset of PL and ENT”.
>
> Thank you for this suggestion. We added the following footnote to Sec. 1:
> > “We define self-learning as the superset of different pseudo-labeling variants and entropy minimization to highlight the fact these methods do not require ground truth labels for adaptation.”
>
> We also added the following sentence to Section 3:
> > ”Tsypkin defines self-learning as “learning when there is no external indication concerning the correctness of the response of the automatic system to the presented patterns”', and thus, we use this term as a superset of pseudo-labeling and entropy minimization to highlight that these methods can be used for adaptation to unlabeled data.”
>
> Both changes are displayed in **green** in the revised manuscript because Reviewer 4Aye also asked to provide clarity on the term “self-learning”. We also increased consistency in how we use this term across the manuscript.
>
> References:
> [1] Ya Tsypkin “Self-Learning-What Is It?”, 1968
>
> ### [Sec.2] “Discuss further related work, add as baselines where applicable, improve structure of related work.”
>
> Thank you for your suggestions. We have significantly expanded our related work section based on the proposed papers, such that we now begin with a summary of other test-time adaptation papers which we include as baselines in the paper. The new additions to the related work section take more than a page and thus, we decided to not post them here for space reasons. Please refer to the revised manuscript.
>
> We also expanded Table 4 to include many more baselines to compare self-learning to. Please see the Table below or refer to the revised paper. We want to note that SRL (Mummadi et al., 2021) is unpublished work and EATA (Niu et al., ICML 2022) is concurrent to this submission. We will appreciate the pointer and included discussions and comparisons to these methods, as they are self-learning methods on their own, and corroborate the central claim of our paper.
>
> We reworked the paragraph in Sec. 6 discussing our baseline comparisons in the following way:
>
> > “**Self-learning outperforms other test-time adaptation techniques on IN-C (Tables 4 and 5).** The main point of our paper is showing that self-learning is effective across datasets, model sizes and pretraining methods. Here, we analyze whether our simple techniques can compete with other state-of-the-art adaptation methods. Overall, we find that self-learning outperforms several state-of-the-art techniques, but underperforms in some cases, especially when self-learning is combined with other techniques.
>
> By rigorous and fair model selection, we are able to improve upon TENT (Wang et al., 2021), and find that RPL performs better than entropy minimization on IN-C. We also compare to BN adaptation (Schneider et al., 2020), and find that 1) self-learning further improves the performance upon BN adapt, and 2) self-learning improves performance of a model pretrained on a large amount of data (Mahajan et al., 2018) (Table 3) which is a setting where BN adapt failed.
>
> ENT and RPL outperform the recently published EATA method (Niu et al., 2022). In EATA, high entropy samples are excluded from optimization, and a regularization term is added to prevent the model from forgetting the source distribution. EATA requires tuning of two additional parameters: the threshold for high entropy samples to be discarded and the regularization trade-off parameter β.
>
> [continued in the next comment for space reasons]

---

> > ### Comment · Reviewer_iPbT · 2022-09-27
> > **Thank you for the thorough response.**
> >
> > The response has addressed all of the critical changes, minor changes, and other feedback I provided. I commend the authors for their timely, thorough, and constructive response and revision of the submission.
> >
> > While a few results are still pending, such as how self-learning copes with adversarially filtered inputs on ImageNet-A, I consider these to be merely bonuses, and not at all required.
> >
> > In summary, I am entirely satisfied with the discussion in this thread and with the revisions made to the submission. The submission is now longer, but the additional material earns its place on the page: the text is now more self-contained and clear and it now does a better job of connecting the reader with prior and related work.

---

> > > ### Author Response · Authors · 2022-09-28
> > > **Authors’ response to Reviewer iPbT (post-discussion)**
> > >
> > > Thank you for appreciating our revision, we are very happy to have resolved your concerns.
> > >
> > > Adapting a ResNet50 on ImageNet-A with RPL increases the error from 0% to 0.13% (chance level: 0.1%). Thus, an unadapted ResNet50 has 0% accuracy on ImageNet-A by design and this error “is increased” to chance-level with self-learning. Since all labels on ImageNet-A are wrong by design, predicting wrong labels as pseudo-labels does not lead to improvements beyond restoring chance-level performance.
> > >
> > > We have added this short insight to the manuscript. We also added the missing number for ResNet18 for adaptation with ENT to Table 2.

---

> ### Author Response · Authors · 2022-09-23
> **Authors’ response to Reviewer iPbT [2/7] [Critical changes]**
>
> ### [Continuation of [Sec.2] “Discuss further related work, add as baselines where applicable, improve structure of related work.”]
>
> RPL, ENT and simple hard PL outperform TTT (Sun et al., 2020); in particular, note that TTT requires a special loss function at training time, while our approach is agnostic to the pre-training phase. A detailed comparison to TTT is included in Appendix C.6. Mummadi et al. (2021) (SLR) is unpublished work and performs better than ENT and RPL on the highest severity of IN-C. SLR is an extension of entropy minimization where the entropy minimization loss is replaced with a version to ensure non-vanishing gradients of high confidence samples, as well as a diversity regularizer; in addition, a trainable module is prepended to the network to partially undo the distribution shift. Mummadi et al. (2021) introduce the hyperparameters δ as the trade-off parameter between their two losses, as well as κ as the momentum in their diversity regularization term. The success of SLR over ENT and RPL shows the promise of extending self-learning methods by additional objectives, and corroborates our findings on the effectiveness of self-learning.
>
> We also compare our approach to Meta Test-Time Training (MT3, Bartler et al., 2022), which combines meta-learning, self-supervision and test-time training for test-time adaptation. We find that both ENT and RPL perform better than MT3: using the same architecture as Bartler et al. (2022), our best error on CIFAR10-C is 18.0% compared to their best result of 24.4%. When exchanging GroupNorm layers (Wu & He, 2018) for BN layers, the error further reduces to 13.1% (Table 11). We thus find that self-learning is more effective when adapting affine BN parameters instead of GN layers, which is consistent with the findings in Schneider et al. (2020). We included a detailed comparison to Bartler et al. (2022) in Appendix C.7.
>
> TTT+++ (Liu et al., 2021) outperforms both ENT and RPL on CIFAR10-C. Since Liu et al. (2021) do not report results on IN-C, it is impossible to judge whether their gains would generalize, although they do report much better results compared to TENT on Visda-C, so TTT+++ might also be effective on IN-C. Similar to TTT, TTT+++ requires a special loss function during pretraining and thus, cannot be used as an out-of-the-box adaptation technique on top of any pretrained checkpoint. Bottom-Up Feature Restoration (BUFR; Eastwood et al., 2022) outperforms self-learning on CIFAR10-C. The authors note that BUFR is applicable to dataset shifts with measurement shifts which stem from measurement artefacts, but not applicable to more complicated shifts where learning new features would be necessary”.
>
> The Tables below represent the new Tables 4 and 5 in the revised manuscript.
>
> | mCE [\%] on IN-C test ($\searrow$) | w/o adapt | BN Adapt | TENT | EATA(lifelong) | RPL          | ENT |
> |------------------------------------|-----------|----------|------|----------------|---------------|------|
> | ResNet50                           | 76.7      | 62.2     | 53.5 | 51.2           | **50.5** | 51.6 |
>
> |  | literature results | |our results  | | |
> |--|---|--|---|--|--|
> | | w/o adapt | w/ adapt | w/o adapt | w/ adapt (RPL) | w/ adapt (ENT) |
> | top1 error [\%] on IN-C test, sev. 5($\searrow$) |  | |  |
> | TTT, ResNet18 (Sun et al., 2020) | 86.6 | 66.3           | 85.4 | **61.9**   | tbd. |
> | SLR, ResNet50 (Mummadi et al., 2021)  | 82.0 | (**46.9**) | 82.0 | 54.6 | 54.7 |
> | |  |  |  |  | |
> | top1 error [\%] on CIFAR10-C ($\searrow$) |   |    |  |
> | MT3, WRN-26-1-GN (Bartler et al., 2022) | 35.7 | 24.4  | 18.6 | 18.4  | **18.0** |
> | TTT+++, ResNet50 (Liu et al., 2021) | 29.1 | **9.8**  | 24.9 | 14.6  | 12.4 |
> | BUFR, ResNet18 (Eastwood et al., 2022) | 42.4 | **10.6**  | 25.5 | 13.4  | 12.9 |

---

> ### Author Response · Authors · 2022-09-23
> **Authors’ response to Reviewer iPbT (3/7) [Critical changes]**
>
> ### [Sec. 6] “Please attribute other works who showed some of the reported results on other architectures/ datasets first.”
>
> Thanks a lot for pointing us to this. We will discuss the points individually below:
>
> **usage of the full dataset without a threshold (in Wang et al., 2021)**:
>
> Wang et al. 2021 show that TENT outperforms PL on the highest severity of CIFAR-C 10/100 (cf. Table 2, Wang et al. 2021). We substantially extend this finding. We still acknowledge that we should have better attributed the previous work, and changed our text accordingly:
>
> >  “We corroborate the results of Wang et al. (2021) who showed that TENT outperforms standard hard labeling with a threshold on the highest severity of CIFAR10-C and CIFAR100-C. We show that this result transfers to IN-C, for a variety of thresholds and pseudo-labeling variants.”
>
> **adaptation of only affine parameters**:
>
> We adapted the text for better attribution of Wang et al. (2021). However, note that Wang et al. claim that “Updating the full model parameters θ never improves over the unadapted source model”. No experimental details or numbers are given in their paper, but we assume that this finding was generated on CIFAR-C during development of the method. Our paper provides evidence against this statement for larger scale data and other models, on both IN-C (where small gains can be observed), and especially for other model architectures (e.g., the vision transformer we adapted) where full model adaptation is actually performing best. We better attributed Wang et al. in the revised manuscript, and highlighted our contribution:
>
> > “Wang et al. (2021) also used the affine BN parameters for test-time adaptation with TENT, and report that last layer optimization can improve performance but degrades with further optimization. They suggest that full model optimization does not improve performance at all. In contrast, we find gains with full model adaptation, but stronger gains with adaptation of only affine parameters.”
>
> **adaptation across architectures**:
>
> > “The finding that self-learning can be effective across model architectures has also been made by Wang et al. (2021) who show improved adaptation performance on CIFAR100-C for architectures based on self-attention (Zhao et al., 2020) and equilibrium solving (Bai et al., 2020), and also by (unpublished) Mummadi et al. (2021) who showed adaptation results for TENT and TENT+ which combines entropy minimization with a diversity regularizer for a DenseNet121 (Huang et al., 2017), a MobileNetV2 (Sandler et al., 2018), a ResNeXt50 (Xie et al., 2017), and a robust model trained with DAug+AM on IN-C and IN-R.”
>
> ### “Please set the optimal hyperparameters (short updates/adaptation of affine parameters/…) into context with related work.”
>
> We added a new paragraph as well a new Table to summarize the optimal hyperparameters we found as well as credit previous work who used them:
>
> > “**Our analysis confirms hyperparameter choices from the literature** Having searched over a broad space of hyperparameters and self-learning algorithms, we identified ENT and RPL as the best performing variants across different model sizes, architectures and pretraining techniques. We summarize the most important hyperparameter choices in Table 11 and compare them to those used in the literature when applying self-learning for test-time adaptation. Wang et al. (2021) identified that on CIFAR-C, entropy minimization outperforms hard PL, and found that affine adaptation of BN parameters works better than full model adaptation, and adapted to the full dataset since TENT does not have a threshold in contrast to hard PL. EATA (Niu et al., 2022) and SRL (Mummadi et al., 2021) are extensions of TENT, and thus, followed their hyperparameter choices. EATA does not adapt to the full dataset as they do not adapt to very similar samples or samples with high entropy values. MEMO (Zhang et al., 2021) adapts to a single sample and adapts all model weights. It would be interesting to study whether the performance of MEMO can be improved when adapting only affine BN parameters.”
>
> The Table below represents the new Table 12 in the revised manuscript.
>
> | Method | short updates | adapt affine params | use BN instead of GN | adapt to full dataset |
> |--|--|--|--|--|
> | TENT (Wang el al., 2021)  | &check; | &check;| &check;| &check;|
> | EATA (Niu et al., 2022) | &check; | &check; | &check; | &cross; |
> | SRL (Mummadi et al., 2022)   | &check; | &check; | &check; | &check; |
> | MEMO (Zhang et al., 2021)    | &cross; | &check; | &check; | &cross; |
> | RPL/ ENT (ours)| &check; | &check; | &check; | &check; |
>
> Please let us know if we missed any important hyperparameters or missed to attribute more results to previous work.

---

> ### Author Response · Authors · 2022-09-23
> **Authors’ response to Reviewer iPbT (4/7) [Minor changes]**
>
> ## Responses to “Minor Changes”:
>
> ### “Sec. 3: Please discuss parametrization (affine vs full model adaptation) and regularization (information maximization/diversity regularization).
>
> We have moved the paragraph describing the adaptation parameters from Sec. 4 to Sec.3:
>
> > “**Adaptation parameters.** Following Wang et al. (2021), we only adapt the affine scale and shift parameters γ and β following the batch normalization layers (Ioffe & Szegedy, 2015) in most of our experiments. We verify that this type of adaptation works better than full model adaptation for large models in an ablation study in Section 6.”
>
> We added a paragraph discussing additional regularization to Sec. 3.
> > “**Additional regularization in self-learning** Different regularization terms have been proposed as a means to stabilize entropy minimization. Niu et al. (2022) propose an anti-forgetting weight regularization term, Mummadi et al. (2021); Chen et al. (2022) add a diversity regularizer, and Chen et al. (2022) use an additional consistency regularizer. These methods show improved performance with these regularization terms, but also introduce additional hyperparameters, the tuning of which significantly increases compute requirements. In this work, we do not experiment with additional regularization, as the main point of our analysis is to show that pure self-learning is effective at improving the performance over the unadapted model across model architectures/sizes and pre-training schemes. For practitioners, we note that regularization terms can further improve the performance if the new hyperparameters are tuned properly.”
>
> ### “Show both robust pseudo-labeling and entropy minimization results in Table 1 & 2”
>
> Thank you for the suggestion, we updated Tables 1+2 to show both ENT and RPL. Please refer to the revised manuscript.
>
> ### “Sec. 7 Theory: Please clarify the connection of this model with the difference in small-scale vs. large-scale results or introduce. The theoretical component is vastly simplified from the experiments studied. [...] The purpose of the theory could be better indicated.”
>
> We aimed to overall better understand the learning dynamics and aimed for the simplest possible model that still replicates essential properties that can be observed on real data. A key theme of our paper is to contrast entropy minimization and pseudo-labeling techniques and understanding their effectiveness and failure modes.
>
> The proposed two-point model covers both of these model classes (via the option to have a shared student/teacher, vs. a stop-gradient operation) and allows to continuously interpolate (via the temperature parameter, which is also used in empirical work) between hard entropy minimization/pseudo-labeling and soft entropy-minimization/pseudo-labeling.
>
> Interestingly, the learning dynamics (when is training stable, when unstable) that are observed and validated on CIFAR10-C can be recovered by this simple model. The finding suggests that the learning dynamics are tightly connected to the loss function and the ratio between student and teacher temperatures, rather than the exact properties of the dataset and model, which is interesting for additional theoretical analysis.
>
> The finding has interesting implications on empirical work, and has been corroborated by a number of recent works [Caron et al, 2021] that move away from a 1:1 ratio between teacher and student temperatures, and sharpen the predictions of the teacher model by a lower temperature.
>
> Please let us know if this clarifies your concern.
>
> ### “Sec. 6, Table 9: Please specify the architecture for a more self-contained table.”
>
> Thank you for this suggestion. We added the architecture to Table 9, and also to Table 8 to improve clarity.
>
> ### “Sec. 7, Table 10: Please include the IN error rate to gauge the degree of shift.”
>
> Thank you for this suggestion, we updated Table 10 to show the IN error rate. Please refer to the revised manuscript.

---

> ### Author Response · Authors · 2022-09-23
> **Authors’ response to Reviewer iPbT (5/7) [Miscellaneous Feedback and Questions 1/2]**
>
> ## Responses to “Miscellaneous Feedback and Questions”:
>
> ### [Sec.3] standard vs softmax cross-entropy
> We renamed the “standard cross-entropy loss” with “softmax cross-entropy loss” as suggested.
> ### [Sec. 4] Matching the architecture of the vanilla and UDA methods for Table 2. The vanilla performs better than the UDA model.
> The WRN-26-16 model is larger than the WRN-28-10 model and thus, one would expect it to actually perform even better, which would further surpass the UDA-SS result. Robustbench does not provide a trained checkpoint for the WRN-28-10 model, and thus, we trained the vanilla model with the widely used CIFAR10 training code available at https://github.com/kuangliu/pytorch-cifar (1.9k forks, 4.8k stars). The clean accuracy on the CIFAR10 test set for this model is at 96.5%. The UDA-SS model has an average accuracy of 82.6% on clean CIFAR10 where the average was taken over all corruptions and severities as UDA-SS is co-trained on source and target, and the target CIFAR10-C dataset has 75 distribution shifts. The adaptation accuracies with RPL and ENT can be seen below.
>
> || w/o adapt | w/ adapt ($\Delta$) | w/ adapt ($\Delta$) | w/ adapt ($\Delta$) |
> |--|--|--|--|--|
> | top1 error [\%] on CIFAR10-C ($\searrow$) || BN adapt | RPL| ENT|
> |WRN-26-16 UDA-SS (Sun et al., 2019) |   27.7 | 19.9 (-7.8) | 18.2 (-9.5) | 16.7 (-11.0) |
> |WRN-26-16  vanilla | 24.2 | 12.2 (-12.0)  |  11.8 (-12.4) | 11.2 (-13.0) |
>
> We see that the vanilla model performs better than the UDA-SS model, both with and without adaptation. We think that the UDA-SS model would need hyperparameter tuning; we did not perform any tuning for this model, especially because the authors provided scripts with hyperparameters they found to be optimal for different setups. In addition, the clean accuracy of the vanilla model on CIFAR10 (96.5%) is much higher than the average clean accuracy of the UDA-SS model (82.6%), which may explain or imply generally higher robustness under distribution shift [1]. The finding that self-learning is more effective in the vanilla model compared to the UDA-SS model points towards the hypothesis that the network weights of the vanilla model trained on the source distribution are sufficiently general and can be tuned successfully using only the affine BN parameters, while the weights of the UDA-SS model are already co-adapted to both the source and the target distribution, and thus, self-learning is less effective. We observe that BN adaptation is also less effective for the UDA-SS models compared to the vanilla trained models.
>
> Our main point we wanted to make with Tables 1+2 is that self-learning is effective on top of checkpoints pretrained in different ways, and this point still stands. We have added the vanilla WRN-26-16 model to Table 2, and included the discussion above to the text.
>
> [1] Miller et al. “Accuracy on the line: on the strong correlation between out-of-distribution and in-distribution generalization”, PMLR 2021
>
> ### [Sec. 5] Adaptation objective in the ViT experiments
>
> For our ViT experiments, we only experimented with the DINO loss, and did not perform TTA with ENT or RPL. We clarified this in the revised manuscript in the following way: “We highlight that we specifically test the self-supervised DINO objective for its practicality as a test-time adaptation method, and did not switch the DINO objective for ENT or RPL to do test-time adaptation.”
>
> ### [Sec. D.5] Consider discarding the results without the mapping of permissible classes.
>
> Thank you for this suggestion, we removed the results in the first column of Table 35.
>
> ### [Sec. E3] How do the results in E.3 compare with the results reported on the digits datasets in Wang et al. (2021), Table 3?
>
> The comparison between the results in Table 3 in Wang et al. (2021) and our results in E.3. is not straight-forward as we start from the jointly optimized checkpoint to test whether we can further improve it with self-learning, while Wang et al. (2021) start from the model pretrained on source and adapt it to the target distribution. Our results in E.3. show that UDA techniques using self-learning at their core cannot be effectively adapted with self-learning at test time (while models trained with non-self-learning UDA methods could be further improved, e.g., DANN).
>
> Further, on three out of four shifts, the baseline accuracy numbers are already above 98% which makes further gains hard.
>
> ### [Trivial, Sec. D.3]:
>
> Thanks a lot for pointing these out. We performed the following changes:
>
> Should GCE be RPL? -> We have changed Figure 8 accordingly.
> Typo on pg.2 -> We fixed the typo by removing “both”
> Sec. C.6. Why is TTT labeled contrastive? We removed the word “contrastive” from the title in C.6.

---

> ### Author Response · Authors · 2022-09-23
> **Authors’ response to Reviewer iPbT (6/7) [Miscellaneous Feedback and Questions 2/3]**
>
> ### [Sec. 9] “Sec. 9 Conclusions: Can we recommend self-learning to practitioners without further understanding of stability and deployment data distributions? The topic is certainly of interest, but it may not be ready to deploy. A more nuanced statement could be appropriate.”
>
> Thanks for pointing this out. Of course, there are no guarantees that self-learning will help in scenarios not tested in this work. Instead, we hope that our work provides evidence that self-learning is a suitable candidate to cope with distribution shifts given its simplicity and its performance in many scenarios. To provide a more nuanced statement, we changed the last sentence in the conclusion to:
>
> > “Across the large diversity of (systematic) distribution shifts, architectures and pre-training methods we tested in this paper, we found that self-learning almost universally improved test-time performance. An important limitation of current self-learning methods is the observed instability over longer adaptation time frames. While we mitigate this issue through model selection (and showed its robustness across synthetic and natural distribution shifts), this might not universally hold across distribution shifts encountered in practice. Concurrent work, e.g. Niu et al. (2022) tackle this problem through modifications of self-learning algorithms, and we think this direction will be important to continue to explore in future work. That being said, we hope that our results encourage both researchers and practitioners to experiment with self-learning if their data distribution shifts. ”
>
> ### [Sec. E.1 & E.2]: The limitations on mixed shifts and architecture are worth pointing to from the main paper. Consider a footnote or pointer from the results section.
>
> We agree and have added a sentence on WILDS which has also been requested by Reviewer qrwv. Considering the BigTransfer results, we actually have one more point of comparison, as Reviewer 4Aye asked us to compare to Bartler et al. [1] who used an architecture with GroupNorm layers to do TTA on CIFAR10 -> CIFAR10-C. We tested their architecture both with GN and BN layers and found that the unadapted GN architecture was more robust to the distribution shift compared to the model with BN but the model with BN had better performance after adaptation. With this, we corroborate the results by Schneider et al. [2] who found the same for simple BN adaptation. Please see our full comparison with Bartler et al. in Appendix C.7. in the revised manuscript.
>
> To highlight the suitability of GN vs BN for test-time adaptation, we added a paragraph to Sec. 6, as well as a new Table:
>
> > “Affine BN parameters work better for test-time adaptation compared to GN parameters. (Table 10). Schneider et al. (2020) showed that models with batch normalization layers are less robust to distribution shift compared to models with group normalization (Wu & He, 2018) layers. However, after adapting BN statistics, the adapted model outperformed the non-adapted GN model. Here, we show that these results also hold for test-time adaptation when adapting a model with GN or BN layers. We show that a WideResNet-26-1 (WRN-26-1) vanilla model with BN layers pretrained on clean CIFAR10 has a much higher error on CIFAR10-C than the same model with GN layers, but it has a much lower error after adaptation. The full results for the WRN-26-1 model can be found in Appendix C.7. Further, we test the pretrained BigTransfer (Kolesnikov et al., 2020) models which have GN layers, and find only small improvements with RPL , and no improvements with ENT. There are no pretrained weights released for the BigTransfer models which have BN layers, thus, a comparison similar to the WRN-26-1 model is not possible. A more detailed discussion on our BigTransfer results as well as a hyperparameter selection study can be found in Appendix E.2.”
>
> The Table below represents the new Table 11 in the revised manuscript.
>
> || number of parameters| w/o adapt | w/ adapt ($\Delta$) | w/ adapt ($\Delta$) |
> |---|--|--|--|--|
> | top1 error [\%] on CIFAR10-C ($\searrow$) ||| RPL| ENT|
> | WRN-26-1-BN (Zagoruyko \& Komodakis, 2016)| $1.5\times10^6$ | 25.8| 15.1 (-10.7)| 13.1 (-12.7)|
> | WRN-26-1-GN  (Bartler et al., 2022)| $1.5\times10^6$ | 18.6| 18.4 (-0.2)| 18.0 (-0.6)|
> |  || | ||
> | mCE [\%] on IN-C test ($\searrow$)|||||
> | ResNet50 BigTransfer (Kolesnikov et al., 2020)| $2.6\times10^7$ | 55.0| 54.4 (-0.6)| 56.4 (+1.4)|

---

> ### Author Response · Authors · 2022-09-23
> **Authors’ response to Reviewer iPbT (7/7) [Miscellaneous Feedback and Questions 3/3]**
>
> ### Why does ENT perform better on small datasets while RPL is more effective on
> ### large datasets? Is this specific to the dataset or the architecture?
>
> We think that model capacity is not the cause of this dichotomy. Our CIFAR10 models actually have more parameters than our ImageNet models in some cases: A ResNet50 has fewer parameters than the WRN-28-10 or the WRN-26-16. We have now also added results for a ResNet50 and a ResNet18 on CIFAR10-C for baseline comparisons, and we see that both models perform better with ENT on CIFAR10-C. Thus, we think this effect is related to the dataset, and might be explained with the number of classes in the respective datasets: Some IN classes are very similar to each other, and maximizing confidence may work worse than in datasets with coarse and well-separated classes, such as CIFAR10. On IN-D which has fewer classes than IN but more than CIFAR10, we observe that ENT and RPL perform similarly and there is no clear winner like on IN-C/CIFAR10-C.
>
> ### Missing results:
>
> - [Sec. 4] ResNet50 adaptation on ImageNet-A: Thank you for the pointer. We are currently running this experiment and will post and discuss the result later during the discussion period.
> - [Table 5]. Adaptation of a ResNet18 model with ENT to compare to TTT for completeness. We are currently running this experiment and will post the result later during the discussion period.

---

### Author Response · Authors · 2022-11-09
**Thank you!**

Dear reviewers and Area Chair,

We are very excited that our paper has been accepted at TMLR and would like to thank you for the constructive feedback and helpful suggestions which improved the quality of our paper. We have now uploaded the camera-ready version.

---

### Author Response · Authors · 2022-12-15
**Typo in equation 21**

Thanks to Zhiheng Li for spotting the following typo in equation 21. The correct formulation should be:
$DE_d^f = \frac{E_{d}^f}{E_{d}^{AlexNet}},$
$\mathrm{mDE} = \frac{1}{D} \sum_{d=1}^D \mathrm{DE}_d^f.$

---

### Decision · Action_Editors · 2022-10-28

**Recommendation:** Accept as is

**Comment:**

At the end of the discussion period, all reviewers were leaning towards acceptance. The authors meticulously responded to all of the reviewers' concerns and added large amounts of convincing experiments for each relevant concern during the rebuttal phase. Overall I believe that this submission fits TMLR well, due to its reproducible claims that are supported by a detailed and extensive set of experiments.

**Audience:**

Authors present extensive evaluations for a setting between robustness and unsupervised data augmentation for two well known methods (pseudo-labelling and entropy-minimization). I believe that the findings in this paper will be of interest to researchers in this relatively nascent setting. Due to the closeness of this setting to the two more established ones (i.e. robustness and unsupervised data augmentation), I believe that the paper should also be of interest to the larger deep learning community that care about OOD generalization and adaptation.

**Claims And Evidence:**

Authors have evaluated well known methods such as pseudo-labelling and entropy-minimization for the task of adaptation on robustness benchmarks. Their comparisons include other test-time adaptation methods, methods for improving robustness, and unsupervised data augmentation. Even though the authors achieve improved performance on robustness benchmarks, these results should not be compared to other methods primarily created for robustness since the proposed method allows for the model to adapt to new datasets/domains.

Authors claim to show the efficacy of pseudo-labelling and entropy-minimization for adaptation to domain shift, and claim to present a wide range of experiments that cover different architectures, pre-training protocols, and domain shifts. I agree with the reviewers that these claims are supported by the experiments in the paper.